# Individual differences in information-seeking

Christopher. A. Kelly 1,2✉ & Tali Sharot 1,2✉

Vast amounts of personalized information are now available to individuals. A vital research challenge is to establish how people decide what information they wish to obtain. Here, over five studies examining information-seeking in different domains we show that information-seeking is associated with three diverse motives. Specifically, we find that participants assess whether information is useful in directing action, how it will make them feel, and whether it relates to concepts they think of often. We demonstrate that participants integrate these assessments into a calculation of the value of information that explains information seeking or its avoidance. Different individuals assign different weights to these three factors when seeking information. Using a longitudinal approach, we find that the relative weights assigned to these information-seeking motives within an individual show stability over time, and are related to mental health as assessed using a battery of psychopathology questionnaires.

[1] Department of Experimental Psychology, University College London, London WC1H 0AP, UK. [2] Max Planck University College London Centre for Computational Psychiatry and Ageing Research, London WC1B 5EH, UK. ✉email: christopher.kelly.16@ucl.ac.uk; t.sharot@ucl.ac.uk

Thanks to advances in technology, massive amounts of information are now easily accessible. This includes personalized information about people's past, present and future. Individuals must make many decisions regarding which information they would like to receive and which they would rather avoid. It is unclear how people make these choices.

Despite the relevance of this question to domains such as health, politics and science, we know surprisingly little about what drives information seeking. Nor do we have a clear understanding of why an individual decides to seek out particular information, while another actively avoids it. For example, a recent study[1] found that approximately half of individuals surveyed wanted to know if they had a genetic predisposition to cancer, while the other half did not; half wanted to know the estimated global temperature in 2100, half did not; half wanted to know the amount of calories in meal options, half did not. Here, we characterize and quantify motives of information seeking and show how they explain individual differences in information-seeking choices.

We have recently proposed a theory which characterizes the key motives for information seeking[2]. According to this theory, when deciding whether to seek information, people first estimate what the information will reveal and then estimate the expected impact of that information on their action, affect and cognition. With regards to action, the prediction is that people want information more when it can aid in selecting action that will help gain rewards and avoid harm[2]. For example, people would be more likely to want to know about automobile safety ratings if they are about to buy a car, as the information can inform their purchasing decision. With regards to affect, all else being equal, people will be more likely to want information when they expect knowledge to make them feel better than ignorance (and vice versa)[2–10]. For example, the prediction is that a student would be more likely to want to know their mark on an exam if they believe they had done well. With regards to cognition, people will want information about concepts they think of often[2]. This is because such information is especially relevant to their internal representation of their world and highly connected with many other concepts[2]. For example, the prediction is[2] that a person who thinks about dogs frequently, would be more interested in learning whether dogs are related to wolfs compared to someone who rarely thinks about dogs.

The estimated impact of information on action, affect and cognition is referred to as instrumental utility, hedonic utility and cognitive utility, respectively[2]. Each of these estimates can be positive (increasing information seeking), negative (increasing information avoidance) or zero (inducing indifference)[2]. We hypothesized that these estimates are integrated into a computation of the value of information, which will trigger information seeking or its active avoidance[2]. Here, over five studies testing 543 participants we provide an empirical test of this theory. To examine if the theory is domain general or domain specific, we test information seeking in three different domains—information about self-traits, finance and health.

We had further proposed that each of the three factors may be weighted differently, influencing the decision to seek or avoid information to different degrees[2] (Fig. 1a). Individual differences in information seeking may be related to the different weight individuals assign to each motive. For example, certain individuals may care most about the instrumental utility of information, whereas others may care most about the need to regulate their affective state, while other may assign equal weight to all three motives when seeking information, etc. Here, we quantify those differences and examine to what degree they are stable, or change, over time within and across domains, by conducting three longitudinal studies.

We had hypothesized that the weights people assign to each motive are related to self-reported mental health[2]. The reason for this hypothesis is that many psychopathology symptoms can be broadly characterized as problems in affective processes, cognitive functions as well as action planning and execution[11,12]. Abnormalities in these domains may reveal themselves in the type of information people choose to seek or avoid. For example, depression is characterized by a reduction in the belief that one has agency over outcomes[13], which may lead to a reduction in the impact of instrumental utility on information seeking. As poor mental health is often associated with problems related to self-perception and thoughts regarding the self[14–20], we test the relationship between mental health and information seeking in the domain of self-referential knowledge. If indeed psychopathology symptoms are related to specific patterns of information seeking, there is potential for using measured markers of information seeking to diagnose mental health problems.

Given this rich potential, it is surprising how limited our knowledge is of the links between mental health and information seeking. In fact, despite information seeking being central to human behaviour, we know remarkably little about how to quantify it or the mechanisms that underlie it. To address these unknowns, we conducted five studies in which participants were asked to indicate whether they would want to receive 40 pieces of information. In Experiment 1, 2 and 5 the information was related to self-traits, in Experiment 3 to finance and in Experiment 4 to health. Participants also provided ratings which served as proxies for the instrumental, hedonic and cognitive utility they assigned to each potential piece of information. These proxies were then used to quantify participants' information-seeking motives and explain individual differences in participants' choices. Experiment 1 and 3 were longitudinal studies that enabled us to quantify the stability of the motives over time within an individual and domain, and Experiment 4 examined stability over time across domains. Additionally, in Experiments 1 and 2 we assessed participants' mental health using a battery of self-report psychopathology questionnaires[21–29] and examined these responses for an association between mental health and information-seeking motives. In particular, we implemented a dimensionality approach[30–32], which considers the possibility that a specific symptom is predictive of several psychiatric conditions, thus allowing an investigation that cuts through classic clinical boundaries.

## Results

**Task overview (Experiment 1)**. Participants were asked to imagine that their family/friends had rated them on different attributes (for example, 'intelligent', 'unreliable'). In block one, on each of the 40 trials, participants indicated whether they would like to know how others had rated them on a specific attribute using a six-point Likert scale from −3(definitely don't want to know) to +3(definitely want to know), with '0' not included (Supplementary Fig. 2). On average participants rated their desire to receive information as 0.43 (SD = 1.30), which is significantly different from the mid-point of the scale, $t(79) = 2.970$, $p = 0.004$.

In block two, participants provided the following ratings on a seven point Likert scale for each of the 40 traits: (i) their expectations regarding how useful it would be to know how others rated them on that trait (from −3 'not useful ' to +3 very useful), which provided an estimate of Instrumental Utility (e.g. how useful would it be to know how others rated you on 'intelligence'?); (ii) how they expect to feel if the rating was revealed to them (from −3 'very bad' to +3 'very good'; e.g. how will you feel if you knew how others rated you on 'intelligence'?) and how they expect to feel if the rating was never revealed to them (from −3 'very bad' to +3 'very good'; e.g. how will you feel

if you never knew how others rated you on 'intelligence'?). The difference between the last two ratings provided an estimate for Hedonic Utility and (iii) how often they think about each attribute (from −3 'never' to +3 'very often'; e.g. how often do you think about 'intelligence'?), which provided an estimate of Cognitive Utility. The questions were selected based on the theory paper[2] in which we had introduced the three utilities of information seeking and suggested quantifiable predictions. We note that these are not necessarily the only questions one can use to measure the three utilities, but we had proposed them as central ones in our original theory paper[2].

Additionally, we asked participants to indicate how they expected others would rate them (from −3 'not at all this trait' to +3 'very much this trait'). This was done for two reasons. First, our theory suggests that people's estimates of utilities are partially based on what they expect the information would reveal. For example, in order to estimate one's affective response to information one needs to predict what the content of the information would be. Second, this question then allowed us to ask participants about their confidence in the above rating (−3 'not certain' to +3 'very certain'). That is, how confident (certain) they are of what information would reveal. Many studies suggest that uncertainty is related to information seeking[33–38]. Sometimes people want information about things they are certain about (a form of conformation bias[36–38]) and sometimes they want information about things they are uncertain about[33–35], with one study suggesting that the sign of the effect can vary according to the environment[39]. Descriptive statistics of all these ratings and their inter-relationships are displayed in Supplementary Table 1.

**Information seeking is best explained by taking into account instrumental, hedonic and cognitive utilities (Experiment 1).** We tested 99 participants on the information-seeking task described above. Eighty participants passed the attention check and had enough variability in their rating data to generate three beta coefficients (that is did not insert the same rating for all stimuli on any of the scales). We submitted their data into a mixed-effects model to estimate the relationship between Instrumental Utility, Hedonic Utility and Cognitive Utility (which were estimated using the ratings as described above) and the desire to receive information (see methods). Each of these three factors were centered within participant for each rating across all trials and included in the model as fixed and random effects. Random intercept and slope were estimated for each participant as well as random intercept for each item (see methods). This revealed a significant fixed effect of Instrumental Utility ($\beta = 0.114 \pm 0.029$ (SE), $t(60.17) = 3.918$, $p = 0.001$, Fig. 1b), Hedonic Utility ($\beta = 0.123 \pm 0.022$ (SE), $t(61.28) = 5.531$, $p = 0.0001$, Fig. 1b) and Cognitive Utility ($\beta = 0.091 \pm 0.031$ (SE), $t(89.98) = 2.935$, $p = 0.004$, Fig. 1b). In particular, participants expressed a greater desire for knowledge when they believed the information would be useful, would have a more positive impact on their affect than ignorance, and also for stimuli they thought of frequently (see Supplementary Information for a study testing three additional motives of information seeking).

We tested thirteen additional models to test if any account for information-seeking choices better than the hypothesized model. These included models in which only a subset of the three utilities were entered and also models including how confident

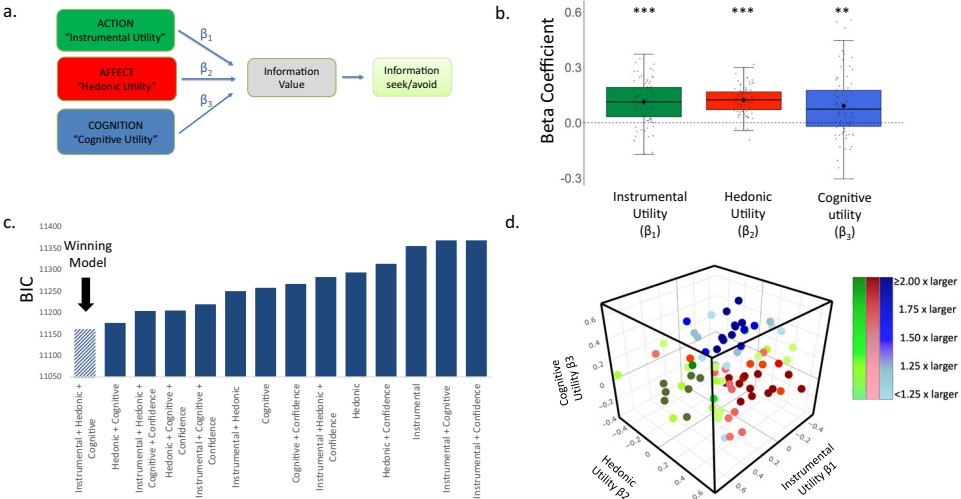

**Fig. 1 Information-seeking motives. a** Information seeking and its avoidance is hypothesized to be driven by Instrumental Utility, Hedonic Utility and Cognitive Utility[2]. These values reflect the predicted impact of information on action, affect and cognition, respectively. These estimates are hypothesized to be integrated into a computation of the value of information, with different weights ($\beta_{1–3}$) assigned to each of the three factors. The integrated value can lead to information seeking or avoidance. **b** Plotted are the beta coefficients from a linear mixed-effects model ($N = 80$ participants), showing that participants' desire to receive information was greater when the Instrumental Utility ($p = 0.001$, two sided), Hedonic Utility ($p = 0.0001$, two sided) and Cognitive Utility ($p = 0.004$, two sided) of information were higher. These were estimated respectively by participants' ratings of how useful the information would be, how they would feel to know vs not to know, and how frequently they think about the stimulus. The horizontal lines indicate median values, boxes indicate 25–75% interquartile range and whiskers indicate 1.5 × interquartile range; individual scores are shown as dots. **c** BIC scores reveal that the model described in **b** fit the data better than models including alternate combinations of the utilities and also those including participants' confidence regarding what the information would reveal. The same was true when examining AIC scores (see Supplementary Table 8). Smaller BIC and AIC scores indicate better fit. **d** Plotted are the weights each individual put on each motive when seeking information. Beta coefficients of Instrumental Utility are on the x-axis, of Cognitive Utility on the y-axis and of Hedonic Utility on the z-axis. Green dots represent participants who put the largest weight on Instrumental Utility when seeking information. Red dots represent participants who put the largest weight on Hedonic Utility when seeking information. Blue dots represent participants who put the largest weight on Cognitive Utility when seeking information. The colour gradient represents how dominant the largest weight was in comparison to the other two weights. Individuals who put more than twice as much weight on their dominant utility than the other two utilities are represented in darkest colours. Those whose dominant utility was less than 1.25 times larger than the other two are represented in the lightest colours. ***$P < 0.001$, **$P < 0.01$ (two sided). Source data are provided as a Source Data file.

participants were regarding the information to be revealed, which they also provided as a rating. The hypothesized model, which included instrumental, hedonic and cognitive utilities as predictors of information seeking, fit the data better than all other thirteen models. This is indicated both by a lower Bayesian Information Criterion (BIC) (Fig. 1c) and Akaike information criterion[40] (AIC) (Supplementary Table 8), both of which penalizes models for complexity[41].

While across our sample all three motives (action, affect, cognition) were strongly associated with information seeking, there may be significant individual differences in the importance participants assign to these when seeking information. To characterize such differences, we conducted for each participant separately a general linear model predicting information choice on each trial from the three utilities. As can be observe in Fig. 1d there were large individual differences in the weight participants assign to each motive. Most of the participants had a dominant motive; over one third of participants (34.75%) assigned more than twice the weight to one utility relative to the other two, and most participants (73.75%) assigned at least 1.25 times more weight to one utility than the other two. Different motives were dominant for different individuals, with action being dominant for 20% of individuals in this sample, affect for 27.5%, cognition for 26.25% and 26.25% did not have one particularly strong motive (that is no motive was assigned a weight at least 1.25 greater than the rest).

### Individual differences in the weights assigned to information-seeking motives provide a window into mental health (Experiment 1).

As described in the introduction, our hypothesis was that the different weights individual assigned to the different motives were related to mental health. Thus, we tested for a relationship between beta coefficients across individuals and mental health. We measured mental health using a dimensionality approach[30–32]. This approach considers the possibility that a specific psychopathology symptom is predictive of several conditions, allowing an investigation that cuts through classic clinical psychopathology boundaries. In particular, previous work[30–32] used a factor analysis across items in a large battery of traditional psychopathology questionnaires[21–29] and identified three-psychopathology dimensions[30,31] across those items: "Anxious-Depression", "Social-Withdrawal" and "Compulsive-Behaviour and Intrusive Thought". The factor analysis provided a weight to each item in relation to each dimension[30] (Fig. 2a). Thus, a person's symptom severity for each dimension can be quantified by having an individual complete a battery of traditional psychopathology questionnaires[21–29] and then calculating a weighted average across items' ratings. Indeed, this is what we did for each participant. First, we Z-scored the ratings of each questionnaire item separately across participants (not Z-scoring does not alter the significance of results). Then, for each participant we calculated the three-dimension scores which we submitted into a mixed ANOVA with psychopathology scores ("Anxious-Depression", "Social-Withdrawal", "Compulsive-Behaviour and Intrusive Thought") indicated as a within-subjects factor and the weight put on Instrumental Utility ($\beta_1$), Hedonic Utility ($\beta_2$), and Cognitive Utility ($\beta_3$) when seeking information all indicating within-subject modulating covariates. Participants' age and gender indicated between-subject modulating covariates. We observed a significant main effect of Cognitive Utility on psychopathology scores ($F_{(1,65)} = 6.061$, $p = 0.016$, partial eta square $= 0.085$).There were no significant effects of Instrumental Utility ($F_{(1,65)} = 2.882$, $p = 0.094$, partial eta square $= 0.042$) or Hedonic Utility ($F_{(1,65)} = 0.027$, $p = 0.870$, partial eta square $= 0.000$). No other effects or interactions were significant (all

p's > 0.188). These results suggest that the weight participants' assign to Cognitive Utility, but not the other two utilities, when seeking self-referntial information is related to their mental health across the three-psychopathology dimensions, with greater weight on Cognitive Utility associate with better mental health.

To illustrate this result in a more simplified manner, we conducted a linear regression with mental health as the dependent measure (quantified as the average psychopathology score across the three dimensions) and the following predictors: the weight assigned to Instrumental Utility ($\beta_1$) when seeking information, as well as that assigned to Hedonic Utility ($\beta_2$) and to Cognitive Utility ($\beta_3$). Age and gender were also included as predictors. Confirming the analysis above, a significant inverse relationship was observed between mental health and the weight assigned to Cognitive Utility when seeking information ($\beta = -1.053$, $p = 0.016$), suggesting that participants who seek information more on issues they think of often are the ones who report less psychopathology symptoms across the board. No other predictor was significant (Instrumental Utility: $\beta = -0.710$, $p = 0.094$; Hedonic Utility: $\beta = -0.072$, $p = 0.870$; Age: $\beta = -0.010$, $p = 0.893$; Gender: $\beta = -0.211$, $p = 0.296$; Fig. 2b). Finally, correlating each beta with the average psychopathology score across participants (controlling for age and gender), again reveals a significant association with the weight assigned to Cognitive Utility when seeking information ($r = -0.244$ (67) $p = 0.043$), but not with the weight assigned to Instrumental ($r = -0.136$ (67) $p = 0.264$) or Hedonic ($r = 0.09$ (67), $p = 0.463$) utilities.

### Stability of information-seeking motives over time (Experiment 2).

Thus far, we have shown that the weights individuals place on motives for information seeking are meaningful as they provide a window into mental health, which is known to be a function of both of 'trait' and 'state'. If information-seeking styles reflect mental health, they too may be a function of 'trait' and 'state'. One may thus predict that the weights assigned to information-seeking motives may show some stability over time, which also allows for changes due to factors such as altering mood, environment etc.

To quantify the stability of the motive weights of information seeking over time, we conducted a second longitudinal, study. This study also provided a replication test for the results obtained in Experiment 1. We tested 200 participants on the same information-seeking task as described above (Time 1), of which 176 participants passed attention checks and had enough variability in their rating data (that is did not insert the same rating for all stimuli on any of the scales) to generate three beta coefficients. Three weeks later we contacted these participants again, inviting them to participate in a follow up study (Time 2). One thirty seven completed the follow up study, on average 22 days following Time 1. Of these, 124 participants passed attention checks and had enough variability in their rating data to generate three beta coefficients. The task at Time 2 was identical to Time 1 except that we used a different list of attributes. This design allowed us to test how stable the relative importance of the three motives of information seeking were over time and stimuli sets. Descriptive statistics of ratings and their inter-relationships are displayed for Time 1 in Supplementary Table 2 and Time 2 in Supplementary Table 3.

Analysis was conducted as in Experiment 1. We observed a significant fixed effect of Instrumental Utility (Time 1: $\beta = 0.078 \pm 0.018$ (SE), $t(160.53) = 4.382$, $p = 0.0001$, Fig. 3a; Time 2: ($\beta = 0.086 \pm 0.020$ (SE), $t(87.56) = 4.267$, $p = 0.0001$, Fig. 3c), Hedonic Utility (Time 1: $\beta = 0.104 \pm 0.016$ (SE), $t(139.18) = 6.348$, $p = 0.0001$, Fig. 3a; Time 2: $\beta = 0.135 \pm 0.019$ (SE), $t(90.66) = 7.245$, $p = 0.0001$, Fig. 3c) and Cognitive Utility (Time 1: $\beta = 0.050 \pm 0.015$

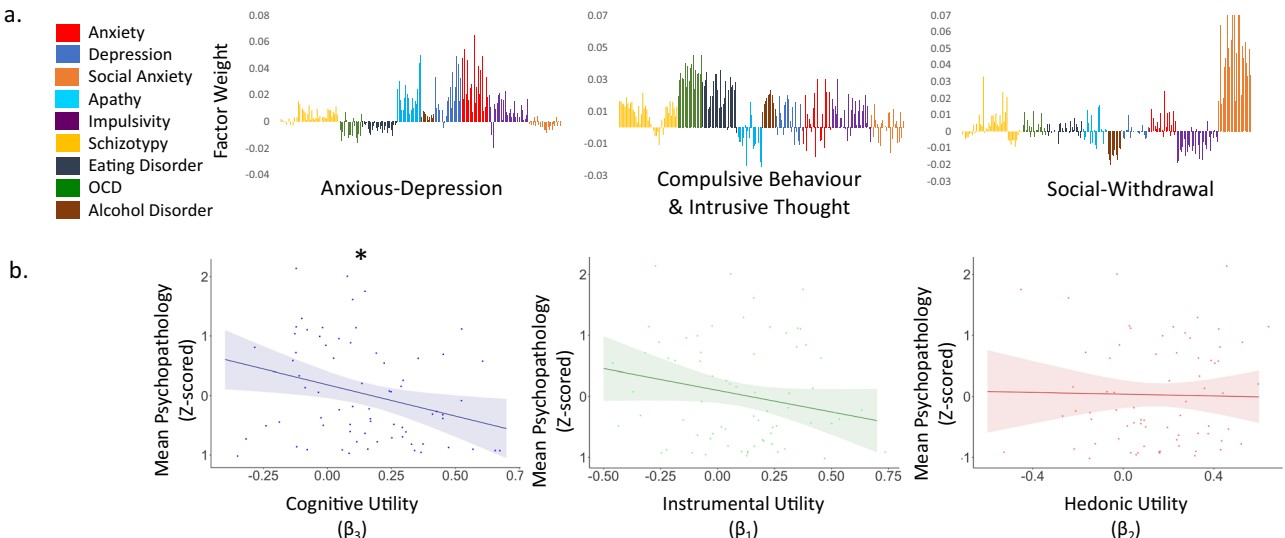

**Fig. 2 Information seeking related to psychopathology. a** Plotted are the weights (based on ref. [30]) given to each questionnaire item[21-29] when calculating the weighted score for each participant on each of the three psychopathology dimensions identified previously ("Anxious-Depression", "Social-Withdrawal" and "Compulsive-Behaviour and Intrusive Thought"). **b** Plotted on the y-axis is the average psychopathology score across the three dimensions described in **a**, Z-scored. On the x-axis are the weights assigned to each information-seeking motive from a linear regression predicting information seeking from Instrumental Utility (green), Hedonic Utility (red) and Cognitive Utility (blue). Dots represent individual participants. Shading represents confidence interval. Line represents the relationship between the abscissa and ordinate controlling for the effect of the other two motives as well as of age and gender. As can be observed, participants who placed a large positive weight on Cognitive Utility when seeking information reported less psychopathology symptoms ($p = 0.016$, two sided), while we observed no effect of Instrumental Utility ($p = 0.094$, two sided) or Hedonic Utility ($p = 0.870$, two sided). Error bars SEM. *$P < 0.05$ (two sided). $N = 71$ participants. Source data are provided as a Source Data file.

(SE), t(173.50) = 3.298, $p = 0.001$, Fig. 3a; Time 2: $\beta = 0.085 \pm 0.019$ (SE), t(124.76) = 4.500, $p = 0.0001$, Fig. 3c). As in Experiment 1, at Time 1 and Time 2 the model which included Instrumental, Hedonic and Cognitive Utilities as predictors of information seeking, fit the data better than comparison models according to the AIC score (see Supplementary Table 8). This was also true at Time 1 according to the BIC score (see Fig. 3b), while at Time 2 this model was second best, with a simpler model without Instrumental Utility receiving a lower BIC score. We suggest caution in interpreting this specific score as evidence against the importance for instrumental utility, as this conclusion will go against the AIC result, which penalizes less for complexity, as well as all other BIC results in all experiments described in this study (Exp 1, Exp 2 Time 1, Exp 3 Time 1, Exp 3 Time 2, Exp 4). Note, that at Time 1 one competing model (confidence + the three factors) did not converge.

Once again, most individuals had a dominant motive. 43.75% of individuals assigned more than twice the weight to one motive than the other two at Time 1 and 44.35% at Time 2, and 81.18% of individuals assigned at least 1.25 times more weight to one motive than the other two at Time 1 and 81.45% at Time 2. Different motives were dominant for different individuals, with action being dominant for 25.57% of individuals at Time 1 and 20.97% at Time 2, affect for 32.95% at Time 1 and 30.65% at Time 2, cognition for 23.30% at Time 1 and 29.84% at Time 2, and 18.18% at Time and 18.55% at Time 2 did not have one particularly strong motive (that is no motive was assigned a weight at least 1.25 greater than the rest).

We next tested to what extent the relative importance of the three information-seeking motives are stable over time within individuals. First, we measured by how much each participant moved over time within the three-dimensional space plotted in Fig. 3e, f. This indicates changes in the relative weights a participant assigned to the three betas. We then tested whether the magnitude of that change was significantly smaller than

chance. To test this, we reran the exact same analysis above for each participant, but each time mismatching one participant's T1 data with another participant's T2 data (i.e. permutation test). We then compared the average distance participants actually moved in the three-dimensional space from T1 to T2 to the average distance calculated from the permutation test. We did this 10,000 times and found that 100% of the time the average distance participants actually moved from T1 to T2 was smaller than chance (mean difference between iterations and actual mean movement = 0.103, range of differences = 0.04–0.157). Second, we calculated the Intraclass Correlation Coefficient (ICC) of each beta type across time (see methods). The ICC for each of beta type across time was significant (Instrumental Utility: ICC = 0.302, $p = 0.001$; Hedonic Utility: 0.543, $p = 0.001$; and Cognitive Utility: 0.560, $p = 0.001$).

**The relationship between information seeking and mental health is robust to replication (Experiment 2).** We next examined whether the three motives for information seeking were related to mental health in Experiment 2. To do so we calculated each participants' scores on the three psychopathology dimensions[30,31] as indicated in Experiment 1 and entered these into a mixed ANOVA with psychopathology dimension ("Anxious-Depression", "Social-Withdrawal", "Compulsive-Behaviour and Intrusive Thought") as a within-subjects factor and beta coefficients (averaged across time points) of Instrumental Utility ($\beta_1$), Hedonic Utility ($\beta_2$) and Cognitive Utility ($\beta_3$) as within-subject modulating covariates as well as participants' age and gender as between-subjects modulating covariates. Once again we observed a significant main effect of Cognitive Utility on psychopathology (F(1,117) = 4.471, $p = 0.037$, partial eta square = 0.037). There was no significant effect of Instrumental Utility (F(1,117) = 1.669, $p = 0.199$, partial eta square = 0.014) or Hedonic Utility (F(1,117) = 3.408, $p = 0.067$, partial eta

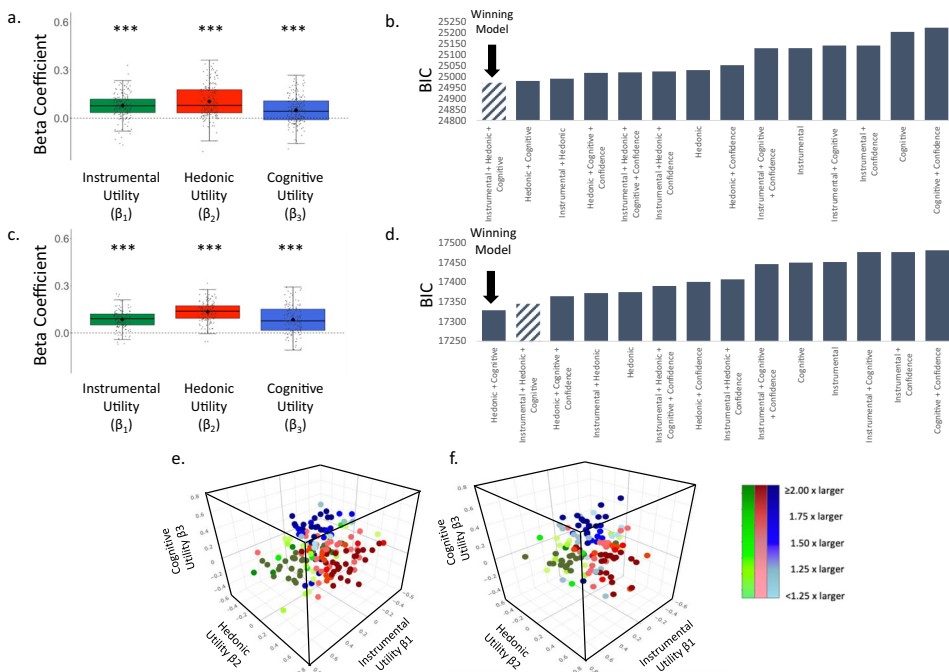

**Fig. 3 Information-seeking motives, Experiment 2. a, c** Plotted is a boxplot depicting the beta coefficients from a linear mixed-effects model at Time 1 ($N = 176$ participants) (**a**) and Time 2 ($N = 124$ participants) (**c**), which shows that participants' desire to receive information was greater when the Instrumental Utility (Time 1 $p = 0.0001$, Time 2 $p = 0.0001$; two sided), Hedonic Utility (Time 1 $p = 0.0001$, Time 2 $p = 0.0001$; two sided) and Cognitive Utility (Time 1 $p = 0.001$, Time 2 $p = 0.0001$; two sided) of information were higher. These were estimated, respectively, by participants' ratings of how useful the information would be, how they would feel to know vs not know, and how frequently they think about the stimulus. For each boxplot, the horizontal lines indicate median values, boxes indicate 25–75% interquartile range and whiskers indicate 1.5 × interquartile range; individual scores are shown separately as dots. **b** BIC scores from Time 1 reveal that the model described in **a** fit the data better than models including other combinations of the utilities and those including participants' confidence regarding what the information would reveal. **d** For Time 2 the model described in **c** fit the data second best according to the BIC model. AIC values (reported in Supplementary Table 8), however, indicate that the model described in **a**, **d** did fit the data best in comparison to control models for Time 1 and Time 2. Smaller BIC and AIC scores indicate better fit[40, 41]. **e**, **f** Plotted are the weights each individual put on each motive when seeking information at Time 1 (**e**) and Time 2 (**f**). Beta coefficients of Instrumental Utility are on the x-axis, of Cognitive Utility on the y-axis and of Hedonic Utility on the z-axis. Green dots represent participants who put the largest weight on Instrumental Utility when seeking information. Red dots represent participants who put the largest weight on Hedonic Utility when seeking information. Blue dots represent participants who put the largest weight on Cognitive Utility when seeking information. The colour gradient represents how dominant the largest weight was in comparison to the other two weights. Individuals who put more than twice as much weight on their dominant utility than the other two utilities are represented in darkest colours. Those whose dominant utility was less than 1.25 times larger than the other two are represented in the lightest colours. ***$P < 0.001$ (two sided). Source data are provided as a Source Data file.

square = 0.028). No other effects or interactions were significant (all p's > 0.265) except for gender, with females reporting more symptoms ($F(2,117) = 4.025$, $p = 0.02$, partial eta square = 0.064). These results suggest that the weight participants' assign to Cognitive Utility, but not the other two utilities, when seeking self-referential information is related to their mental health across the three psychopathology dimensions. As in Experiment 1, doing the analysis on raw numbers does not alter the significance of results.

To illustrate this result in a more simplified manner, we conducted a linear regression with mental health as the dependent measure (quantified as the average psychopathology score across the three dimensions) and the following predictors: the weight assigned to Instrumental Utility ($\beta_1$) when seeking information, as well as that assigned to Hedonic Utility ($\beta_2$) and to Cognitive Utility ($\beta_3$) (all averaged across the two time points). Age and gender were also included as predictors. Confirming the analysis above, a significant inverse relationship was observed between mental health and the weight assigned to Cognitive Utility when seeking information ($\beta = -0.790$, $p = 0.034$), suggesting that participants who seek information more on issues they think of often are the ones who report less psychopathology symptoms across the board. Gender was also significant with females scoring higher on psychopathology symptoms (Gender: $\beta = 0.345$, $p = 0.005$). No other factor was significant

(Instrumental Utility: $\beta = 0.498$, $p = 0.200$; Hedonic Utility: $\beta = 0.637$, $p = 0.063$; Age: $\beta = -0.010$, $p = 0.196$; Fig. 4). Finally, correlating each beta with the average psychopathology score across participants (controlling for participants' age and gender), again reveals a significant association with the weight assigned to cognitive utility when seeking information ($r = -0.241$ (120) $p = 0.008$), but not with the weight assigned to instrumental ($r = 0.114$ (120) $p = 0.212$) or hedonic (trend: $r = 0.175$ (120) $p = 0.053$) utilities.

**Across domains information seeking is best explained by taking into account instrumental, hedonic and cognitive utilities (Experiment 3).** We next asked whether the three motives identified in Experiments 1 and 2 are significantly related to information seeking in different domains. To that end we conducted a third study in which participants were asked whether they wanted financial information. As in Experiment 2, this study was longitudinal.

We tested 149 participants on a similar information-seeking task as described above in Experiment 1 and 2, however here we included 40 stimuli related to finance (e.g. *"Do you want to know what the unemployment rate is in Europe?", "Do you want to know the exchange rate between Dollar and Pound?"*). Once again, we included all participants who passed the attention check and had

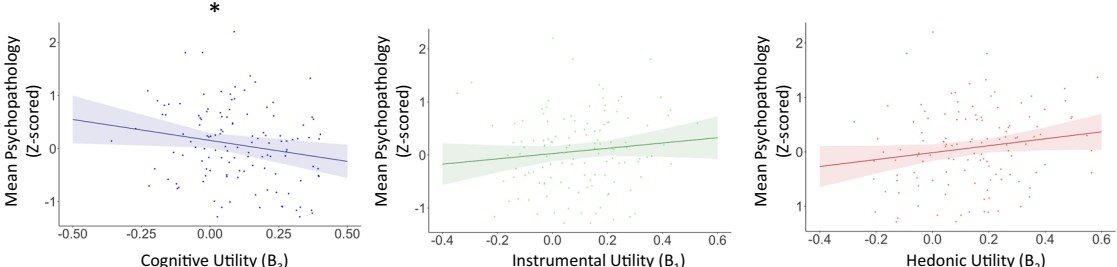

**Fig. 4 Association between information seeking and mental health is robust to replication.** Plotted on the $y$-axis is the average psychopathology scores across the three dimensions, Z-scored. On the $x$-axis are the weights assigned to each information-seeking motive from a linear regression predicting information seeking from Instrumental Utility (green), Hedonic Utility (red) and Cognitive Utility (blue), averaged across the two time points. Dots represent individual participants. Shading represents confidence interval. Line represents the relationship between the abscissa and ordinate controlling for the effect of the other two motives as well as for age and gender. As can be observed, participants who placed a large positive weight on Cognitive Utility when seeking information reported less psychopathology symptoms ($p = 0.034$, two sided), while we observed no effect of Instrumental Utility ($p = 0.200$, two sided) or Hedonic Utility ($p = 0.063$, two sided). Error bars SEM. *$P < 0.05$ (two sided). $N = 124$ participants. Source data are provided as a Source Data file.

enough data variability that allowed us to generate three beta coefficients (that is did not insert the same rating for all stimuli on any of the scales, Time 1 N = 122). Three weeks later, we invited these participants to participate in a follow up study (Time 2). Ninety-five participants completed the follow up study on average 23.43 days following Time 1. Two participants were not included due to providing a different Prolific ID at the two time points. Eighty-two participants passed the attention check and had enough data variability that allowed us to generate three beta coefficients. Descriptive statistics of ratings and their inter-relationships are displayed for Time 1 in Supplementary Table 4 and Time 2 in Supplementary Table 5.

The data were analyzed as in Experiment 1 and 2. We observed a significant fixed effect of Instrumental Utility (Time 1: $\beta = 0.266 \pm 0.022$ (SE), $t(109.56) = 12.223$, $p = 0.0001$, Fig. 5a; Time 2: $\beta = 0.279 \pm 0.029$ (SE), $t(78.98) = 9.497$, $p = 0.0001$, Fig. 5c), Hedonic Utility (Time 1: $\beta = 0.094 \pm 0.017$ (SE), $t(106.45) = 5.646$, $p = 0.0001$, Fig. 5a; Time 2: $\beta = 0.097 \pm 0.018$ (SE), $t(61.76) = 5.293$, $p = 0.0001$, Fig. 5c) and Cognitive Utility (Time 1: $\beta = 0.154 \pm 0.018$ (SE), $t(120.09) = 8.787$, $p = 0.0001$, Fig. 5a; Time 2: $\beta = 0.190 \pm 0.022$ (SE), $t(82.98) = 8.473$, $p = 0.0001$, Fig. 5c). Once more, the models which included Instrumental Utility, Hedonic Utility and Cognitive Utility as predictors of information seeking, fit the data better than models including only a subset of the three utilities and also of models including participants' confidence regarding the information to be revealed according to the BIC (see Fig. 5b, d) and the AIC (see Supplementary Table 8). Note, that at Time 1 two competing model (Hedonic + Instrumental and Hedonic + Cognitive) did not converge.

Once again, most individuals had a dominant motive. 44.26% of individuals assigned more than twice the weight to one motive than the other two motives at Time 1 and 50% at Time 2, and 80.32% of individuals assigned at least 1.25 times more weight to one motive than the other two at Time 1 and 80.49% at Time 2 (Fig. 5e, f). Different motives were dominant for different individuals, with action being dominant for 42.62% of individuals at Time 1, 46.34% at Time 2, affect for 18.03% at Time 1 and 10.98% at Time 1, cognition for 19.67% at Time 1, and 23.17% at Time 2, and 19.67% did not have one particularly strong motive at Time 1 and 19.51% at Time 2 (that is no motive was assigned a weight at least 1.25 greater than the rest).

We next tested to what extent the relative weight of the three information-seeking motives are stable over time within individuals. First, we measured by how much each participant moved over time within the three-dimensional space plotted in Fig. 5e, f.

This indicates changes in the relative weights a participant assigned to the three betas. We then tested whether the magnitude of change was significantly smaller than chance. To test this, we reran the exact same analysis above for each participant, but each time mismatching one participant's T1 data with another participant's T2 data (i.e. permutation test). We then compared the average distance participants actually moved in the three-dimensional space from T1 to T2 to the average distance calculated from the permutation test. We did this 10,000 times and found that 100% of the times the average distance participants actually moved from T1 to T2 was smaller than chance (mean difference between iterations and actual mean movement = 0.087, range = 0.015–0.15). Second, we calculated the Intraclass Correlation Coefficient (ICC) of each beta type across the time points (see methods). The ICC across time was significant for Instrumental Utility (ICC = 0.317, $p = 0.044$) and Hedonic (ICC = 0.329, $p = 0.039$) utilities, but not for Cognitive Utility (ICC = 0.019, $p = 0.446$), suggesting that the weights assigned to frequency of thought, while stable across time in the self-trait domain, is not in the finance domain. The weight assigned to expected affect and instrumental utility when seeking information show some stability across time in both the financial and self-referential domains.

**Stability of information-seeking motives across domains (Experiment 4).** Next, we wanted to know whether the three motives identified in Experiments 1–3 significantly predicted information seeking in a third domain, health, and whether these motives were stable within an individual across domains. To investigate this, we conducted a fourth study in which we invited 101 new participants as well as all participants who completed Experiment 3, Time 1 ($N = 122$) to complete another information-seeking task, but this time in the domain of Health. One-hundred and forty-eight participants completed the study, which included 47 participants from Experiment 3, Time 1 (which was conducted on average 166 days previous).

The task was similar to Experiment 1–3; however, here we included 40 stimuli related to health (e.g. *"Would you like to know if you have a gene that increases your likelihood of Alzheimer's disease?", "Would you like to know if you have a gene that increases your likelihood of a Strong Immune System?"*). Once again, data were analyzed for all participants who passed the attention check and who had enough data variability that allowed us to generate three beta coefficients (that is did not insert the same rating for all stimuli on any of the scales, $N = 116$). Descriptive statistics of ratings and their inter-relationships are displayed in Supplementary Table 6.

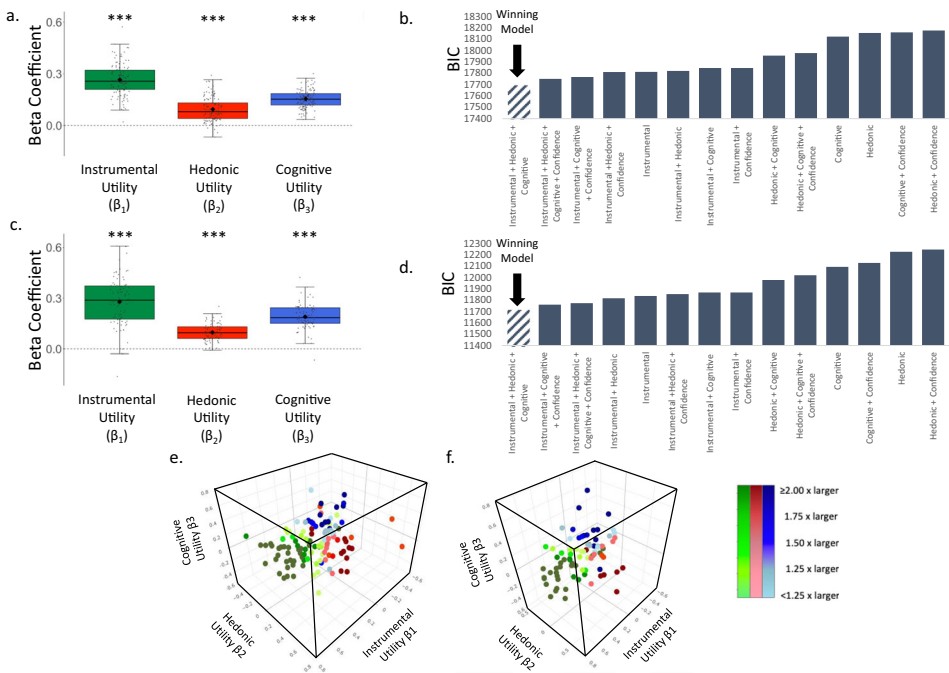

**Fig. 5 Information-seeking motives in the financial domain. a, c** Plotted is a boxplot depicting the beta coefficients from a linear mixed-effects model at Time 1 (N = 122 participants) (**a**) and Time 2 (N = 82 participants) (**c**), which shows that participants' desire to receive information was greater when the Instrumental Utility (Time 1 p = 0.0001, Time 2 p = 0.0001; two sided), Hedonic Utility (Time 1 p = 0.0001, Time 2 p = 0.0001; two sided) and Cognitive Utility (Time 1 p = 0.0001, Time 2 p = 0.0001; two sided) of information were higher. These were estimated respectively by participants' ratings of how useful the information would be, how they would feel to know vs not know, and how frequently they think about the stimulus. For each boxplot, the horizontal lines indicate median values, boxes indicate 25–75% interquartile range and whiskers indicate 1.5× interquartile range; individual scores are shown separately as dots. **b, d** BIC scores from Time 1 (**b**) and Time 2 (**d**) reveal that the model described in **a, c** fit the data better than models including other combinations of the utilities and those including participants' confidence regarding what the information would reveal. The same was true when examining AIC scores (see Supplementary Table 8). Smaller BIC and AIC scores indicate better fit[41]. **e, f** Plotted are the weights each individual put on each motive when seeking information at Time 1 (**e**) and Time 2 (**f**). Beta coefficients of Instrumental Utility are on the x-axis, of Cognitive Utility on the y-axis and of Hedonic Utility on the z-axis. Green dots represent participants who put the largest weight on Instrumental Utility when seeking information. Red dots represent participants who put the largest weight on Hedonic Utility when seeking information. Blue dots represent participants who put the largest weight on Cognitive Utility when seeking information. The colour gradient represents how dominant the largest weight was in comparison to the other two weights. Individuals who put more than twice as much weight on their dominant utility than the other two utilities are represented in darkest colours. Those whose dominant utility was less than 1.25 times larger than the other two are represented in the lightest colours. ***P < 0.001 (two sided). Source data are provided as a Source Data file.

The data were analyzed as in Experiment 1, 2 and 3. We observed a significant fixed effect of Instrumental Utility (β = 0.229 ± 0.026(SE), t(126.52) = 8.918, p = 0.0001, Fig. 6a), Hedonic Utility (β = 0.090 ± 0.020 (SE), t(103.74) = 4.447, p = 0.0001, Fig. 6a) and Cognitive Utility (β = 0.096 ± 0.015 (SE), t(128.60) = 6.295, p = 0.0001, Fig. 6a). Once more, the models which included Instrumental Utility, Hedonic Utility and Cognitive Utility as predictors of information seeking, fit the data better than models including only a subset of the three utilities and also of models including participants' confidence regarding the information to be revealed according to the BIC (see Fig. 6b) and the AIC (see Supplementary Table 8).

Once again, most individuals had a dominant motive. 52.59% of individuals assigned more than twice the weight to one motive than the other two motives, while 89% of individuals assigned at least 1.25 times more weight to one motive than the other two (Fig. 6c). Different motives were dominant for different individuals, with action being dominant for 57.76% of individuals, affect for 19.83%, cognition for 11.21% and 11.21% did not have one particularly strong motive (that is no motive was assigned a weight at least 1.25 greater than the rest).

We next tested to what extent the relative weight of the three information-seeking motives are stable across domain (i.e. Finance and Health) and time within individuals. Data was

analyzed for all those participants who completed Experiment 3, Time 1 and Experiment 4 and who passed the attention check and who had enough data variability that allowed us to generate three beta coefficients (N = 38). We first measured by how much they moved over domain/time within the three-dimensional space plotted in Figs. 5e, 6c. This indicated changes in the relative weights a participant assigned to the three betas. We then tested whether the magnitude of change was significantly smaller than chance. To test this, we reran the exact same analysis above for each participant, but each time mismatching one participant's Experiment 3, Time 1 data with another participant's Experiment 4's data (i.e. permutation test). We then compared the average distance participants actually moved in the three-dimensional space from Experiment 3, Time 1 to Experiment 4 to the average distance calculated from the permutation test. We did this 10,000 times and found that 99.73% of the times the average distance participants actually moved from Experiment 3, Time 1 to Experiment 4 was smaller than chance (mean difference between iterations and actual mean movement = 0.08, range = 0.02–0.17). Second, we calculated the Intraclass Correlation Coefficient (ICC) of each beta type across the time points (see methods). As in Experiment 3, the ICC for Instrumental Utility (ICC = 0.621, p = 0.002) and Hedonic Utility (0.445, p = 0.042) was significant, and the ICC for Cognitive Utility (0.272, p = 0.172) was not.

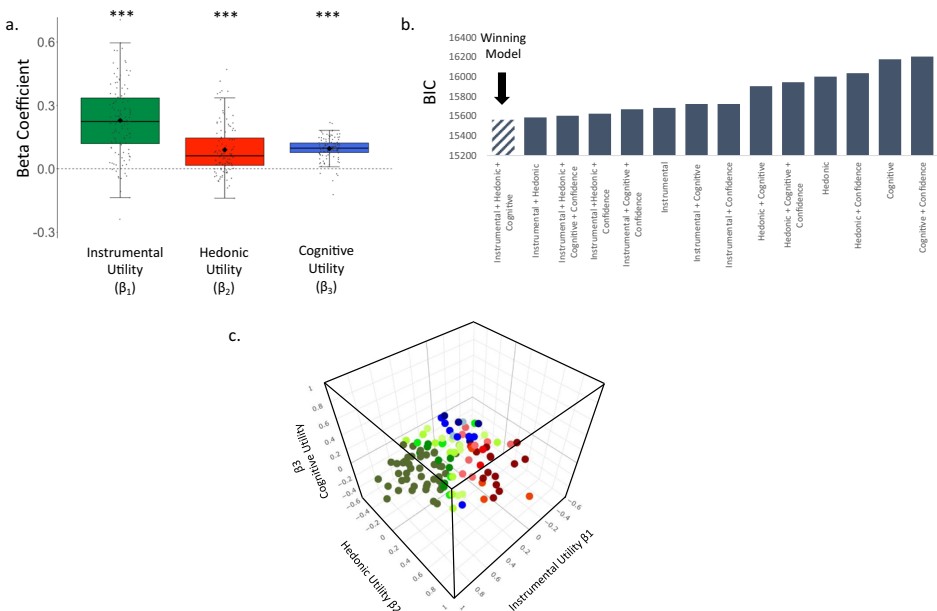

**Fig. 6 Information-seeking motives in the health domain. a** Plotted are the beta coefficients from a linear mixed-effects model (two sided; $N = 116$ participants), showing that participants' desire to receive health related information was greater when the Instrumental Utility ($p = 0.0001$, two sided), Hedonic Utility ($p = 0.0001$, two sided) and Cognitive Utility ($p = 0.0001$, two sided) of information were higher. These were estimated respectively by participants' ratings of how useful the information would be, how they would feel to know vs not to know, and how frequently they think about the stimulus. The horizontal lines indicate median values, boxes indicate 25–75% interquartile range and whiskers indicate 1.5 × interquartile range; individual scores are shown as dots. **b** BIC scores reveal that the model described in **a** fit the data better than models including alternate combinations of the utilities and also those including participants' confidence regarding what the information would reveal. The same was true when examining AIC scores (see Supplementary Table 8). Smaller BIC and AIC scores indicate better fit. **c** Plotted are the weights each individual put on each motive when seeking information in the health domain. Beta coefficients of Instrumental Utility are on the x-axis, of Cognitive Utility on the y-axis and of Hedonic Utility on the z-axis. Green dots represent participants who put the largest weight on Instrumental Utility when seeking information. Red dots represent participants who put the largest weight on Hedonic Utility when seeking information. Blue dots represent participants who put the largest weight on Cognitive Utility when seeking information. The colour gradient represents how dominant the largest weight was in comparison to the other two weights. Individuals who put more than twice as much weight on their dominant utility than the other two utilities are represented in darkest colours. Those whose dominant utility was less than 1.25 times larger than the other two are represented in the lightest colours. ***$P < 0.001$, **$P < 0.01$ (two sided). Source data are provided as a Source Data file.

Note, on average, the action motive was greater in health and finance domains than the self-trait domain. These findings together indicated that while 'trait' impacts the importance people assign to information-seeking motives other factors such as state and domain may matter too.

## Discussion

The desire for knowledge is a fundamental part of human nature[42]. People spend a substantial amount of time actively pursuing information, for example by asking questions, reading or conducting online searches. These activities, often referred to as 'information-seeking' behaviours, are integral to learning, social engagement and decision making[43–45].

Here we show that people want information more when they believe information (i) will be useful in guiding their actions, (ii) will have a positive impact on their affective state and (iii) is related to concepts they often think about. A model which incorporates these three motives, reflecting the influence participants expect information to have on their action, affect and cognition, explained individuals' information-seeking choices better than a range of other models. These results were replicated across four studies and three different domains—information about self-traits, finance and health—suggesting that the model is likely domain general.

We observed individual differences with regard to the weights participants assigned to the three motives when seeking information. Many participants assigned a particularly large weight to

one of the motives relative to the other two. That is, some participants were driven mostly to seek information according to its (i) predicted usefulness (action-driven), (ii) its predicted impact on their feelings (affect-driven), (iii) while others mostly sought information that relate to concepts they think of frequently (cognitive-driven). The individual differences in the weight people assign to the different motives when seeking information can help explain individual differences in what people want to know[2]. For example, a participant who assigns more weight to instrumental utility than hedonic utility may be more inclined to want to know if they have a predisposition to breast cancer than a participant who assigns more weight to hedonic utility than instrumental utility.

Our longitudinal studies indicate that these individual differences are fairly stable over time. Moreover, in study 4 (which included a smaller sample size than the other studies) we found that individuals who tended to assign a large weight to a motive in one domain (i.e. finance) relative to other individuals, tended to do so in another domain (e.g. health). We also saw interesting differences across domains, with the action motive being much greater in health and finance domains than the self-trait domain, and the cognitive motive being much more stable across time within the self-trait domain and not finance domain. Together, these findings suggest that the weights people assign to the different motives are likely a combined function of trait, state and domain.

The individual differences in the weights participants assigned to the three motives were related to mental health within the domain of self-traits. Specifically, those individuals who assigned a

larger weight to the cognitive motive when seeking information reported less psychopathology symptoms across the board. Our approach differs from the past few attempts to test for a relationship between information seeking and psychopathology[46–52] in two fundamental ways. First, rather than examining for an association between psychopathology and the frequency of information seeking (an approach which has led to mixed results[46–50]), we examined for an association between mental health and participants' motives for seeking information. Second, instead of using traditional psychopathology nosology, we adopted a dimensionality approach[30,31]. This approach considers the possibility that a specific symptom is predictive of several psychiatric conditions, allowing an investigation that cuts through classic clinical boundaries. Our results suggest that the relative importance of the information-seeking motives about the self are related to general mental health.

We have previously theorized that the relationship between mental health and information seeking is bidirectional[2]. Our study, however, is correlational and thus we cannot conclude whether certain patterns of information seeking lead to increase/decrease in psychopathology symptoms, and/or the other way around. Moreover, our findings suggest that the three motives measured here are associated with information seeking but cannot speak of causation. We also note that according to our theory[2] people first predict what information will likely reveal and based on that prediction estimate utilities. In some situations, expected information can be quantified and is highly correlated with a quantifiable estimated utility. For example, a person's expectations on how they will be rated on intelligence by others will be correlated with how they expect to feel when they receive that information (i.e. if they expect to be rated positively, they will probably feel good knowing the rating). In this specific case, a researcher could interchangeably use expected information or expected affect to predict information-seeking choice, because the former is simply the subjective assessment of the latter. In most cases, however, the two are not easily interchangeable. For example, if a person expects information will reveal the Dollar to Pound exchange rate is high that on its own does not tell us how they likely expect to feel about such information.

We have provided evidence that people's decisions about whether to seek or avoid information are related to an integration of the instrumental value, hedonic value and cognitive value of information. We further show that individual differences in information seeking reflect varying emphasis on these values, which in turn provides clues about participants mental health. These findings could be used to facilitate policy makers' ability to calculate the costs and benefits of information disclosure[53,54]. Moreover, by presenting information in a way that taps into the three motives of information seeking, policy makers may increase the likelihood that individuals will engage with and benefit from vital information.

## Methods

**Participants (Experiment 1)**. Ninety-nine participants completed the task on Mechanical Turk online system. Data of 3 participants who did not pass the attention checks were excluded from further analysis. In particular, participants were asked five times throughout the experiment to select a particular answer (for example: "*Please click answer two*"). This is to ensure that participants are being attentive. Participants who answered more than one of the attention checks incorrectly, were excluded from analysis. Of those who passed the check, 16 gave the same exact response on all trials in at least one of the utility ratings and thus their beta coefficients could not be calculated. Thus, data of 80 participants were analyzed (age = 37.69, SD = 9.18; females = 46.3%). One stimulus was repeated twice due to a coding error and thus data of the second repetition was removed from analysis leaving data from 39 trials per participant. Participants received £7.50 for their participation.

Note, for all experiments presented in this article, ethical approval has been provided by the Research Ethics Committee at University College London and all participants have given their informed consent to participate.

**Procedure (Experiment 1)**. Participants were asked to imagine that their family/friends had rated them on different attributes taken from ref. [55] (for example, 'intelligent', 'unreliable', see Materials for all attributes). In block one, on each of 40 trials participants indicated whether they would want to know how others rated them on a specific attribute using a six-point Likert scale from −3(definitely don't want to know) to +3(definitely want to know), with '0' not included (Supplementary Fig. 2). This was self-paced. Half the attributes were positive and half negative. Traits were presented in a random order.

In block two, to assess Instrumental, Hedonic and Cognitive Utility participants provided the following ratings for each attribute respectively (self-paced): (i) Their expectations regarding how useful each piece of information would be (from −3 '*not useful* ' to +3 *very useful*), which provided an estimate of Instrumental Utility (e.g. how useful would it be to know how others rated you on 'intelligence'?); (ii) How they expect to feel if the rating was revealed to them (from −3 '*very bad*' to +3 '*very good*') (e.g. how will you feel if you knew how others rated you on 'intelligence'?) and how they expect to feel if the rating was never revealed to them (from −3 '*very bad*' to +3 '*very good*') (e.g. how will you feel if you never knew how others rated you on 'intelligence'?). The difference between the last two ratings provided an estimate for Hedonic Utility; (iii) How often they think about each attribute (from −3 '*never* 'to +3 '*very often*') (e.g. how often do you think about 'intelligence'?), which provided an estimate of Cognitive Utility. This can be in relation to themselves, to others or to the concept itself. What we are measuring is how often the concept is thought of regardless of the exact context. The questions were selected based on the theory paper[2] in which we had introduced the three utilities of information seeking and suggested these quantifiable predictions. We note that these are not necessarily the only questions one can use to measure the three utilities, but we had proposed them as central ones[2]. Participants also indicated how they expected others would rate them (from −3 '*not at all this trait*' to +3 '*very much this trait*'; scores were reversed for negative valanced stimuli) and their confidence in this rating (−3 '*not certain*' to +3 '*very certain*'). The reason we asked about expectations is that it allowed us to then assess whether people were more likely to seek knowledge when they are confident or unconfident about what the information will reveal[3,9,10,34,35,56,57]. Each question was displayed separately for each attribute. Descriptive statistics of these ratings and their inter-relationships are displayed in Supplementary Table 1.

Next, participants completed self-report questionnaires which assess psychopathology symptoms[21–29] (the list is adapted from[30]) These included: Obsessive-Compulsive Inventory—Revised (OCI-R)[21], Self-Rating Depression Scale (SDS)[22], State-Trait Anxiety Inventory (STAI)[23], Alcohol Use Disorder Identification Test (AUDIT)[24], Apathy Evaluation Scale (AES)[25], Eating Attitudes Test (EAT-26)[26], Barratt Impulsivity Scale (BIS-11)[27], Short Scales for Measuring Schizotypy[28], Liebowitz Social Anxiety Scale (LSAS)[29]. Participants also indicate their age, gender, annual income and level of education. The task was coded using the Qualtrics online platform (https://www.qualtrics.com). Analysis was conducted using IBM SPSS 27 and R studio (Version 1.3.1056). All statistical tests conducted in the present article are two sided.

**Materials (Experiment 1)**. The following traits[55] were used: Courageous, Shy, Honest, Enthusiastic, Lazy, Mean, Trustworthy, Cooperative, Self-centered, Generous, Incompetent, Considerate, Rude, Conscientious, Boring, Easy-Going, Careless, Curious, Sophisticated, Unhelpful, Cowardly, Deceitful, Sociable, Confident, Unmotivated, Unfriendly, Unreliable, Organized, Greedy, Selfish, Polite, Disorganized, Imaginative, Adaptable, Ignorant, Competent, Immature, Helpful, Narrow-minded, Kind.

**Model testing (Experiment 1)**. We first tested the prediction that information-seeking choices across participants are best explained considering Instrumental Utility, Hedonic Utility and Cognitive Utility. To that end we ran a general linear mixed-effects model to assess the effect of the three utilities on information-seeking choice. The dependent variable was choice which is defined as the rating a participant gave to the question how much they wanted to know how others rated them on the respective trait. We quantified the scale such that one end ('definitely don't want to know') was given a −3 and the other ('definitely want to know') a + 3, '0' was not included. The three predictors were (i) Instrumental Utility (i.e. participants' rating of how useful it would be to receive that piece of information), (ii) Hedonic Utility (i.e. participants' rating of how they would feel if they received information minus how they would feel if they remain ignorant), (iii) Cognitive Utility (i.e. participants' rating of how often they think of the concept).

Each of these three factors was mean centered within participant and rating across all trials before entering in the model as fixed effects and random effects. Random intercepts and slopes were included for each participant as well as random intercepts for each item. This model (model 1) is the "hypothesized model". Six comparison models were tested which included only one or two utilities each. We compared the Bayesian Information Criterion[41] (BIC) and the Akaike Information Criterion[40] (AIC) scores of all seven models (the full hypothesized model and six comparison models) to test whether the full hypothesized model fits best. The BIC and AIC penalizes models for complexity[40,41]. We also attempted to include a random slope for each item[58–60], however, the theorized and comparison models frequently failed to converge across experiments. Thus, in line with recommendations[61,62], we reduced the item random effect structure (taking away

the item random slope) which successfully improved the convergence of models. Importantly, we did not observe any difference in the significance of the predictors between the model structures for the times that the model was able to converge.

Additional comparison models examined whether adding participants' confidence regarding what the information will reveal provided a better fit. In particular, we added a fourth factor to the full model: participants' rating of how confident they are of how others will rate them (again mean centered within participant). We compared the BIC and AIC scores of that model to the original hypothesized model, which only includes the three factors. We also tested models including subsets of those four factors that include the confidence rating (i.e. all models including only three factor or two factors, where one of the factors is the confidence rating and a model that includes only confidence ratings) to see whether any provide a better fit to the data than our hypothesized three-factor model. The winning model (i.e. model with lowest BIC and AIC score) was used for all the analyses below.

**Relating information-seeking types to mental health (Experiment 1).** Each participant was scored on the three-psychopathology dimensions identified by Gillan and colleagues[30] and replicated by Rouault and colleagues[31] "Anxious-Depression", "Social-Withdrawal" and "Compulsive-Behaviour and Intrusive Thought". To generate these scores, we first Z-scored the ratings for each questionnaire item separately across participants. Next, we multiplied each Z-scored item by its factor weight as identified earlier[30] (Fig. 2a). Then for each participant the three-psychopathology dimension scores were calculated by summing all of the weighted items assigned to each dimension. Nine participants did not compete all questionnaires and therefore were not included in the mental health analysis.

For each participant a general linear model was conducted predicting information choice on each trial from the three utilities. This generated three beta coefficients, indicating the weight each participant assigned to each motive when seeking information. These were then related to the psychopathology dimensions by submitting the three-psychopathology dimension scores into a mixed ANOVA with psychopathology dimension as a within-subject factor and Instrumental Utility ($\beta1$), Hedonic Utility ($\beta2$), and Cognitive Utility ($\beta3$) each as within-subject modulating covariates as well as participants' age and gender as between subjects modulating covariates. This analysis was then followed up with a simplified analysis in which the average of the three-psychopathology scores of each individual were entered as a dependent measure in a linear regression with each of the three beta coefficients (the weight put on Instrumental Utility ($\beta1$), Hedonic Utility ($\beta2$) and Cognitive Utility ($\beta3$)) entered as an independent measure as well as age and gender.

We report whether the three betas reflecting the weight each participant assigned to each information-seeking motive ($\beta1$, $\beta2$, $\beta3$) relate to demographics (age, gender and education), information-seeking choice, utility ratings, expected information and confidence in this estimation and scores on individual psychopathology questionnaires[21–29] by submitting each into a one-way ANOVA. All significant results were followed up with post-hoc pairwise comparisons. Psychopathology questionnaire scores were corrected for multiple comparisons across nine questionnaires using Bonferroni correction.

We also correlated each of the three-psychopathology dimension scores separately with: information-seeking choice, utility ratings, expected information and confidence in this estimation.

**Participants (Experiment 2).** 200 participants completed at Time 1 the same exact task as in Experiment 1 on Prolific's online platform. All participants who passed the attention check and for whom we could calculate all beta coefficients (i.e. those who did not give the same exact response on all trials in at least one of the utility ratings) ($N = 176$; age = 28.00, SD = 9.66; females = 47.2%) were then invited to complete the task again three weeks later (Time 2). Out of those, 137 participants completed the task at Time 2, of which 124 participants passed the attention check and did not give the same exact response on all trials in at least one of the utility ratings (age = 26.93, SD = 8.30; females = 46.0%). At Time 1, one random attribute was not presented to each participant due to a coding error, leaving 39 of the 40 attributes to be analyzed. At Time 2, participants saw 40 new attributes. Participants received £7.50 for their participation at Time 1 and £3.25 at Time 2.

**Procedure (Experiment 2).** At Time 1, participants were asked to complete the exact same procedure as Experiment 1, outlined previously. Three weeks later (Time 2), participants were asked to complete the same information-seeking task but with 40 different attributes[55] (see below). Descriptive statistics of all ratings and their inter-relationships are displayed in Supplementary Tables 2 and 3.

**Materials (Experiment 2).** In Time 1 we used the same traits[55] as in Experiment 1. In Time 2 we used the following traits[55]: Open-minded, Intelligent, Objective, Admirable, Calm, Loyal, Humble, Disciplined, Efficient, Fair, Stable, Warm, Wise, Impressive, Gracious, Patient, Popular, Creative, Ambitious, Dedicated, Cruel, Indecisive, Naïve, Disruptive, Reserved, Aggressive, Foolish, Cold, Difficult, Disloyal, Shallow, Messy, Thoughtless, Insensitive, Weak, Impulsive, Fearful, False, Dull, Arrogant.

**Analysis (Experiment 2).** We analyzed the data from Time 1 exactly as in Experiment 1. This allowed us to examine for replication of the results of Experiment 1 and provided us with the three beta coefficients (relating the three motives to information seeking) for each participant in Time 1.

Next, we examined whether the relative importance of the three information-seeking motives are stable over time within individuals. To do this, we first calculated for each participant the three beta coefficients (relating the three motives for information seeking) from Time 1 and Time 2 data separately. Then we measured by how much each participant moved over time with respect to each of their 3 motives, with each beta coefficient indicated on a separate axis in a three-dimensional space. $AB$ denotes the distance between participants at Time 1 and Time 2 in a 3-dimensional space, with each axis representing the weight they place on Instrumental Utility ($x$-axis), Hedonic Utility ($y$-axis) and Cognitive Utility ($z$-axis).

$$AB = \sqrt{(x_2 - x_1)^2 + (y_2 - y_1)^2 + (z_2 - z_1)^2} \qquad (1)$$

$x_2$ denotes participants' Instrumental Utility beta at Time 2, $x_1$ denotes its beta at Time 1, $y_2$ denotes participants' Hedonic Utility beta at Time 2, $y_1$ denotes the beta for Hedonic Utility at Time 1, $z_2$ denotes participants' Cognitive Utility beta at Time 2, while $z_1$ denotes the beta for Cognitive Utility at Time 1. If the relative weight individuals place on the motives for information seeking are stable over time, we would expect this change to be significantly less than what would be expected by chance. To test this, we reran the exact same analysis above for each participant, but each time randomly mismatching one participant's T1 data with another participant's T2 data (i.e. permutation test). We then compared the average distance participants actually moved in the three-dimensional space from T1 to T2 to the average distance calculated from the permutation test. We did this 10,000 times and calculated the percentage of the times the average distance participants actually moved from T1 to T2 was smaller than chance.

We also calculated an Intraclass Correlation Coefficient for each relative weight individuals placed on each of the three motives when seeking information across Time 1 and Time 2. To do this, we mean centered the three betas for each participant and time and then conducted a separate ICC test for each pair of equivalent betas.

When examining whether the motives for information seeking were related to mental health in Experiment 2, we implemented the same procedure as in Experiment 1, entering the average betas across the two time points into all analyses.

**Participants (Experiment 3).** One hundred forty-nine participants completed the experiment at Time 1 on Prolific's online platform. All participants who passed the attention check and for whom we could calculate all beta coefficients (i.e. those who did not give the same exact response on all trials in at least one of the utility ratings) ($N = 122$; mean age = 31.91, SD = 9.76 females = 46.7%) were invited to complete the task again three weeks later (Time 2). Out of those, 95 participants completed the task at Time 2. Two participants were not included due to providing different Prolific IDs for each time point. Eighty-two participants (mean age = 32.88, SD = 9.86; females = 52.4%) passed the attention check and did not give the same exact response on all trials in at least one of the utility ratings. Participants that passed the attention checks received £3.25 for their participation at Time 1 and for Time 2.

**Procedure (Experiment 3).** Participants were asked to imagine that we possessed a crystal ball that could reveal the answer to any question. In block one, on each of 40 trials they were asked whether they wanted to know specific information related to finance (e.g. what the exchange rate was between Dollar and Pound, what income percentile they fall into etc., see Supplementary Information for all stimuli). On each trial the stimulus were different and differed between Time 1 and Time 2. They indicated their response using a six-point Likert scale from −3 (definitely don't want to know) to +3 (definitely want to know), '0' was not included (Supplementary Fig. 2). This was self-paced.

In block two, participants provided the following ratings for each of the 40 traits: (i) their expectations regarding how useful it would be to know the information (from −3 'not useful ' to +3 'very useful'), which provided an estimate of Instrumental Utility (e.g. how useful would it be to know X'?); (ii) How they expect to feel if they knew the information (from −3 'very bad' to +3 'very good'; e.g. how will you feel if you knew X?) and how they expect to feel if they never knew the information (from −3 'very bad' to +3 'very good'; e.g. how will you feel if you never knew how X?). The difference between the last two ratings provided an estimate for Hedonic Utility; and (iii) how often they think about each topic (from −3 'never' to +3 'very often'; e.g. how often do you think about X'?), which provided an estimate of Cognitive Utility. Participants also indicated what they expected the information would be ("what do you think the answer is?"). Depending on the question asked, participants either answered on a scale (e.g. for the question about what the Gross Domestic Profit is, the scale went from "low" to "high") or input their answer into a text box (e.g. for the question about what your daily expenses are). Finally, participants indicated their confidence in what they expected the information would reveal (from −3 'not certain' to +3 'very certain').

We highlight a qualitative difference between the expectations scale in Experiment 1 and 2 and that in Experiment 3. In Experiments 1 and 2 participants indicated how they expected others would rate them (from *'not at all this trait'* to *'very much this trait'*). For the analysis scores were reversed for negative valanced stimuli (e.g. boring). Once expectations for negative valanced stimuli are reversed, this measure tell us how good or bad a participant expects information to be. For example, for "intelligence" a high rating will indicate a participant believed other saw him/her as possessing this trait (which is a good thing) for "boring" a low rating will indicate a participant believed other saw him/her as not possessing this trait (which is a good thing). Thus, one could use expectations in these experiments in a model where the motive for information is learning good news. However, in Experiment 3, expectations regarding financial information do not clearly reflect expectations of valence or feelings. If a participant expects the Dollar to Pound exchange rate to be high that does not tell us how they expect to feel if they learn it is high. In fact, there is no clear way to quantify expectations in the financial task nor would there be a consistent way to do so in other tasks like general knowledge questions (e.g. "Do you want to know if dogs are related to wolfs"). To build a model of motives of information seeking that can generalize to other domains any of the three utilities (+ confidence) would be possible to include, but not one that includes expectations of the information to be revealed. Descriptive statistics of all ratings and their inter-relationships are displayed in Supplementary Tables 4, 5.

**Analysis (Experiment 3)**. We carried out the exact analysis as described in Experiment 2 to examine whether the three motives are significant predictors of information seeking and whether the three-factor model is a better fit to the data than other models. We also describe individual differences in the same way and examine stability of weighting of information-seeking motives over time as done in Experiment 2.

**Participants (Experiment 4)**. We invited all participants who completed Experiment 3, Time 1 and an additional 101 new participants to take part in this study, which was run on Prolific's online platform. Data of the 116 participants who completed the study, passed the attention check and for whom we could calculate all beta coefficients (i.e. those who did not give the same exact response on all trials in at least one of the utility ratings) was analyzed (mean age = 31.15, SD = 11.30, females = 56.9%). Thirty-eight of these are participants who also completed Experiment 3, Time 1. Participants that passed the attention checks received £5.00 for their participation.

**Procedure (Experiment 4)**. Participants were asked to imagine that we had information about their genetic makeup. In block one, on each of 40 trials they were asked whether they wanted to know whether or not they carried a gene that increases their likelihood of a particular health condition or trait (e.g. *"Would you like to know if you have a gene that increases your likelihood of Alzheimer's disease?"*, *"Would you like to know if you have a gene that increases your likelihood of a Strong Immune System?"*, see Supplementary Information for all stimuli). On each trial the stimulus was different. They indicated their response using a six-point Likert scale from −3 (definitely don't want to know) to +3 (definitely want to know), not including '0' (Supplementary Fig. 2). This was self-paced.

In block two, participants provided the following ratings for each of the 40 health condition or traits: (i) their expectations regarding how useful it would be to know the information (from −3 *'not useful '* to +3 *very useful*), which provided an estimate of Instrumental Utility (e.g. how useful would it be to know X'?); (ii) How they expect to feel if they knew the information (from −3 *'very bad'* to +3 *'very good'*; e.g. how will you feel if you knew X?) and how they expect to feel if they never knew the information (from −3 *'very bad'* to +3 *'very good'*; e.g. how will you feel if you never knew X?). The difference between the last two ratings provided an estimate for Hedonic Utility and (iii) how often they think about each topic (from −3 *'never'* to +3 *'very often'*; e.g. how often do you think about X'?), which provided an estimate of Cognitive Utility. Participants also indicated their expectations of how likely it is that they carry the gene (from −3 *'not likely'* to +3 *'very likely'*, e.g. how likely is it that you carry this gene?; scores were reversed for negative valanced stimuli). Finally, participants indicated their confidence in what they expected the information would reveal (from −3 *'not certain'* to +3 *'very certain'*). Descriptive statistics of all ratings and their inter-relationships are displayed in Supplementary Table 6.

**Analysis (Experiment 4)**. We carried out the exact analysis as described in Experiment 1, 2 and 3, to examine whether the three motives are significant predictors of information seeking in the health domain and whether the three-factor model is a better fit to the data than other models. We also describe individual differences in the same way as Experiment 2 and 3; however, here we examine the stability of the weights given to the motives of information seeking across time and domain (i.e. finance Experiment 3, Time 1 and health Experiment 4).

**Reporting summary**. Further information on research design is available in the Nature Research Reporting Summary linked to this article.

## Data availability
Anonymized data and code are available at a dedicated Github repository [github.com/affective-brain-lab/Deciding_what_to_know_2020]. A reporting summary for this Article is available as a Supplementary Information file. Source data are provided with this paper.

## Code availability
Code supporting this study are available at a dedicated Github repository [github.com/affective-brain-lab/Deciding_what_to_know_2020].

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

## Acknowledgements

We thank Bastien Blain, Irene Cogliati Dezza, Liron Rozenkrantz, Moshe Glickman, Filip Gesiarz, Laura Globig, Valentina Vellani, Sarah Zheng, Gaia Molinaro and Christina Maher for comments on previous versions of the manuscript. Also, we thank Gloria Feng for help with data collection. The work was funded by a Wellcome Trust Senior Research Fellowship to T.S.

## Author contributions

C.K. and T.S. designed the study. C.K. collected and analyzed the data with guidance from T.S. C.K. and T.S. drafted the manuscript. All authors approved the final version of the manuscript for submission.

## Competing interests

The authors declare no competing interests.
