## [Peer Review File · Nature Communications]

Individual differences in information-seekingREVIEWER COMMENTS

Reviewer #1 (Remarks to the Author):

This paper examines potential factors that drive information seeking. In two online experiments, subjects rated to what extent they would like to find out how family and friends rated them on various attributes. For each attribute, subjects also rated how useful this information will be; how they expect to feel knowing or not knowing the information; and how often they think about the attribute. They also provided their estimate for the rating, and indicated how confident they were about that estimate. The authors identify three groups of subjects, each driven mostly by one of the factors. They also examine potential associations between individual differences in the weights of the three factors and individual differences in mental illness. The second experiment included two time points, which allowed for testing the stability of the behavioral profiles.

This is an interesting paper, employing a novel approach to information seeking. Examining the motivation for information seeking, rather than the degree of motivation seeking, in relation to psychopathology, is innovative. There are, however, some serious concerns about the design and methodology, as well as the interpretation of the results.

Major comments:

1. The choice of a social context for use throughout the experiments complicates the interpretation of the results. To provide ratings, participants need to imagine not only their family and friends' evaluation of themselves, but also to what extent these family and friends will be truthful in revealing their evaluations, how they will be affected by the participant's potential reaction to hearing the evaluations, etc. Also, different friends and family members may provide very different ratings, and have very different relationships with the participant – participants may want to hear some of those evaluations, but not others. In short, these questions might be too complicated and ambiguous for participants to answer. The question of measuring cognitive utility is also very ambiguous, as it does not specify whether participants need to think about each attribute of their family/friends/colleagues, of themselves, or generally of a random person. The authors should clarify why they chose a social context to frame the questions (beyond their brief mentioning of a link to mental health), and how this context might influence the interpretation of results.
2. Among the 40 attributes, some are positive (such as kind, helpful, ...) and others are negative (such as lazy, deceitful, ...). To measure hedonic utility, participants were asked how they would feel both knowing the rating, and not knowing the rating. However, the feeling would strongly depend on whether the attribute is regarded positive or negative. The authors should clarify why they did not separately analyze these attributes, or if they somehow transformed the ratings to tackle this problem.
3. A related point is that the rating estimates provided by the participants are not considered in the main analysis. Participants may be differently driven to seek information based on what they expect to hear. Moreover, here too it is not clear how attributes of positive and negative valence are treated.
4. By using K-means clustering in experiment 1, the authors identified three information-seeking types, with similar number of participants in each group. Would this result indicate equal number of people of these three information-seeking types in the population, or does the application of K-means clustering push the results to achieve similar cluster sizes? Looking at the 3-D scatter plots of beta coefficients in Figure 2b, the coefficients appear to be on a continuous scale, with no obvious effect clusters. This may result from the difficulty of visualization in a 3-D plot, and the authors could also plot three 2-D plots (beta1-beta2, beta1-beta3, beta2-beta3) to better visualize the data. Regardless, the authors should clarify the rationale of applying k-means clustering to identify discrete information-seeking types, instead of using a dimensional approach to look at the relationship between betas and psychopathology scores.
5. In Figure 2c, the authors should clarify what negative betas mean. For example, in the action-

dominant group, the betas of instrumental and cognitive utilities are positive, but the beta of hedonic utility is negative. Does it mean that hedonic utility works against instrumental and cognitive utilities, or is it an artifact from the k-means clustering?

6. The authors used a 3x3 mixed ANOVA to investigate the influence of psychopathology dimension and information-seeking type on the three psychopathology dimension factor scores. This ANOVA seems confusing. First, what does it mean to use psychopathology dimension as a factor, when the dependent variable is the psychopathology factor score? Second, what does it mean that there was no main effect of psychopathology dimension? The psychopathology factor scores were calculated from the z-scored questionnaire items, so their absolute magnitude are hard to interpret. For example, a factor score of zero does not imply “mentally healthy”, and a higher score in “Anxious-Depression” than “Social-Withdrawal” does not necessarily mean that the anxious-depression factor is more severe. The authors should clarify both the rationale for this ANOVA and the interpretation of the results.

7. In experiment 2, the authors indicate that information-seeking types were stable between time1 and time2, based on the significant correlation between coefficients (Figure 6). However, the correlations were not strong, considering that the two time points were only three weeks apart. The claim of stability of information-seeking types may be too strong. Similarly, k-means clustering shows that participants were categorized into the same information-seeking group 17.9% above chance (33%), which is about 50%. This chance of categorizing into the same group is also not high. From these results, readers could also conclude that information-seeking type may be state-dependent rather than stable, and could be related more to state, rather than trait, psychopathological measures. The authors should either provide stronger evidence to convince the readers that information-seeking types are stable, or discuss what potentially could influence its low stability.

8. In experiment 2, the authors compared the psychopathology dimensions among the three information-seeking types, only including participants that were categorized into the same group across time. The authors indicate that the relationship between information-seeking and mental health, observed in experiment 1, is replicated in experiment 2. However, comparing Figure 7 to Figure 3b, the results do not seem to be replicated visually. The action-dominant group showed more positive psychopathology scores in Figure 7, and it seems to drive the “stronger relationship” in experiment 2 described by the authors. This may be related to the Major point 6, that the absolute magnitude of z-scored factor scores is hard to interpret. The authors should find a clear way to quantify and visualize the results.

Minor comments:

1. In both experiments, the authors should presents more results about the desire to know how others had rated them and the estimation of these ratings, beyond the mean and std. In particular, they should present the distribution of these ratings, and their correlations with psychopathology dimension scores.

2. In Figure 2, the colors of the three information-seeking types are difficult to differentiate. The authors could use another set of color.

3. Figure 3a seems confusing, because the number of items (the length of the x axis) of the Social-Withdrawal factor is lower than that of the Anxious-Depression and Compulsive Behavior & Intrusive Thought. The authors should clarify this discrepancy or replot the loadings.

4. In Figure 1c, “confidence” appeared in the model comparison, but was not mentioned before in the text.

Reviewer #2 (Remarks to the Author):

In the present manuscript the authors investigate the motives underlying information seeking. Specifically, they had previously proposed that individuals assess the value of information for action, affect and cognition and that different weights on these values will determine whether information is sought or avoided. Here they test whether individuals exhibit stable patterns of prioritizing any of these motives over the other(s), resulting in information seeking types. Across two studies, the authors show that people group into three clusters consistent with the three motives and in the second study that these types are relatively stable over time. They further show that these information seeking types differentially relate to indices of mental health.

I like the theory behind this work very much. I think it brings to light an important aspect of human information processing and does justice to the often overlooked complexity of the human mind where one piece of information can have independent downstream effects on multiple dimensions of human behavior and experience.

With that said, I feel like the present work falls somewhat short of the promise of the theory and of providing a critical test of its predictions.

My mayor concerns relate to the measurement of the relevant constructs, the claims about stability of the types, and the insufficient demonstration that the types provide any benefit in explaining variance in mental health above and beyond participants individual weights on each motive.

I will unpack these concerns in turn below:

1) Measurement: Cognitive utility (defined as “does it help me better understand reality” in the introduction) is measured as concepts people often think about. I don’t understand why this measures the construct. I could have the opportunity to seek information about something I don’t often think about which would yet help me understand reality better – even more so because that is information that I don’t yet have access to. I might never have thought about how intelligent I am or how intelligent people think I am, but now that I could find out, I might realize how little I know about that and how this might change how I think about myself. How do the authors discriminate between this measure of cognitive utility and confirmation bias? Would information disconfirming my beliefs about something I often think about still have a high cognitive utility? What about novelty seeking? Are there different facets of cognitive utility of which this question really only captures one? I think it would help if the authors could explain why they think their question measures cognitive utility and chose to use it and if they concede that there might be other aspects of cognitive utility that this question does not capture, then I believe this warrants discussion.

2) In the introduction the authors write: “Here, we characterize and quantify the motives that drive information seeking and show how they explain individual-differences in information-seeking choices.” I believe they unfortunately do not quite do that. I think a critical test of the theory – maybe in a subsequent study - would be to also include other questions and show that the proposed motives provide superior explanatory power than the decoys. This was so far done with Confidence, which is indeed a critical one. I acknowledge that a model including Confidence had a larger BIC, however, Model comparison aside, did confidence have a reliable effect? What was the ranking according to AIC? Another approach would be to simply ask people why they would like to find out and do a factor analysis to see whether the three motives the authors propose fall out of the responses (again in a different study, not now and here). At this point all the authors show is that the questions they did ask correlate with behavior. They do not show that they’re the only ones correlating with behavior or that the questions actually measure the constructs they aim to measure. On that note: How much variance in information seeking was explained with the three constructs?

3) In the test for stability of the types: Please quantify the variance explained in the betas of the second rating by the betas of the first rating. While the correlations are significant, they are also not particularly high (maximum 0.36). I think it might be worth-while providing a more sensitive test for the stability of the types. From a psychometrics perspective, these correlations seem insufficient if they are meant to measure the same thing.

4) The obtained types also seem to be similar to the ones obtained in study 1 only insofar, as that in each type one of the three motives dominates. The remaining patterns appear to be very different. I

wonder what would happen if the authors identified clusters across the subjects of both studies (part 1 of study 2 only) and whether participants would remain in the respectively motive dominant group or whether in study 1 the clustering was to a stronger degree driven by the remaining motives (other than the dominant ones).

5) Along the same lines, while the classification was above chance, I don't think it warrants the conclusion of stable groups. Classification accuracy was 50.9%. This is far from a stable trait despite being above chance. How many participants were classified with confidence and included in the subsequent analyses? If the types are supposed to be useful in terms of psychiatry, a limitation in the use of the approach should be explicitly stated and evaluated.

6) What is the benefit of the typing when looking at mental health? Why would you want to reduce variance in your predictors if you need to collect that data anyway to even do the classification which does not seem overly robust? Do the information seeking types outperform the raw ratings in terms of variance explained? I.e. if you regress the individual psychopathology dimensions onto participants scores on each information seeking motive, does that explain more or less data than the group identity?

Reviewer #3 (Remarks to the Author):

The authors provide evidence for three motives that explain decisions to seek information about oneself, using hypothetical decisions from individuals about whether they would want to know how their friends and family rate them on different characteristics. A major strength of this work is the novel approach that the authors take to understand information-seeking behavior. Whereas prior work has focused on uncertainty reduction or curiosity state/trait factors as motives for information seeking, the authors put forward an intuitive model suggesting that individuals evaluate potential information along three dimensions. Furthermore, they also make a novel suggestion that there are stable individual differences in the weights that people assign to each dimension and that individuals can be grouped according to the dimension they weigh most. This work is a creative and intuitively appealing approach to understand decisions to seek information. However, there are significant limitations regarding the scope of information-seeking behavior examined, the scope of potential dimensions they evaluate that may contribute to information seeking, and the evidence supporting distinct groups of information seekers. These limitations are described below.

1. The manuscript describes the aims and results as about how people decide what they want to know, but the research examines a specific type of decision about a specific type of information (hypothetical decisions about how close others view oneself). There is no evidence given that the findings might be true for other examples of information seeking because the same primary question (would you like to know how your friends/family rated you?) is examined in both experiments. There is no argument given for why this particular example of information seeking would be representative of how people decide what they want to know (about information other than people's ratings about oneself). Even within the domain of ratings about oneself it is not obvious that a person has the same motives for seeking ratings from non-close others as they do for seeking ratings from family/friends.

2. Variables that were examined for their potential to explain information-seeking decisions include the three hypothesized motives plus one additional measure of confidence/certainty about the attribute. It is not clear how other variables were ruled out of consideration. Other variables that might have been considered for each attribute could be importance (how important is this attribute?), centrality (how central is this attribute to who you are?), distinctiveness (how much [do you differ]/[does the expected rating of you differ] from others on this attribute?), and valence. For the aim to understand how people decide what they want to know it would make sense to examine a broader field of (possibly overlapping) judgments.

3. An additional variable was collected (what rating do you expect?) but was not included as an explanatory variable in the model. This judgment appears to be related to affective utility but the decision to exclude it from the model was not explained. The affective utility questions themselves (how would you feel if you found out/never found out) may to some extent be based on the expected rating and attribute valence.

4. The rationale behind the measure of cognitive utility (how often do you think about this attribute?) is not given. That is, how does frequently thinking about an attribute make knowing your rating more helpful for comprehending reality? Could it not be helpful (for comprehending reality) to know about an attribute you don't think about often?
5. It would be helpful if the authors gave more information for readers to evaluate the separability of the three groups. The clusters are difficult to see in the 3D plots and there is no context to evaluate the average silhouette index between .2 and .3 (in Figs 2 and 3). The decision to use this cluster assignment as a predictor for the mental health symptom analysis was also unclear. As the clusters were based on 3 continuous variables, why not use the continuous variables (rather than convert them to a discrete group membership value)? Also the correlation across time for the motive regression coefficients ($r < .36$) seems low to justify labeling them as stable.
6. The number of participants excluded for the Experiment 2 mental health analysis is not directly reported but is about half of the reduced sample. By excluding participants without a stable group membership the remaining participants will be more extreme cases on the dominant motive (yielding a higher effect on mental health symptoms). The authors comment (p. 12) that the sample in Exp 1 is noisier for this reason but it could alternatively be stated that the Exp 2 sample is not representative.
7. The partial eta squared value for the main effect of information-seeking type in Experiment 1 is reported as .90 – this appears too high to be the correct value.
8. For the mixed effects model (information seeking explained by three motives) the authors should give some measure of variance explained by fixed effects.

Reviewer #4 (Remarks to the Author):

Review of Deciding what to know: Individual Differences in information-seeking

I'm a bit torn about this paper.

The paper addresses three major questions:

- (1) In an earlier paper the senior author proposed a tripartite division of motives for information seeking/avoidance. Is this a meaningful/the right division?
- (2) Is the propensity for one's information-seeking behavior to be driven by one motive or another a stable trait over time?
- (3) Is this propensity connected to mental health?

I think some of the findings are interesting, and would stimulate further research on information seeking and avoidance. However, I don't really think that the paper answers these questions that it sets out to address. Here are the reasons for this evaluation:

In both experiments, the respondent imagines that family members/friends rate the participant on attributes; half positive, half negative. This seems to me to be an awfully narrow task to then claim generality to information seeking in general. Information about one's own traits is only one variety of information, from a very wide range of different types of information. I would be much more convinced if, e.g., in the second phase of the second study, it wasn't virtually the same task. It is a good thing, however, that the second phase did involve a non-overlapping set of personal descriptors.

I was curious about why, in the regressions, confidence was included by not the expected ratings of others. The correlation matrices show that this is much more strongly correlated with other measures of interest than confidence. I would absolutely want to see regressions including this variable (both dividing out expected evaluations on positive and negative items, and also recoding the negative items so that all the expectations were in the same direction)(hence, two additional regressors in one set of regressions and one in another).

Two of the dimensions (action and understanding) seem unidimensional, but affect can definitely lead to information avoidance. The fact that mean information seeking was .44 on a scale ranging from -3

to +3 suggests that there was probably a lot of active information avoidance, but this is not discussed in the paper, which is all about when people seek information.

Let me now turn to the three major questions in the paper.

(1) In an earlier paper the senior author proposed a tripartite division of motives for information seeking/avoidance. Is this a meaningful/the right division?

If there are meaningful dimensions, I'm not convinced that the paper adequately addresses the question of whether these are the right dimensions. To be persuaded on this point, I'd like to see these motives pitted against other motives. What might such other motives be? A few that come instantly to mind would be "To answer questions they have," "to help them make sense of things that have happened in their lives," "To give them a better understanding of themselves." I would be far more persuaded if questions were asked that got at other possible motives, but the three postulated by the researchers were the ones that emerged as key.

I'm a bit skeptical of the cluster analysis. You gave it three variables, and three clusters emerged, one for each variable. I would want to know what patterns in the data would have given rise to a different set of clusters.

I find the third motive and the way it is measured to be the most dubious/amorphous. Participants are asked, for example, "How OFTEN do you think about kindness, and some who report that they think about it often also say they want to know how kind other people find them to be. I don't see how this is a measure of the degree to which "the information [will] improve, or impair my ability to comprehend reality."

(2) Is the propensity for one's information-seeking behavior to be driven by one motive or another a stable trait over time?

I'm the reviewer who asked for the additional data. I was provided with this table from the authors (which I reproduce for the benefit of the other reviewers and the editor):

T2: action affect cognitive

T1 (below)

action 9 15 8

affect 8 24 12

cognitive 9 8 30

The second and third classifications seem reasonably stable, but the first is not at all. Clearly, less than 1/3 of people classified in the first cluster at time 1 are classified in the same cluster at time 2. I'm also, again, concerned that this is all about desire for information about one's own traits, not desire for information in general.

(3) Is this propensity connected to mental health?

I'm especially skeptical on the paper's answer to this question. It does seem that there are some correlations, but they don't make much sense to me. Why, especially in the second study, would the desire to seek information for instrumental reasons (because it is useful) be quite a bit more associated with psychopathology than the desire to seek (or avoid) it for affective reasons or to achieve better understanding. I would have thought that affective reasons would be the one that would predict psychopathology.

But, in any case, it isn't at all clear how the causality runs. If I'm a neurotic person, then maybe I'm worried that others will view me negatively on all these traits, which would lead me to avoid the information. I don't really see the point of examining these connections.

Thank you for considering our manuscript "Deciding What to Know: Individual differences in information-seeking" for publication in *Nature Communications*. We were encouraged by the assessment of the reviewers, who noted that "This is an interesting paper, employing a novel approach to information seeking. Examining the motivation for information seeking, rather than the degree of information seeking, in relation to psychopathology, is innovative." (Reviewer 1); That the theory behind the paper "brings to light an important aspect of human information processing and does justice to the often overlooked complexity of the human mind" (Reviewer 2); as well as that "A major strength of this work is the novel approach that the authors take to understand information-seeking behaviour... This work is a creative and intuitively appealing approach to understand decisions to seek information" (Reviewer 3); and that "the findings are interesting and would stimulate further research on information seeking and avoidance" (Reviewer 4).

The Reviewers also raised conceptual and technical issues that needed to be addressed. We were able to address all of these issues in the revised manuscript. First, we conducted a new longitudinal study that demonstrates that our information-seeking model generalizes to a different domain. In particular, the domain of finance (**Experiment 3**). Second, we conducted an additional study that demonstrates that the information-seeking motives we had put forward explain information-seeking choices better than a range of new motives/questions (**Experiment 4, Supplementary Material**). Third, in accordance with the reviewers' suggestion, instead of clustering individuals according to the weight they assign the motives, we now examine individual differences and relate them to mental health by looking at the full scale of the weights. This new approach does not alter the original findings. We also address the rest of the reviewer's comments including justifying the scale used to measure Cognitive Utility.

We would like to thank the reviewers for a thorough and thoughtful commentary on our work. Incorporating their suggestions has strengthened the manuscript which we hope is now suitable for publication in *Nature Communication*. To ensure we address all of the Reviewers' comments, and for ease of reference, we include the reviews below (in bold) followed by our response to each concern.

Reviewer #1:

This paper examines potential factors that drive information seeking. In two online experiments, subjects rated to what extent they would like to find out how family and friends rated them on various attributes. For each attribute, subjects also rated how useful this information will be; how they expect to feel knowing or not knowing the information; and how often they think about the attribute. They also provided their estimate for the rating, and indicated how confident they were about that estimate. The authors identify three groups of subjects, each driven mostly by one of the factors. They also examine potential associations between individual differences in the weights of the three factors and individual differences in mental illness. The second experiment included two time points, which allowed for testing the stability of the behavioral profiles.

This is an interesting paper, employing a novel approach to information seeking. Examining the motivation for information seeking, rather than the degree of motivation seeking, in relation to psychopathology, is innovative. There are, however, some serious concerns about the design and methodology, as well as the interpretation of the results.

Major comments:

1. The choice of a social context for use throughout the experiments complicates the interpretation of the results.

- Following the Reviewer's comment, we conducted an additional longitudinal study examining information-seeking in a different domain. In particular, we asked participants if they wanted information regarding 40 different stimuli related to *finance* (e.g., if they wanted to know what is the exchange rate of dollar to pound, what is the unemployment rate etc.) Once again, they rated how useful the information would be, how they would feel if they knew and if they remained ignorance, and how often they thought of this issue. The new results support our previous conclusions. Once again, we observe that Instrumental Utility (Time 1: $\beta = 0.613 \pm 0.042$ (SE), $t(127) = 14.707$, $p = 0.0001$, **Figure 5a**; Time 2: $\beta = 0.682 \pm 0.056$ (SE), $t(90) = 12.212$, $p = 0.0001$, **Figure 5c**), Hedonic Utility (Time 1: $\beta = 0.247 \pm 0.034$ (SE), $t(109) = 7.267$, $p = 0.0001$, **Figure 5a**; Time 2: $\beta = 0.257 \pm 0.041$ (SE), $t(89) = 6.208$, $p = 0.0001$, **Figure**

5c) and Cognitive Utility (Time 1: $\beta = 0.285 \pm 0.032$ (SE), $t(107) = 9.015$, $p = 0.0001$, **Figure 5a**; Time 2: $\beta = 0.396 \pm 0.042$ (SE), $t(80) = 9.497$, $p = 0.0001$, **Figure 5c**) explain information-seeking choice (**pg. 14**). A model that includes these three factors fit the data better than alternative models. Moreover, we find fair stability across time regarding the weights participants assign to the different motives (Interclass Correlation Coefficient = 0.479, $p = 0.0001$). We would like to thank the reviewer for prompting us to examine the generality of our model. The new study suggests that the results are not tied to a specific domain or a specific wording of the stimuli and thus makes the conclusions stronger.

To provide ratings, participants need to imagine not only their family and friends' evaluation of themselves, but also to what extent these family and friends will be truthful in revealing their evaluations, how they will be affected by the participant's potential reaction to hearing the evaluations, etc. Also, different friends and family members may provide very different ratings, and have very different relationships with the participant – participants may want to hear some of those evaluations, but not others. In short, these questions might be too complicated and ambiguous for participants to answer.

- As mentioned above, we now add a study that looks at information seeking in a different domain - finance. We ask questions such as 'do you want to know the exchange rate of dollar to pound?' 'the unemployment rate?' etc. This new study supports the original conclusions as detailed above. As the study does not require participants to imagine receiving evaluation etc. it addresses the concerns raised by the reviewer.
- We would like to note, however, that our theory rests precisely on the notion that "the first stage in deciding whether to seek information involves solving a *prediction problem*. People must predict the likely content of information and its influence on action, affect and cognition." (Sharot & Sunstein, 2020). Indeed, to solve this problem, people need to imagine (consciously or unconsciously) the content of the information, which may well be ambiguous, as well as their reaction to it. As the reviewer says – this is difficult. Indeed, "Prediction problems can be extremely hard to solve and are notoriously vulnerable to biases. First, people have problems in estimating probability and thus may commit systematic errors in predicting the likely content of information. Second, there are potential gaps between the utility expected at the time of deciding whether to seek information and the utility experienced during consumption of information." (Sharot & Sunstein, 2020). The task mirrors real-life information-seeking problems. Importantly, despite the problem being complex and at time ambiguous (as it is in real life), subjects do a good job solving it. If they did not, we would not be able to detect meaningful relationships between the different ratings and information-seeking choices.

The question of measuring Cognitive Utility is also very ambiguous, as it does not specify whether participants need to think about each attribute of their family/friends/colleagues, of themselves, or generally of a random person.

- The question participants need to answer is how often they think about the trait (e.g., "how often do you think about intelligence?"). This can be in relation to themselves, to others or to the concept itself. What we care about is how often the concept is 'activated' regardless of the context (that is regardless of whether a subject thinks of intelligence regarding themselves, others or as an abstract concept). We now clarify this in the methods (**pg 17**).

The authors should clarify why they chose a social context to frame the questions (beyond their brief mentioning of a link to mental health), and how this context might influence the interpretation of results.

- We indeed selected the social context because we hypothesized that information-seeking in this domain will be related to psychopathology. However, as mentioned above following the reviewer's comments, we now ran a study that looks at information seeking in a different domain (finance). This study provides evidences for generality of the importance of the three motives. We thank the reviewer for prompting us to do so as the new study greatly strengthens the core argument.

2. Among the 40 attributes, some are positive (such as kind, helpful, ...) and others are negative (such as lazy, deceitful, ...). To measure Hedonic Utility, participants were asked how they would feel both knowing the rating, and not knowing the rating. However, the feeling would strongly depend on whether the attribute is regarded positive or negative. The authors should clarify why they did not separately analyze these attributes, or if they somehow transformed the ratings to tackle this problem.

- Indeed, feelings in Experiment 1 & 2 are tied to whether attributes are positive or negative as one would expect them to be. However, there is no need to transform hedonic ratings because high scores always indicated a positive feeling for knowing and low scores always indicated a negative feeling for knowing regardless of whether the trait is positive or negative. For example, a subject may expect people to rate them high on intelligence and thus have a positive feeling for knowing (so they click +3 for the feeling rating). They also expect others to rate them low on boring, which also will lead to a positive feeling of knowing (so they also click +3). They could also expect to feel neutral in either case (and click 0) etc.
- The rationale for including both positive and negative stimuli in the study was to obtain a broad range of feeling ratings, which allows us to capture their impact. Dividing all stimuli into two and then running two models would be suboptimal, as there would be a loss of power because (i) trials would be cut in half and (ii) the range of feelings would be significantly reduced. At the same time such an analysis would not enable us to answer any additional questions. Note, that in the new study (i.e. finance domain), most stimuli cannot be considered positive or negative per-se. Yet, the model still describes information-seeking well.

3. A related point is that the rating estimates provided by the participants are not considered in the main analysis. Participants may be differently driven to seek information based on what they expect to hear. Moreover, here too it is not clear how attributes of positive and negative valence are treated.

- The reviewer is suggesting that subjects will be driven to seek information based on what they expect to hear. Indeed, this is exactly what our theory suggests (Sharot & Sunstein, 2020). The theory is that subjects consciously or unconsciously predict what information will likely reveal. This expectation then triggers an estimated affective response (Sharot & Sunstein, 2020). It can also inform the other two utilities. We now clarify the theory in the introduction of the revised paper (pg 3).
- As the reviewer suggests, transformation is required and was indeed performed for expectation ratings (see pg 17). In particular, we reversed the expectations scores for negative stimuli, such that for all stimuli high numbers will indicate positive expectations (and thus likely trigger positive feelings) and low numbers will indicate negative expectations (and thus likely trigger negative feelings). For instance, for “intelligence” a high rating indicates a subject believed other saw him/her as possessing this trait (which is a good thing). While for “boring” scores were reversed (that is multiplied by -1), such that after reversal a high rating indicating a subject believed other did not see him/her as possessing this trait (which is a good thing).
- As expectations are correlated with Hedonic Utility in Exp 1 & 2 (Exp 1: mean $r(79) = 0.38$, $p = 0.0001$; Exp 2, Time1: Mean $r(175) = 0.32$, $p = 0.0001$; Exp 2, Time 2: $r(123) = 0.31$) one could in theory swap one for the other in the model. However, this is not possible for the new Experiment 3, as expectations regarding financial information do not clearly reflect feelings. If I expect the Dollar to Pound exchange rate to be high that does not tell you how I expect to feel if I learn it is high. This will likely depend if I earn in Dollars, Pounds or neither. The reason we select to measure feelings as we did, is that this measure can be generalized to any information-seeking task (even to a question like “Do you want to know if dogs are related to wolfs”) while expectations cannot. Moreover, it directly tests a specific motive, while expectation is not a direct measure of motives per-se. We now detail this rationale in the revised manuscript (pg 20).

4. By using K-means clustering in experiment 1, the authors identified three information-seeking types, with similar number of participants in each group. Would this result indicate equal number of people of these three information-seeking types in the population, or does the application of K-means clustering push the results to achieve similar cluster sizes?

- The former is true – the result indicates approximately equal number of people in the three information-seeking types in the population. The application of K-means clustering does not push the results to achieve similar cluster sizes. Note, that following the suggestions of several of the reviewers we now use a distributional analysis rather than a clustering approach.

Looking at the 3-D scatter plots of beta coefficients in Figure 2b, the coefficients appear to be on a continuous scale, with no obvious effect clusters. This may result from the difficulty of visualization in a 3-D plot, and the authors could also plot three 2-D plots (beta1-beta2, beta1-beta3, beta2-beta3) to better visualize the data. Regardless, the authors should clarify the rationale of applying k-means clustering to identify discrete information-seeking types, instead of using a dimensional approach to look at the relationship between betas and psychopathology scores.

- Following the reviewer’s comment, we now use the continuous variables for our analysis rather than a clustering approach. Using these continuous variables instead of the clusters leads to the same original conclusions. Specifically, we find large individual differences in the weights participants assign to the different motives (pg 5 & 9). We also find that those differences are related to mental health, with subjects who put more weight on Cognitive Utility exhibiting less psychopathology symptoms (Experiment 1: $F(1,65) = 6.061$, $p = 0.016$, partial eta square = 0.085; Experiment 2: $F(1,117) = 4.471$, $p = 0.037$, partial eta square = 0.037) (pg 7 & 11).

5. In Figure 2c, the authors should clarify what negative betas mean. For example, in the action-dominant group, the betas of instrumental and cognitive utilities are positive, but the beta of Hedonic Utility is negative. Does it mean that Hedonic Utility works against instrumental and cognitive utilities, or is it an artifact from the k-means clustering?

- As mentioned above we no longer use K clustering in our analysis, but use the continuous scales. Thus, the aforementioned Fig 2c is not in the revised manuscript. The mean group betas are all significantly positive in all four experiments. Some subjects may have negative betas. Individuals with a negative Hedonic utility beta tend to select knowledge when they expect it would make them feel worse than ignorance. Individuals with a negative Cognitive Utility beta tend to select knowledge for items they think of rarely. Individuals with a negative Instrumental Utility beta tend to select knowledge when it is not useful.

6. The authors used a 3x3 mixed ANOVA to investigate the influence of psychopathology dimension and information-seeking type on the three psychopathology dimension factor scores. This ANOVA seems confusing. First, what does it mean to use psychopathology dimension as a factor, when the dependent variable is the psychopathology factor score? Second, what does it mean that there was no main effect of psychopathology dimension? The psychopathology factor scores were calculated from the z-scored questionnaire items, so their absolute magnitude are hard to interpret. For example, a factor score of zero does not imply “mentally healthy”, and a higher score in “Anxious-Depression” than “Social-Withdrawal” does not necessarily mean that the anxious-depression factor is more severe. The authors should clarify both the rationale for this ANOVA and the interpretation of the results.

- Given that we no longer use clustering we no longer use a 3x3 mixed ANOVA. To answer the question, however, you may think of this as a Math test with three sub sections (i.e. “dimensions”): geometry, algebra and logic questions. There are three groups of subjects from three different schools. The question is whether Math scores is a function of school (so do students do better in one school than the other), a function of math “dimension” (that is are students in general better in one type of math, such as geometry or algebra across schools) or an interaction (that is, in school A students are better in algebra than the rest of the schools but in school B they are better in geometry than the rest of the schools). You enter the three math scores into a 3 (school) by 3 (math dimension) ANOVA and find a main effect of school. This means that one school (let’s say school A) is better than all other schools across the board regardless of math sub-section (“dimension”).
- In our case the analysis was meaningful only for detecting the effects of interest of the paper, which were a main effect of group and a group by psychopathology dimension interaction. Indeed, reporting “main effect of psychopathology dimension” is meaningless in our case and of no interest to the research question.
- In Z-scoring the data we followed the original studies using this specific factor analysis approach (Gillan et al., 2016; Seow et al., 2020). The rationale is that the different items are rated using different scales. Without Z-scoring, certain scales will contribute more/less to the overall score simply because they happen to be on a specific scale range. Nevertheless, conducting the analysis on raw scores reveals the same results; subjects who put more weight on Cognitive Utility exhibited less psychopathology symptoms (Experiment 1: $F(1,65) = 5.893$, $p = 0.018$, partial eta square = 0.083; Experiment 2: $F(1,117) = 4.117$, $p = 0.045$, partial eta square = 0.034). We now mention this in the text (pg 7 & 11).

7. In experiment 2, the authors indicate that information-seeking types were stable between time1 and time2, based on the significant correlation between coefficients (Figure 6). However, the correlations were not strong, considering that the two time points were only three weeks apart. The claim of stability of information-seeking types may be too strong. Similarly, k-means clustering shows that participants were categorized into the same information-seeking group 17.9% above chance (33%), which is about 50%. This chance of categorizing into the same group is also not high. From these results, readers could also conclude

that information-seeking type may be state-dependent rather than stable, and could be related more to state, rather than trait, psychopathological measures. The authors should either provide stronger evidence to convince the readers that information-seeking types are stable, or discuss what potentially could influence its low stability.

- Following Reviewers' 2 suggestion (below) we now use a more sensitive measure to test for stability. Specifically, we calculated the commonly used ICC measure. We find what is considered in the literature "fair" stability (Cicchetti and Sparrow, 1981) for the weights participants put on the three motives in Experiment 2 (ICC = 0.456, $p=0.0001$) and replicate these results in the new Experiment 3 (ICC = 0.479, $p=0.0001$). This indicates that both "trait" and "state" likely matter. Varying factors, such as mood and stress, for example, may also alter information-seeking motives. We now report the new analysis in the revised manuscript and provide the above interpretation (**pg 11 & 15**). We thank the reviewer for this comment.

8. In experiment 2, the authors compared the psychopathology dimensions among the three information-seeking types, only including participants that were categorized into the same group across time.

- Thank you for this comment. We now include all participants in all analysis, showing the same results (**pg 11**).

The authors indicate that the relationship between information-seeking and mental health, observed in experiment 1, is replicated in experiment 2. However, comparing Figure 7 to Figure 3b, the results do not seem to be replicated visually. The action-dominant group showed more positive psychopathology scores in Figure 7, and it seems to drive the "stronger relationship" in experiment 2 described by the authors. This may be related to the Major point 6, that the absolute magnitude of z-scored factor scores is hard to interpret. The authors should find a clear way to quantify and visualize the results.

- We thank the reviewer for prompting us to find a clearer way to quantify and visualize our results. We now conduct two complementary analyses to examine the relationship between mental health and the weights participants assign to each motive. In one analysis, which we portray in **Fig 2** and **Fig 4**, we conducted a linear regression predicting the mean psychopathology scores across the three dimensions from the weights participants assign to the three utilities, controlling for age and gender. We find a negative relationship between psychopathology scores and the weights participants assign on Cognitive Utility (Exp 1: $\beta = -1.053$, $p = 0.016$; Exp 2: $\beta = -0.790$, $p = 0.035$). With none of the other utilities reliably revealing significant findings in both studies (Experiment 1: Instrumental Utility: $\beta = -0.710$, $p = 0.094$; Hedonic Utility: $\beta = -0.072$, $p = 0.870$; Experiment 2: Instrumental Utility: $\beta = 0.498$, $p = 0.200$; Hedonic Utility: $\beta = 0.637$, $p = 0.063$). We also conduct a similar, but more complex, analysis using an ANOVA which shows the same results (a significant effect only of Cognitive Utility on psychopathology scores Experiment 1: $F(1,65) = 6.061$, $p = 0.016$, partial eta square = 0.085; Experiment 2: $F(1,117) = 4.471$, $p = 0.037$, partial eta square = 0.037).
- The main result is that subjects who put more weight on Cognitive Utility report less psychopathology symptoms. This result replicates across Experiments. The relationship between mental health and Instrumental Utility and Hedonic Utility are not significant in either Experiment. It is thus not surprising that the numerical tendencies of these non-significant effects do not replicate. It is only the significant effect (which replicates) that we discuss in detail in the paper.
- As explained above Z-scoring is necessary as the questionnaire items are recorded on scales of different ranges and thus cannot be averaged and compared without Z-scoring. We note, however, that the graphs and results are the same when the analysis is done on raw scores.

Minor comments:

1. In both experiments, the authors should present more results about the desire to know how others had rated them and the estimation of these ratings, beyond the mean and std. In particular, they should present the distribution of these ratings, and their correlations with psychopathology dimension scores.

- As requested by the reviewer we now present the distribution of the information-seeking choice and the expectation rating as well as their correlation with the psychopathology scores (**Supplementary Material**).

2. In Figure 2, the colors of the three information-seeking types are difficult to differentiate. The authors could use another set of color.

- The colours have now been altered.

3. Figure 3a seems confusing, because the number of items (the length of the x axis) of the Social-Withdrawal factor is lower than that of the Anxious-Depression and Compulsive Behavior & Intrusive Thought. The authors should clarify this discrepancy or replot the loadings.

Thank you for this comment. Fig 2a has now been amended to show the factor weights of all items for each factor, which are equal.

4. In Figure 1c, “confidence” appeared in the model comparison, but was not mentioned before in the text.

- Thank you for this comment. We now mention confidence in the text (pg 4).

Reviewer #2 (Remarks to the Author):

In the present manuscript the authors investigate the motives underlying information seeking. Specifically, they had previously proposed that individuals assess the value of information for action, affect and cognition and that different weights on these values will determine whether information is sought or avoided. Here they test whether individuals exhibit stable patterns of prioritizing any of these motives over the other(s), resulting in information seeking types. Across two studies, the authors show that people group into three clusters consistent with the three motives and in the second study that these types are relatively stable over time. They further show that these information seeking types differentially relate to indices of mental health.

I like the theory behind this work very much. I think it brings to light an important aspect of human information processing and does justice to the often overlooked complexity of the human mind where one piece of information can have independent downstream effects on multiple dimensions of human behavior and experience.

With that said, I feel like the present work falls somewhat short of the promise of the theory and of providing a critical test of its predictions.

My mayor concerns relate to the measurement of the relevant constructs, the claims about stability of the types, and the insufficient demonstration that the types provide any benefit in explaining variance in mental health above and beyond participants individual weights on each motive.

I will unpack these concerns in turn below:

1) Measurement: Cognitive Utility (defined as “does it help me better understand reality” in the introduction) is measured as concepts people often think about. I don’t understand why this measures the construct. I could have the opportunity to seek information about something I don’t often think about which would yet help me understand reality better – even more so because that is information that I don’t yet have access to. I might never have thought about how intelligent I am or how intelligent people think I am, but now that I could find out, I might realize how little I know about that and how this might change how I think about myself.

- Following the reviewer’s comment, it became clear that in the previous manuscript we did not accurately explain the concept of Cognitive Utility or why the question we use is relevant. In the original paper describing the theory (Sharot & Sunstein, 2020) we had made the clear prediction that “people will be more likely to want information relating to concepts that are frequently activated ... This information will have greater positive cognitive value.” (Sharot & Sunstein, 2020). By “activate” we meant “thought about”, which is the prediction we tested here. As we explain in the original paper (Sharot & Sunstein, 2020), the reason we had put forward this precise prediction is that concepts people think about often are central to their internal representations of their unique world (e.g., their ‘mental models’). One reason they are frequently thought off is that they are associates with a large number of other concepts. Thus,

information about such concepts will be perceived as especially important for strengthening their mental models - the essence of 'cognitive utility'. For example, all else being equal, a person who thinks about dogs frequently would be more interested in learning whether dogs are related to wolfs compared to someone who rarely thinks about dogs. The suggestion is that *on average* information related to highly activated concepts will be subjectively perceived as relevant to people's mental models and thus sought out. Now, whether information is related to frequently activated concepts is not the only factor that determines Cognitive Utility (see answer to question below), however it is one factor that matters. We selected this specific factor because of the clear prediction put forward in the original paper. We now clarify and expand the definition of Cognitive Utility and explain how the question used in this study relates to it (pg 3, 17).

How do the authors discriminate between this measure of Cognitive Utility and confirmation bias? Would information disconfirming my beliefs about something I often think about still have a high Cognitive Utility? What about novelty seeking?

- The original theory paper (Sharot & Sunstein, 2020) indeed speaks directly to the relationship between Cognitive Utility and confirmation bias:

“It has been suggested that people strive to minimize the difference between their mental models and external reality. This can be achieved in two ways. The first is to improve existing models by seeking out information that will generate new connections among concepts, or that will strengthen connections that are suspected but of which people are uncertain. This approach will improve the fit of the model to reality by refining it, which will in turn increase people's ability to comprehend and anticipate reality. The second approach, less intuitive, is to avoid information that people suspect will weaken strong existing inter-connected ties within the model (e.g., disconfirming information). In this approach, people maintain a fit between internal representation and external reality not by actively changing the model but by actively changing the reality of which they are aware. Such avoidance can improve the *sense* of comprehension at present, because disconfirming information can cause confusion and a need to rebuild large parts of the model. Thus, disconfirming information may be assigned negative 'cognitive value', despite the fact that avoiding such information could create less accurate mental models.” (Sharot & Sunstein, 2020).

In our data we find that people want information more when they are more confident of what information will reveal (which may indicate confirmation bias), however this effect is significant only in a subset of the experiments (Experiment 1: mean $r(79) = 0.05$, $p = 0.061$; Experiment 2, Time 1: mean $r(175) = 0.07$, $p = 0.0001$, Experiment 2, Time 2 mean $r(123) = 0.02$, $p = 0.246$, Experiment 3, Time 1: mean $r(121) = 0.11$, $p = 0.0001$, Experiment 3, Time 2: mean $r(82) = 0.14$, $p = 0.0001$). One may interpret this as the opposite of novelty seeking, albeit unreliably. Moreover, adding this factor to our model does not improve the fit (Figures 1, 3 & 5) and reveals again that confidence is an unreliable predictor (Exp1: Confidence $\beta = -0.008 \pm 0.039$ (SE), $t(50) = -0.199$, $p = 0.843$; Experiment 2, Time 1: Confidence $\beta = 0.063 \pm 0.022$ (SE), $t(107) = 2.871$, $p = 0.005$; Experiment 2, Time 2: Confidence $\beta = 0.002 \pm 0.029$ (SE), $t(103) = 0.076$, $p = 0.939$; Experiment 3, Time 1: Confidence $\beta = 0.042 \pm 0.026$ (SE), $t(78) = 1.632$, $p = 0.107$; Experiment 3, Time 2: Confidence $\beta = 0.018 \pm 0.030$ (SE), $t(69) = 0.599$, $p = 0.551$), while the other three factors are significant. Thus, we conclude frequency of thought drives participants' choices more than confirmation when the two compete for variance.

Are there different facets of Cognitive Utility of which this question really only captures one? I think it would help if the authors could explain why they think their question measures Cognitive Utility and chose to use it and if they concede that there might be other aspects of Cognitive Utility that this question does not capture, then I believe this warrants discussion.

- Yes, there are different facets of Cognitive Utility of which this question captures one. Albeit, a central one. Whether information is related to frequently activated concepts is not the only factor that determines Cognitive Utility (see answers above). For example, there may be concepts that are associated with central concepts in a mental model which in themselves are not activated frequently. People may indeed want information about those concepts. However, a central prediction of the original theory is that concepts that are activated more frequently (that is thought of often) on average will have high Cognitive Utility. We now clarify and expand the definition of Cognitive Utility and explain how the question use in this study relates to it (see answer to Reviewer's comments above and pg 3-4).

2) In the introduction the authors write: “Here, we characterize and quantify the motives that drive information seeking and show how they explain individual-differences in information-seeking choices.” I believe they unfortunately do not quite do that. I think a critical test of the theory – maybe in a subsequent study - would be to also include other questions and show that the proposed motives provide superior explanatory power than the decoys. This was so far done with Confidence, which is indeed a critical one. I acknowledge that a model including Confidence had a larger BIC, however, Model comparison aside, did confidence have a reliable effect? What was the ranking according to AIC?

- The ranking according to AIC was the same as BIC in Experiment 1, Experiment 2 Time 2, Experiment 3 Time 1 and Experiment 3 Time 2. The only exception was Experiment 2, Time 1 in which the model with confidence had the lowest AIC. We now add both AIC and BIC in the revised manuscript (pg 6, 10, 14 & 29). The relationship between confidence and information seeking was the following for each experiment: Experiment 1: mean $r(79) = 0.05$, $p = 0.061$; Experiment 2, Time 1: mean $r(175) = 0.07$, $p = 0.0001$, Experiment 2, Time 2 mean $r(123) = 0.02$, $p = 0.246$, Experiment 3, Time 1: mean $r(121) = 0.11$, $p = 0.0001$, Experiment 3, Time 2: mean $r(82) = 0.14$, $p = 0.0001$.
- Following the reviewers comment we ran an additional study with three extra questions (these questions were suggested by some of the reviewers). The questions were distinctiveness (how much do you differ from others on ‘kindness?’), sense making (if you knew how others rated you on being ‘kind’, would it help you make sense of things that happened in your life?), as well as recency (before today, when was the last time you thought of whether others view you as ‘kind?’). We found that while the original questions were significantly related to information seeking (Instrumental Utility $\beta = 0.144 \pm 0.063$ (SE), $t(43) = 2.262$, $p = 0.029$, Hedonic Utility $\beta = 0.220 \pm 0.071$ (SE), $t(33) = 3.123$, $p = 0.004$, and Cognitive Utility $\beta = 0.198 \pm 0.055$ (SE), $t(43) = 3.632$, $p = 0.001$), the additional questions were not (Confidence $\beta = 0.091 \pm 0.071$ (SE), $t(37) = 1.275$, $p = 0.210$; Distinctiveness $\beta = 0.041 \pm 0.044$ (SE), $t(1713) = 0.928$, $p = 0.353$, Sense-making $\beta = -0.004 \pm 0.076$ (SE), $t(37) = -0.053$, $p = 0.958$, and Recency $\beta = 0.156 \pm 0.082$ (SE), $t(42) = 1.900$, $p = 0.058$). We thank the reviewer for prompting us to run this additional check which we now report in the revised manuscript (pg 26).

Another approach would be to simply ask people why they would like to find out and do a factor analysis to see whether the three motives the authors propose fall out of the responses (again in a different study, not now and here).

- We thank the reviewer for this suggestion of a future study. We hypothesize that some or all of the motives driving information seeking are unconscious. It is thus possible that participants are not aware of the reasons they want information and thus will have a difficult time explicitly stating them. However, this is an empirical question that could be studied in the future.

At this point all the authors show is that the questions they did ask correlate with behavior. They do not show that they’re the only ones correlating with behavior or that the questions actually measure the constructs they aim to measure. On that note: How much variance in information seeking was explained with the three constructs?

- As stated above we now ran a new study with three additional questions (suggested by the other reviewers) and find that the three original questions are significantly related to information seeking while the additional questions were not. The specific questions used were the ones theorized in the original theory paper as questions that will measure the theorised constructs (Sharot & Sunstein, 2020). However, we are certainly not suggesting they are the only ones that will correlate with information-seeking.
- The variance explained was 0.458 in Experiment 1, 0.363 in Experiment 2 time 1, 0.287 in Experiment 2 time 2, 0.383 in Experiment 3 time 1, 0.484 in Experiment 3 time 2, 0.386 in study 4. These numbers have now been added to the manuscript.

3) In the test for stability of the types: Please quantify the variance explained in the betas of the second rating by the betas of the first rating. While the correlations are significant, they are also not particularly high (maximum 0.36). I think it might be worth-while providing a more sensitive test for the stability of the types. From a psychometrics perspective, these correlations seem insufficient if they are meant to measure the same thing.

- We thank the reviewer for this helpful suggestion. We now use a more sensitive measure to test for stability. Specifically, we calculated the commonly used interclass correlation coefficient (ICC) measure (Cicchetti and Sparrow, 1981). We find what is considered in the literature “fair stability” for the weight participants put on the motives in Experiment 2 (ICC = 0.456, $p = 0.0001$) and then replicate this finding in Experiment 3 using financial stimuli (ICC = 0.479, $p = 0.0001$). This suggests that both trait and state likely matter for information-seeking. See **pg 15** for discussion.

4) The obtained types also seem to be similar to the ones obtained in study 1 only insofar, as that in each type one of the three motives dominates. The remaining patterns appear to be very different. I wonder what would happen if the authors identified clusters across the subjects of both studies (part 1 of study 2 only) and whether participants would remain in the respectively motive dominant group or whether in study 1 the clustering was to a stronger degree driven by the remaining motives (other than the dominant ones).

- Following the reviewers’ comment 6 below we now use the continuous variable rather than clustering.

5) Along the same lines, while the classification was above chance, I don’t think it warrants the conclusion of stable groups. Classification accuracy was 50.9%. This is far from a stable trait despite being above chance. How many participants were classified with confidence and included in the subsequent analyses? If the types are supposed to be useful in terms of psychiatry, a limitation in the use of the approach should be explicitly stated and evaluated.

- Following the reviewers’ comment below (Comment 6) we now use the continuous variable rather than clustering. As such all participants are included in subsequent analysis and the stability, quantified using the interclass correlation coefficient (ICC) measure, is found to be what is considered in the literature (Cicchetti and Sparrow, 1981) as fair (Experiment 2: ICC = 0.456, $p = 0.0001$). We now also provide a replication of this result in the domain of finance (Experiment 3: ICC = 0.479, $p = 0.0001$). The fair stability suggests a contribution both for trait and state. See **pg 11 & 15** of the revised manuscript for discussion.

6) What is the benefit of the typing when looking at mental health? Why would you want to reduce variance in your predictors if you need to collect that data anyway to even do the classification which does not seem overly robust? Do the information seeking types outperform the raw ratings in terms of variance explained? I.e. if you regress the individual psychopathology dimensions onto participants scores on each information seeking motive, does that explain more or less data than the group identity?

- Following the reviewer’s suggestion, we now use the continuous variable rather than clustering and observe the same results. Specifically, subjects who put more weight on Cognitive Utility exhibited less psychopathology symptoms (Experiment 1: ($F(1,65) = 6.061$, $p = 0.016$, partial eta square = 0.085; Experiment 2: ($F(1,117) = 4.471$, $p = 0.037$, partial eta square = 0.037).

Reviewer #3 (Remarks to the Author):

The authors provide evidence for three motives that explain decisions to seek information about oneself, using hypothetical decisions from individuals about whether they would want to know how their friends and family rate them on different characteristics. A major strength of this work is the novel approach that the authors take to understand information-seeking behavior. Whereas prior work has focused on uncertainty reduction or curiosity state/trait factors as motives for information seeking, the authors put forward an intuitive model suggesting that individuals evaluate potential information along three dimensions. Furthermore, they also make a novel suggestion that there are stable individual differences in the weights that people assign to each dimension and that individuals can be grouped according to the dimension they weigh most. This work is a creative and intuitively appealing approach to understand decisions to seek information.

However, there are significant limitations regarding the scope of information-seeking behavior examined, the scope of potential dimensions they evaluate that may contribute to information seeking, and the evidence supporting distinct groups of information seekers. These limitations are described below.

1. The manuscript describes the aims and results as about how people decide what they want to know, but the research examines a specific type of decision about a specific type of information (hypothetical decisions

about how close others view oneself). There is no evidence given that the findings might be true for other examples of information seeking because the same primary question (would you like to know how your friends/family rated you?) is examined in both experiments. There is no argument given for why this particular example of information seeking would be representative of how people decide what they want to know (about information other than people's ratings about oneself). Even within the domain of ratings about oneself it is not obvious that a person has the same motives for seeking ratings from non-close others as they do for seeking ratings from family/friends.

- The reviewer is correct. We expect our model to explain information-seeking in many different domains. Thus, following the reviewer's comment we conducted an additional longitudinal study examining information-seeking in a different domain. In particular, we asked participants if they wanted information regarding 40 different stimuli related to *finance* (e.g., what is the exchange rate of dollar to pound, what is the unemployment rate etc.) The new results support our previous conclusions. Once again, we observe that Instrumental Utility (Time 1: $\beta = 0.613 \pm 0.042$ (SE), $t(127) = 14.707$, $p = 0.0001$, **Figure 5a**; Time 2: $\beta = 0.682 \pm 0.056$ (SE), $t(90) = 12.212$, $p = 0.0001$, **Figure 5c**), Hedonic Utility (Time 1: $\beta = 0.247 \pm 0.034$ (SE), $t(109) = 7.267$, $p = 0.0001$, **Figure 5a**; Time 2: $\beta = 0.257 \pm 0.041$ (SE), $t(89) = 6.208$, $p = 0.0001$, **Figure 5c**) and Cognitive Utility (Time 1: $\beta = 0.285 \pm 0.032$ (SE), $t(107) = 9.015$, $p = 0.0001$, **Figure 5a**; Time 2: $\beta = 0.396 \pm 0.042$ (SE), $t(80) = 9.497$, $p = 0.0001$, **Figure 5c**) explain information seeking choice (pg. 13). A model that includes these three factors fit the data better than alternative models. Moreover, we find fair stability across time regarding the weights participants assign to the different motives (Interclass Correlation Coefficient = 0.479, $p = 0.0001$). The new study suggests that the results are not tied to a specific domain or a specific wording of the stimuli and thus makes the conclusions stronger. We thank the reviewer for prompting us to conduct this additional study with significantly strengthens the manuscript.

2. Variables that were examined for their potential to explain information-seeking decisions include the three hypothesized motives plus one additional measure of confidence/certainty about the attribute. It is not clear how other variables were ruled out of consideration. Other variables that might have been considered for each attribute could be importance (how important is this attribute?), centrality (how central is this attribute to who you are?), distinctiveness (how much (do you differ)/(does the expected rating of you differ) from others on this attribute?), and valence. For the aim to understand how people decide what they want to know it would make sense to examine a broader field of (possibly overlapping) judgments.

- The reviewer is asking how we selected the exact motives/questions. The study was designed to test the “integrative framework of information seeking” outlined in Sharot & Sunstein (2020). The theory suggests three motives: Instrumental Utility, Hedonic Utility and Cognitive Utility. The exact questions used were the ones originally proposed by Sharot & Sunstein (2020) as good questions to measure the constructs. However, these are by no means the only questions that can measure the constructs. For example, valence is also a good measure of expected Hedonic Utility as people on average expect to feel better when learning about positive stimuli than negative stimuli. The reason the feeling question is preferred is that it allows to measure Hedonic Utility also for questions for which valence is unknown or for stimuli with no valence. For example, in our new finance study many of the stimuli have no clear valence (e.g., Do you want to know the exchange rate of Dollars to Pound?) also general questions often have no valence (e.g., Do you want to know if dogs are related to wolfs?). Similarly, asking about frequency of thought is not the only question that can be used to measure Cognitive Utility. This question measures one facet of Cognitive Utility. We selected this particular question because the original paper (Sharot & Sunstein, 2020) explicitly suggest it as a central facet of Cognitive Utility.
- The reviewer suggests we examine a broader field of judgements. This is an excellent suggestion. Following the reviewer's comment we ran an additional study where we added three question. In particular, we added the suggested question of distinctiveness of this reviewer (how much do you differ from others on this attribute?) as well as sense-making which was suggested by another reviewer (if you knew how others rated you on being kind, would it help you make sense of things that happened in your life?), as well as recency (before today, when was the last time you thought of whether others view you as kind?). Note, that the ‘importance’ question suggested by the reviewer is likely related to all three utilities rather than measures a different motive. Likewise, ‘centrality’ is closely related to Cognitive Utility and ‘valence’ to Hedonic Utility. We tried to select questions which are less clearly associated with the three motives already measured. We found that while the original questions were significantly related to information seeking (Instrumental Utility $\beta = 0.144 \pm 0.063$ (SE), $t(43) = 2.262$, $p = 0.029$, Hedonic Utility $\beta = 0.220 \pm 0.071$ (SE), $t(33) = 3.123$, $p = 0.004$, and Cognitive Utility $\beta = 0.198 \pm 0.055$

(SE), $t(43) = 3.632$, $p = 0.001$), the additional questions were not (confidence $\beta = 0.091 \pm 0.071$ (SE), $t(37) = 1.275$, $p = 0.210$; distinctiveness $\beta = 0.041 \pm 0.044$ (SE), $t(1713) = 0.928$, $p = 0.353$, sense-making $\beta = -0.004 \pm 0.076$ (SE), $t(37) = -0.053$, $p = 0.958$, and recency $\beta = 0.156 \pm 0.082$ (SE), $t(42) = 1.900$, $p = 0.058$). We thank the reviewer for prompting us to run this additional study which we now report in the revised manuscript (pg 26).

3. An additional variable was collected (what rating do you expect?) but was not included as an explanatory variable in the model. This judgment appears to be related to affective utility but the decision to exclude it from the model was not explained. The affective utility questions themselves (how would you feel if you found out/never found out) may to some extent be based on the expected rating and attribute valence.

- The reviewer is exactly right. The feelings ratings (how would you feel if you found out/never found out) is indeed based on what the participants expects the rating to be. This is exactly the theory outlined in Sharot & Sunstein (2020). That is, if the subject expects to be rated high on ‘intelligence’ they will score high on feeling of knowing, if the subject expects to be rated high on ‘boring’ they will score low on feeling of knowing. There was a significant correlation between expectations and feeling of knowing (Exp 1. mean $R = 0.38$, $p = 0.0001$; Exp 2. Mean $R = 0.32$, $p = 0.0001$). Note that expectations are transformed such that expectations for negative stimuli are multiplied by -1 such that high scores always mean positive expectations (pg 17). Indeed, in the social-context study one could swap the score of feeling of knowing for expectations as a proxy for Hedonic Utility (Experiment 1: expectations $\beta = 0.121 \pm 0.023$ (SE), $t(71) = 5.166$, $p = 0.0001$, Instrumental Utility $\beta = 0.288 \pm 0.051$ (SE), $t(60) = 5.646$, $p = 0.0001$, and Cognitive Utility $\beta = 0.334 \pm 0.057$ (SE), $t(87) = 5.823$, $p = 0.0001$; Experiment 2. Time 1: Expectations $\beta = 0.279 \pm 0.029$ (SE), $t(171) = 9.617$, $p = 0.0001$, Instrumental Utility $\beta = 0.223 \pm 0.029$ (SE), $t(170) = 7.616$, $p = 0.0001$, and Cognitive Utility $\beta = 0.161 \pm 0.026$ (SE), $t(156) = 6.092$, $p = 0.0001$; Experiment 2, Time 2: Expectations $\beta = 0.326 \pm 0.037$ (SE), $t(115) = 8.729$, $p = 0.0001$, Instrumental Utility $\beta = 0.220 \pm 0.031$ (SE), $t(94) = 7.076$, $p = 0.0001$, and Cognitive Utility $\beta = 0.244 \pm 0.032$ (SE), $t(120) = 7.619$, $p = 0.0001$). The reason we opted to include feeling of knowing rather than expectations is because of generalizability of this task to different domains. In other domains, such as in our new finance-domain task, subjects’ expectations do not necessarily indicate feelings. For example, if we ask a subject “do you want to know the exchange rate between \$ to £?” their answer will not indicate whether they expect good news (and thus have positive Hedonic Utility) or bad (and thus have negative Hedonic Utility). Thus, asking “how would you feel if you knew?” is a preferred question as it can be used in all domains. Importantly, in the finance domain there is no simple way to quantify expectations in order to use them in a model. This is true for many other domains, such as general knowledge (e.g., “Do you want to know if dogs are related to wolfs”). Expectations will not generalize to any information-seeking task and thus is not an optimal measure. Crucially, it does not directly test a specific motive, which is what our model is trying to capture. The reason we asked about expectations is that it allowed us to then ask “how confident are you in this estimation?”. We now detail this rationale in the revised manuscript (pg 17, 20).

4. The rationale behind the measure of Cognitive Utility (how often do you think about this attribute?) is not given. That is, how does frequently thinking about an attribute make knowing your rating more helpful for comprehending reality? Could it not be helpful (for comprehending reality) to know about an attribute you don’t think about often?

- Following the reviewer’s comment, it became clear that in the previous manuscript we did not accurately explain the concept of Cognitive Utility or why the question we use is relevant. In the original paper describing the theory (Sharot & Sunstein, 2020) we had made the clear prediction that “people will be more likely to want information relating to concepts that are frequently activated ... This information will have greater positive cognitive value..” (Sharot & Sunstein, 2020). By “activate” we meant “thought about”, which is the prediction we tested here. As we explain in the original paper (Sharot & Sunstein, 2020), the reason we had put forward this precise prediction is that concepts people think about often are central to their internal representations of their unique world (e.g., their ‘mental models’). One reason they are frequently thought off is that they are associated with a large number of other concepts. Thus, information about such concepts will be perceived as especially important for strengthening their mental models - the essence of ‘cognitive utility’. For example, all else being equal, a person who thinks about dogs frequently would be more interested in learning whether dogs are related to wolfs compared to someone who rarely thinks

about dogs. The suggestion is that on average information related to highly activated concepts will be subjectively perceived as relevant to people's mental models and thus sought out. Whether information is related to frequently activated concepts is not the only factor that determines Cognitive Utility, however it is a central one and we selected this specific factor because of the clear prediction put forward in the original paper. We now clarify the definition of Cognitive Utility and explain how the question used in this study relates to it (pg 3).

5. It would be helpful if the authors gave more information for readers to evaluate the separability of the three groups. The clusters are difficult to see in the 3D plots and there is no context to evaluate the average silhouette index between .2 and .3 (in Figs 2 and 3). The decision to use this cluster assignment as a predictor for the mental health symptom analysis was also unclear. As the clusters were based on 3 continuous variables, why not use the continuous variables (rather than convert them to a discrete group membership value)?

- We now follow the reviewer's suggestion and use the continuous variables for our analysis rather than the three clusters. Using these continuous variables instead of the clusters leads to the same original conclusions. Specifically, we find large individual differences in the weights participants assign to the different motives (pg 5, 9 & 13). We also find that those differences are related to mental health, with subjects who put more weight on Cognitive Utility reporting less psychopathology symptoms (Experiment 1: $F(1,65) = 6.061$, $p = 0.016$, partial eta square = 0.085; Experiment 2: $F(1,117) = 4.471$, $p = 0.037$, partial eta square = 0.037) (pg 7 & 11).

Also the correlation across time for the motive regression coefficients ($r < .36$) seems low to justify labeling them as stable.

- Following the suggestion of another reviewer we now use a more sensitive measure to test for stability. Specifically, we calculated the commonly used interclass correlation coefficient (ICC) measure (Cicchetti and Sparrow, 1981). We find what is considered in the literature "fair" stability for the weight participants put on the motives in Experiment 2 (ICC = .456, $p = 0.0001$) and then replicate this finding in Experiment 3 using financial stimuli (ICC = 0.479, $p = 0.0001$). This suggests that both trait and state likely matter for information-seeking. See pg 15 for discussion.

6. The number of participants excluded for the Experiment 2 mental health analysis is not directly reported but is about half of the reduced sample. By excluding participants without a stable group membership the remaining participants will be more extreme cases on the dominant motive (yielding a higher effect on mental health symptoms). The authors comment (p. 12) that the sample in Exp 1 is noisier for this reason but it could alternatively be stated that the Exp 2 sample is not representative.

- Following the reviewer's comment all participants in Experiment 2 who completed the longitudinal study are now included in the mental health analysis. We find the same effects to be significant as originally reported (pg 11). We thank the reviewer for this comment.

7. The partial eta squared value for the main effect of information-seeking type in Experiment 1 is reported as .90 – this appears too high to be the correct value.

- Thank you for pointing out this typo. It has now been corrected.

8. For the mixed effects model (information seeking explained by three motives) the authors should give some measure of variance explained by fixed effects.

- We now state the conditional R^2 (Nakagawa and Shielzeth, 2012) of the winning model in all experiments. The variance was 0.458 in Experiment 1, 0.363 in Experiment 2 time 1, 0.287 in Experiment 2 time 2, 0.383 in Experiment 3 time 1, 0.484 in Experiment 3 time 2, 0.386 in study 4. These numbers have now been added to the manuscript.

Reviewer #4 (Remarks to the Author):

Review of Deciding what to know: Individual Differences in information-seeking. I'm a bit torn about this paper. The paper addresses three major questions:

- (1) In an earlier paper the senior author proposed a tripartite division of motives for information seeking/avoidance. Is this a meaningful/the right division?**
- (2) Is the propensity for one's information-seeking behavior to be driven by one motive or another a stable trait over time?**
- (3) Is this propensity connected to mental health?**

I think some of the findings are interesting, and would stimulate further research on information seeking and avoidance. However, I don't really think that the paper answers these questions that it sets out to address. Here are the reasons for this evaluation:

In both experiments, the respondent imagines that family members/friends rate the participant on attributes; half positive, half negative. This seems to me to be an awfully narrow task to then claim generality to information seeking in general. Information about one's own traits is only one variety of information, from a very wide range of different types of information. I would be much more convinced if, e.g., in the second phase of the second study, it wasn't virtually the same task. It is a good thing, however, that the second phase did involve a non-overlapping set of personal descriptors.

- Following the reviewer's comment we conducted an additional longitudinal study examining information-seeking in a different domain. In particular, we asked participants if they wanted information regarding 40 different stimuli related to *finance* (e.g., what is the exchange rate of dollar to pound, what is the unemployment rate etc.) The new results support our previous conclusions. Once again, we observe that Instrumental Utility (Time 1: $\beta = 0.613 \pm 0.042$ (SE), $t(127) = 14.707$, $p = 0.0001$, **Figure 5a**; Time 2: $\beta = 0.682 \pm 0.056$ (SE), $t(90) = 12.212$, $p = 0.0001$, **Figure 5c**), Hedonic Utility (Time 1: $\beta = 0.247 \pm 0.034$ (SE), $t(109) = 7.267$, $p = 0.0001$, **Figure 5a**; Time 2: $\beta = 0.257 \pm 0.041$ (SE), $t(89) = 6.208$, $p = 0.0001$, **Figure 5c**) and Cognitive Utility (Time 1: $\beta = 0.285 \pm 0.032$ (SE), $t(107) = 9.015$, $p = 0.0001$, **Figure 5a**; Time 2: $\beta = 0.396 \pm 0.042$ (SE), $t(80) = 9.497$, $p = 0.0001$, **Figure 5c**) explain information seeking choice (**pg. 13**). A model that includes these three factors fit the data better than alternative models. Moreover, we find fair stability across time the regarding the weights participants' assign to the different motives was (Interclass Correlation Coefficient = 0.479, $p = 0.0001$). We would like to thank the reviewer for prompting us to examine the generality of our model. The new study suggests that the results are not tied to a specific domain or a specific wording of the stimuli and thus makes the conclusions stronger.

I was curious about why, in the regressions, confidence was included but not the expected ratings of others. The correlation matrices show that this is much more strongly correlated with other measures of interest than confidence. I would absolutely want to see regressions including this variable (both dividing out expected evaluations on positive and negative items, and also recoding the negative items so that all the expectations were in the same direction) (hence, two additional regressors in one set of regressions and one in another).

- The reason expectations are not included in the model is that expectations in and of themselves are not a motive for information-seeking. Rather, the theory suggests that subjects consciously or unconsciously predict what information will likely reveal and then based on those expectation estimate the utilities; that is their affective response, usefulness etc. (Sharot & Sunstein, 2020). They can also generate an estimate of their confidence in their expectations which can be a motive for information seeking (i.e. to increase confidence). In Exp 1 and 2 we can indeed add expectations (reverse scored for negative valenced stimuli) to the model as the reviewer asks for. This measure will simply tell us how good or bad a subject expects information to be and thus correlated with hedonic utility. For example, for "intelligence" a high rating indicates a subject believed other saw him/her as possessing this trait (which is a good thing) and thus tends to induce positive feelings. Adding expectations to the model reveals all three motives and expectations are significant (Experiment 1: Expectations $\beta = 0.159 \pm 0.049$ (SE), $t(74) = 3.223$, $p = 0.002$, Instrumental Utility $\beta = 0.224 \pm 0.053$ (SE), $t(61) = 4.249$, $p = 0.0001$, Hedonic Utility $\beta = 0.269 \pm 0.050$ (SE), $t(76) = 5.394$, $p = 0.0001$, and Cognitive Utility $\beta = 0.306 \pm 0.054$ (SE), $t(84) = 5.642$, $p = 0.0001$; Experiment 2, Time 1: Expectations $\beta = 0.224 \pm 0.028$ (SE), $t(178) = 7.918$, $p = 0.0001$, Instrumental

Utility $\beta = 0.158 \pm 0.029$ (SE), $t(177) = 5.448$, $p = 0.0001$, Hedonic Utility $\beta = 0.203 \pm 0.032$ (SE), $t(149) = 6.405$, $p = 0.0001$, Cognitive Utility $\beta = 0.153 \pm 0.028$ (SE), $t(178) = 7.918$, $p = 0.0001$; Experiment 2, Time 2: Expectations $\beta = 0.273 \pm 0.037$ (SE), $t(120) = 7.391$, $p = 0.0001$, Instrumental Utility $\beta = 0.166 \pm 0.032$ (SE), $t(103) = 5.241$, $p = 0.0001$, Hedonic Utility $\beta = 0.196 \pm 0.032$ (SE), $t(92) = 6.036$, $p = 0.0001$, Cognitive Utility $\beta = 0.233 \pm 0.031$ (SE), $t(121) = 7.417$, $p = 0.0001$.

However, such a model is not possible for Experiment 3, as expectations regarding financial information do not clearly reflect expectation valence or feelings. If I expect the Dollar to Pound exchange rate to be high that does not tell you how I expect to feel if I learn it is high. In fact, there is no clear way to quantify expectations in the financial task nor would there be a consistent way to do so in other tasks like general knowledge questions (e.g., “Do you want to know if dogs are related to wolfs”). In sum, to build a model of motives of information-seeking that can generalize to other domains any of the three utilities (+ confidence) would be possible to include, but not one that includes expectations of the information to be revealed. We now detail this rational in the revised manuscript (pg 17). Similarly, there is no way to divide the stimuli in Exp 3 into positive and negative. We could do that for Exp 1 and 2 but that would reduce power to a significant degree as (i) we will be using half the trials and (ii) the Hedonic Utility range will now be much restricted. This will be a suboptimal model with unclear benefit, which in addition could not be used as a general model for other domains of information-seeking.

Two of the dimensions (action and understanding) seem unidimensional, but affect can definitely lead to information avoidance. The fact that mean information seeking was .44 on a scale ranging from -3 to +3 suggests that there was probably a lot of active information avoidance, but this is not discussed in the paper, which is all about when people seek information.

Indeed, the scale measures ‘information seeking’ on a scale from active seeking to active avoidance. Participants indicated they would rather avoid knowledge (that is selected -3,-2, or -1 on the information-seeking question) on 37.83% of the trials in Experiment 1, 23.4% in Experiment 2 time 1, 30.2% Experiment 2 Time 2, 23.9% Experiment 3 time 1; 28.8% Experiment 3 Time 2. We now provide detailed statistics on these trials in the **Supplementary Material**.

Let me now turn to the three major questions in the paper.

(1) In an earlier paper the senior author proposed a tripartite division of motives for information seeking/avoidance. Is this a meaningful/the right division? If there are meaningful dimensions, I’m not convinced that the paper adequately addresses the question of whether these are the right dimensions. To be persuaded on this point, I’d like to see these motives pitted against other motives. What might such other motives be? A few that come instantly to mind would be “To answer questions they have,” “to help them make sense of things that have happened in their lives,” “To give them a better understanding of themselves.” I would be far more persuaded if questions were asked that got at other possible motives, but the three postulated by the researchers were the ones that emerged as key.

- This is an excellent suggestion. Following the reviewers comment we ran an additional study with three extra questions (these questions were suggested by some of the reviewers). The questions were distinctiveness (how much do you differ from others on ‘kindness?’), sense making (if you knew how others rated you on being ‘kind’, would it help you make sense of things that happened in your life?), as well as recency (before today, when was the last time you thought of whether others view you as ‘kind?’). We found that while the original questions were significantly related to information seeking (Instrumental Utility $\beta = 0.144 \pm 0.063$ (SE), $t(43) = 2.262$, $p = 0.029$, Hedonic Utility $\beta = 0.220 \pm 0.071$ (SE), $t(33) = 3.123$, $p = 0.004$, and Cognitive Utility $\beta = 0.198 \pm 0.055$ (SE), $t(43) = 3.632$, $p = 0.001$), the additional questions were not (confidence $\beta = 0.091 \pm 0.071$ (SE), $t(37) = 1.275$, $p = 0.210$; distinctiveness $\beta = 0.041 \pm 0.044$ (SE), $t(1713) = 0.928$, $p = 0.353$, sense-making $\beta = -0.004 \pm 0.076$ (SE), $t(37) = -0.053$, $p = 0.958$, and recency $\beta = 0.156 \pm 0.082$ (SE), $t(42) = 1.900$, $p = 0.058$). We thank the reviewer for prompting us to run this additional check which we now report in the revised manuscript (**Supplementary Material**).

I'm a bit skeptical of the cluster analysis. You gave it three variables, and three clusters emerged, one for each variable. I would want to know what patterns in the data would have given rise to a different set of clusters.

- Following the suggestions of a couple of the reviewers we now use the continuous variables for our analysis rather than a clustering approach. Using these continuous variables instead of the clusters leads to the same original conclusions. Specifically, we find large individual differences in the weights participants assign to the different motives (**pg 5, 9 & 13**). We also find that those differences are related to mental health, with subjects who put more weight on Cognitive Utility exhibiting less psychopathology symptoms (Experiment 1: $F(1,65) = 6.061$, $p = 0.016$, partial eta square = 0.085; Experiment 2: $F(1,117) = 4.471$, $p = 0.037$, partial eta square = 0.037) (**pg 7 & 11**).

I find the third motive and the way it is measured to be the most dubious/amorphous. Participants are asked, for example, “How OFTEN do you think about kindness, and some who report that they think about it often also say they want to know how kind other people find them to be. I don’t see how this is a measure of the degree to which “the information (will) improve, or impair my ability to comprehend reality.”

- Following the reviewer’s comment, it became clear that in the previous manuscript we did not accurately explain the concept of Cognitive Utility nor why the question we use is relevant. In the original paper describing the theory (Sharot & Sunstein, 2020) we had made the clear prediction that “people will be more likely to want information relating to concepts that are frequently activated ... This information will have greater positive cognitive value.” (Sharot & Sunstein, 2020). By “activate” we meant “thought about”, which is the prediction we tested here. As we explain in the original paper (Sharot & Sunstein, 2020), the reason we had put forward this precise prediction is that concepts people think about often are central to their internal representations of their unique world (e.g., their ‘mental models’). One reason they are frequently thought off is that they are associates with a large number of other concepts. Thus, information about such concepts will be perceived as especially important for strengthening people’s mental models - the essence of ‘cognitive utility’. For example, all else being equal, a person who thinks about dogs frequently would be more interested in learning whether dogs are related to wolfs compared to someone who rarely thinks about dogs. The suggestion is that on average information related to highly activated concepts will be subjectively perceived as relevant to people’s mental models and thus sought out. Whether information is related to frequently activated concepts is not the only factor that determines Cognitive Utility, however it is a central one and we selected this specific factor because of the clear prediction put forward in the original paper. We now clarify the definition of Cognitive Utility and explain how the question used in this study relates to it (**pg 3**).

(2) Is the propensity for one’s information-seeking behavior to be driven by one motive or another a stable trait over time?

I’m the reviewer who asked for the additional data. I was provided with this table from the authors (which I reproduce for the benefit of the other reviewers and the editor):

T2: action affect cognitive

T1 (below)

action 9 15 8

affect 8 24 12

cognitive 9 8 30

The second and third classifications seem reasonably stable, but the first is not at all. Clearly, less than 1/3 of people classified in the first cluster at time 1 are classified in the same cluster at time 2.

- Following the suggestion of two of the other reviewers we now use the three continuous beta coefficients rather than the cluster analysis, which is no longer included in the paper. Following Reviewer 2’s suggestion we now use a more sensitive measure to test for stability. Specifically, we calculated the commonly used interclass correlation coefficient (ICC) measure (Cicchetti and Sparrow, 1981). We find what is considered in the literature “fair” stability for the weight participants put on the motives in Experiment 2 (ICC= 0.456, $p = 0.0001$) and then replicate these findings in Experiment 3 using financial stimuli (ICC = 0.479, $p = 0.0001$). This suggest that both trait and state likely matter for information-seeking. See **pg 15** for discussion.

I'm also, again, concerned that this is all about desire for information about one's own traits, not desire for information in general.

As detailed above, we now conduct a new study that show the same results for financial information.

(3) Is this propensity connected to mental health? I'm especially skeptical on the paper's answer to this question. It does seem that there are some correlations, but they don't make much sense to me. Why, especially in the second study, would the desire to seek information for instrumental reasons (because it is useful) be quite a bit more associated with psychopathology than the desire to seek (or avoid) it for affective reasons or to achieve better understanding. I would have thought that affective reasons would be the one that would predict psychopathology. But, in any case, it isn't at all clear how the causality runs. If I'm a neurotic person, then maybe I'm worried that others will view me negatively on all these traits, which would lead me to avoid the information. I don't really see the point of examining these connections.

- The motive that is significantly associated with psychopathology over both experiments is the cognitive motive. People who put more weight on this motive when seeking information are less likely to report psychopathology symptoms. The fact that the results do not confirm the reviewer's reasonable intuition/prediction is precisely why the study is worthwhile – it provides novel insight regarding the relationship between information-seeking and psychopathology, which is likely to trigger a whole new line of research. We note that there is no significant difference between the relationship of action and psychopathology and affect and psychopathology. Those numeric difference the reviewer is addressing are neither significant nor do they replicate. The result that is both significant and replicate is that in relation to the cognitive motive.
- The association between information-seeking and mental health is correlational. Our study cannot address causation. However, we speculate that the relationship may be bidirectional. In particular, good mental health may cause an individual to prioritize a specific motive in seeking-information. For example, a healthy mind may be free to explore information related to concepts one thinks of often, despite such quests not having a material or affective goal. It is also possible that certain information-seeking patterns may facilitate resilience to psychopathology or aggravate symptoms. For example, perhaps a tendency to prioritize information seeking about concepts one thinks of often may promote mental health by leading to cognitive closure (pg 16).
- Our result regarding the association between the cognitive motive and mental health is novel and surprising, it is not simply confirming the intuition regarding affect and mental health, which the reviewer puts forward. This new association may inform mental health diagnostics that incorporate measured markers of information seeking.

REVIEWER COMMENTS

Reviewer #1 (Remarks to the Author):

The authors did a good job responding to comments, and the paper is much stronger now. The additional experiment in the financial domain is especially important. My only remaining comment is that although the main findings are replicated in this experiment, there are also differences, which the authors may want to discuss (e.g. instrumental utility seems to be more heavily weighted in the financial domain, and the financial information seeking seems more stable).

Reviewer #2 (Remarks to the Author):

The authors have addressed my main concerns. I do wonder however how promising it is to try and use information-seeking behavior as an indicator of mental health, particularly how information-seeking motives would be quantified. Here, participants explicitly report the value of information along the three dimensions. It is unclear how that would be derived from online searches, particularly when a lot of information lies in what is not being sought/avoided.

While I find the observed pattern interesting and there seems to be some consistency across studies, now that it is clear that the relative weight of the motives is not super stable over time, I would like to see how information seeking motives and mental health covary over time within subjects before making sweeping claims about the usefulness of information-seeking motives as a diagnostic tool for mental health. I think the present data is interesting and a useful first step towards understanding that link. I think it can help generate hypotheses about the mechanisms to be tested in future studies and I think that is cool. I do not think that the current discussion of the future applications is supported by the data.

Reviewer #3 (Remarks to the Author):

The authors made substantial revisions to specifically address issues raised in review. The topic is of broad interest and provides a novel approach to understand information seeking. I see one significant issue remaining that deserves attention:

1. The paper claims that the information seeking motives are stable, and I think many readers to interpret this to mean that an individual will tend to prioritize motives in a stable manner across time and across domains. While the paper does present evidence for stability across time, they do not present evidence for stability across domains. The addition of the financial knowledge study was important, but it also shows noticeably different weights for the motives (very high on cognitive utility) compared to the other experiments. This shows that the knowledge domain can have a potentially strong influence on motives. While the authors do mention state vs trait determinants I think the issue of domain deserves specific discussion and the claims of stability should be more specific (motives are stable within domain).

Minor:

2. Figure 5 parts (e) and (f) have different scales on the dimensions, making comparison more difficult.

3. If the authors choose to report conditional r^2 they should also report marginal r^2 , per Nakagawa and Shielzeth (2012).

Reviewer #4 (Remarks to the Author):

In my earlier review I raised three questions:

- (1) Is the tripartite division of motives for information seeking/avoidance the right division (assuming such a division exists)?

- 2) Is the propensity for one's information-seeking behavior to be driven by one motive or another a

stable trait over time?

(3) Is this propensity connected to mental health?

In their long response to the reviewer comments, the author claim to address our points. However, I feel that the revision barely addressed the issues I raised, and the author responses did not really address my comments. In addition, the new reading of the paper left me with more items of dissatisfaction.

Let me start with the first question. The authors run a study similar to that of study 2, but using stimuli related to finance. Some of the finance items are clearly related to things that no one would care about, some are related to things that would impact an individual personally (hedonically) but which they could do little to respond to (e.g., how their salary compares to other peoples'), some related to things that would be practical (instrumental) for them to know (e.g., information about their spending on dining, or information about their future income) and some were neither useful nor helpful, but simply interesting (e.g., the impact of Brexit on the global economy). Not surprisingly, given that items in these three categories were included, these three dimensions emerged as important drivers of individuals' desire to have the answers to the questions. It would be easy to populate the bunch of questions to satisfy almost any theory. For example, if my theory was that people demanded information mostly for ego reasons, one would put a lot of questions in that related to the quality of their own financial decisions.

I noted that confidence made no sense to introduce into the regressions, but that expectations made more sense to me than the variables that the authors actually included. "How would you expect to feel if the rating was revealed to you" reflects two things: How you expect them to rate you and how you expect to react to that expectation. So if they say not particularly good, it could mean that they think they will be rated unfavorably or that they don't care about that attribute. It seems far more straightforward and logical to just ask if you expect them to rate you highly. I asked to see the regressions in which such expectations were included, and the authors apparently ran the regressions, but didn't provide the crucial information that I would have needed to answer the question I asked: the R-squares, which are what the paper compares across models.

Turning to the second question, and assuming that one had somehow solved the first fundamental question of whether these three dimensions are anything other than arbitrary, to show stability one would have to show that the decision weights on these three dimensions <<estimated from one domain>> (e.g., finance) predicted the weighting of these dimensions in another domain (e.g., how family and friends rate one). That would have been the logical study. But instead, the authors ran a new within-domain study very similar to study 2, with exactly the same limitations.

Although one cannot know what the results of the logical study to run – to show that there are meaningful individual differences that carry across domains -- would be, there are strong hints from the results of the current studies that it would probably go against the idea of consistency within individuals and across domains. Whereas, in the domain of family and friend judgments of oneself, hedonic considerations emerged as the most important determinants of information-seeking, in the finance domain, hedonic considerations were the LEAST important. This suggests that the weight placed on these dimensions is not consistent within a person across domains, but differs greatly across domains.

Although the figures are colorful, upon a new reading of the paper, I found them amazingly uninformative. In Figure 1, part b (different in Figures 3 and 4) is better presented in a table. In any case, the relative values of the coefficients, which is what part b shows, is pretty arbitrary; it has to do with the variance of the independent variables. Part c seems especially bogus: Not surprisingly, the winning model has more predictors, and not surprisingly, a model with a non-diagnostic element added to it is less predictive than one with more diagnostic elements. I'm specifically referring to "confidence," a variable that muddies any specification it is added to, but creates the illusion of lots of specifications being tested against one-another. I think that one reason confidence didn't work out in the regressions is that, if it did enter, one would expect it to enter interactively, not linearly. Continuing

to part d, all it says to me is that people differ on their coefficients from one-another, which is scarcely surprising. I doubt many people could get much more out of Part d.

On the point about the weighting of these different dimensions predicting psychopathology, only the dimension that seems by far the least interesting to me – “cognitive utility” predicts psychopathology. Probably lots of arbitrary questions one could ask predict psychopathology. The fact that there is something that B3 (the weight on cognitive utility) is picking up couldn't have been predicted, but nor can it be explained.

Continuing my focus on cognitive utility, it continues to seem completely unintuitive to me that “how often do you think about intelligence” provides a good measure of “how relevant [a concept is] to their internal representation of the world.” In your response to me you write: “In the original paper describing the theory (Sharot & Sunstein, 2020) we had made the clear prediction that “people will be more likely to want information relating to concepts that are frequently activated ... This information will have greater positive cognitive value.” (Sharot & Sunstein, 2020). Beyond the fact that there are whole sentences in the quoted paper that are excluded by the triple dots, so that the self-quote is extremely misleading, the original paper states this as only one thing that one would expect to be true of cognitive value, and not an inherent property that should provide a way of measuring cognitive value. Thinking about something frequently could reflect all sorts of things other than an indication that one wants to make sense of something. For example, I think about teaching a lot; because I teach a lot, not because I want to make sense of teaching.

Detailed response to reviewers' comments

Reviewer #1

The authors did a good job responding to comments, and the paper is much stronger now. The additional experiment in the financial domain is especially important. My only remaining comment is that although the main findings are replicated in this experiment, there are also differences, which the authors may want to discuss (e.g. instrumental utility seems to be more heavily weighted in the financial domain, and the financial information seeking seems more stable).

- We thank Reviewer 1 for their positive assessment of the revision. We now add a discussion of the differences in the magnitude of weights across domains (pg 16).

Reviewer #2:

The authors have addressed my main concerns.

- We thank Reviewer 2 and are glad their main concerns have been addressed.

I do wonder however how promising it is to try and use information-seeking behavior as an indicator of mental health, particularly how information-seeking motives would be quantified. Here, participants explicitly report the value of information along the three dimensions. It is unclear how that would be derived from online searches, particularly when a lot of information lies in what is not being sought/avoided. While I find the observed pattern interesting and there seems to be some consistency across studies, now that it is clear that the relative weight of the motives is not super stable over time, I would like to see how information seeking motives and mental health covary over time within subjects before making sweeping claims about the usefulness of information-seeking motives as a diagnostic tool for mental health. I think the present data is interesting and a useful first step towards understanding that link. I think it can help generate hypotheses about the mechanisms to be tested in future studies and I think that is cool. I do not think that the current discussion of the future applications is supported by the data.

- We thank Reviewer 2 for this comment. We have now toned down our discussion of future application (pg 16) and highlight the need for future studies to test such applications. We would like to share with the reviewer, however, that we have developed algorithms that quantify the three motives from real-life web browsing behaviour across time. These algorithms work without any need for rating from the person browsing. Our first study confirms a significant relationship between motives quantified from real-world web browsing and mental health. We are now in the process of replicating these results.

Reviewer #3:

The authors made substantial revisions to specifically address issues raised in review. The topic is of broad interest and provides a novel approach to understand information seeking. I see one significant issue remaining that deserves attention:

- We thank the reviewer for this positive assessment of our revised manuscript.

1. The paper claims that the information seeking motives are stable, and I think many readers to interpret this to mean that an individual will tend to prioritize motives in a stable manner across time and across domains. While the paper does present evidence for stability across time, they do not present evidence for stability across domains. The addition of the financial knowledge study was important, but it also shows noticeably different weights for the motives (very high on cognitive utility) compared to the other experiments. This shows that the knowledge domain can have a potentially strong influence on motives. While the authors do mention state vs trait determinants I think the issue of domain deserves specific discussion and the claims of stability should be more specific (motives are stable within domain).

- Following the review we ran a fifth study that shows stability across domains. In particular, we approached the participants that completed the financial study and had them complete the task in the domain of health. We show good stability across domains within individuals (ICC = 0.644, $p = 0.0001$, pg 16). The new study also strengthens the paper further by showing that the model holds in yet a third domain. In addition, we add a discussion of this issue (pg 16), suggesting the weights put on the motives are a combined function of trait, state and domain, as true for most individual differences in behaviour.

Minor:

2. Figure 5 parts (e) and (f) have different scales on the dimensions, making comparison more difficult.

- We have now altered the scales to have the same dimensions across Figure 5 e & f.

3. If the authors choose to report conditional r^2 they should also report marginal r^2 , per Nakagawa and Schielzeth (2013).

- Per the reviewers' request we now also report marginal R^2 (see Supplementary Table 3, pg 34). However, marginal R^2 , which reflects the variability explained by fixed effects, is not informative in our case. This is because our model includes not only a random intercept, but importantly random slopes. The full model explains the data well because it allows for each subject to have different beta coefficients (i.e. slopes). This is important because of the large individual differences in the weight subjects assign to the different motives. To counter this we calculated the R^2 for each subject individually from their own linear model (the one we use to calculate their individual betas) and then average those R^2 s, which we also include in the table. Note that in their paper Nakagawa & Schielzeth (2013) develop their equations for models with random intercept, but not random slopes.

Reviewer #4:

In my earlier review I raised three questions:

- (1) Is the tripartite division of motives for information seeking/avoidance the right division (assuming such a division exists)?**
- 2) Is the propensity for one's information-seeking behavior to be driven by one motive or another a stable trait over time?**
- (3) Is this propensity connected to mental health?**

In their long response to the reviewer comments, the author claim to address our points. However, I feel that the revision barely addressed the issues I raised, and the author responses did not really address my comments. In addition, the new reading of the paper left me with more items of dissatisfaction.

- We are sorry Reviewer 4 feels we did not address the questions above. However, we note that all of the other three reviewers felt otherwise. For example, in response to question 2 above regarding stability over time, Reviewer 3 clearly states “the paper does present evidence for stability across time”. Other reviewers initially asked for additional analysis and data to address the questions above and all three now state that the new paper addresses their concerns: Reviewer 1 “The authors did a good job responding to comments, and the paper is much stronger now”; Reviewer 2: “The authors have addressed my main concerns”. Reviewer 3 “The authors made substantial revisions to specifically address issues raised in review”.

Let me start with the first question. The authors run a study similar to that of study 2, but using stimuli related to finance. Some of the finance items are clearly related to things that no one would care about, some are related to things that would impact an individual personally (hedonically) but which they could do little to respond to (e.g., how their salary compares to other peoples’), some related to things that would be practical (instrumental) for them to know (e.g., information about their spending on dining, or information about their future income) and some were neither useful nor helpful, but simply interesting (e.g., the impact of Brexit on the global economy). Not surprisingly, given that items in these three categories were included, these three dimensions emerged as important drivers of individuals’ desire to have the answers to the questions. It would be easy to populate the bunch of questions to satisfy almost any theory. For example, if my theory was that people demanded information mostly for ego reasons, one would put a lot of questions in that related to the quality of their own financial decisions.

- This is quite simply false and mathematically incorrect. The fact, for example, that some pieces of information will be viewed as useful and some not, does not mean that usefulness will predict information-seeking decisions. This is, without doubt, a mathematically wrong and illogical statement. If one was to examine the hypothesis that people prefer information when words are in cool colour font than hot colour font, one would need stimuli that are along the cool-hot colour dimension. Yet, adding stimuli in cool-hot colours would not at all guarantee that people will want information more when stimuli are in cool colour fonts.
- In fact, we show this is not true, as there are a number of scales (such as confidence, recency of thought, uniqueness etc.) that people score using the full range available and they either do not correlate with information seeking or do not remain a significant

predictor when competing over variance with other predictors (see supplementary material).

I noted that confidence made no sense to introduce into the regressions.

- Confidence makes perfect sense to introduce in the regressions. First, confidence alone significantly correlates with information-seeking decisions in this task in three of the experiments (Exp 2, Time 1: Mean $r(175) = 0.07$, $p = 0.0001$; Exp 3, Time 1: Mean $r(121) = 0.11$, $p = 0.0001$; Exp 3, Time 2: Mean $r(81) = 0.14$, $p = 0.0001$) and shows a trend in a fourth (Exp 1: Mean $r(77) = 0.05$, $p = 0.061$). Moreover, confidence (e.g., uncertainty) in what information will reveal has been one of the main factors studied in the literature of information seeking (Afifi & Weiner, 2004; Berlyne, 1957; Kappes, Harvey, Lohrenz, Montague, & Sharot, 2020; Klayman & Ha, 1987; Kreps & Porteus, 1978; Nickerson, 1998).

but that expectations made more sense to me than the variables that the authors actually included. “How would you expect to feel if the rating was revealed to you” reflects two things: How you expect them to rate you and how you expect to react to that expectation. So if they say not particularly good, it could mean that they think they will be rated unfavorably or that they don’t care about that attribute. It seems far more straightforward and logical to just ask if you expect them to rate you highly.

- We stand by our previous response, which the reviewer ignores. The reason expectations are not included in the model is that expectations in and of themselves are not a motive for information-seeking. Rather, the theory suggests that subjects consciously or unconsciously predict what information will likely reveal and then based on those expectations estimate the utilities; that is their affective response, usefulness etc. (Sharot & Sunstein, 2020). In Exp 1 and 2 we can indeed add expectations to the model. However, such a model is not possible for Experiment 3, as expectations regarding financial information do not clearly reflect expectation valence or feelings. If I expect the Dollar to Pound exchange rate to be high that does not tell you how I expect to feel if I learn it is high. In fact, there is no clear way to quantify expectations in the financial task nor would there be a consistent way to do so in other tasks like general knowledge questions (e.g., “Do you want to know if dogs are related to wolfs”). In sum, to build a model of motives of information-seeking that can generalize to other domains any of the three utilities would be possible to include, but not one that includes expectations of the information to be revealed. This will be a suboptimal model which could not be used as a general model for other domains of information-seeking, which is key aim of this paper. We detail this rationale in the revised manuscript (pg 17).

I asked to see the regressions in which such expectations were included, and the authors apparently ran the regressions, but didn’t provide the crucial information that I would have needed to answer the question I asked: the R-squares, which are what the paper compares across

- This is untrue. The reviewer did not ask for R. We quote ‘I was curious about why, in the regressions, confidence was included but not the expected ratings of others. The correlation matrices show that this is much more strongly correlated with other measures of interest than confidence. I would absolutely want to see regressions including this variable (both dividing out expected evaluations on positive and negative

items, and also recoding the negative items so that all the expectations were in the same direction) (hence, two additional regressors in one set of regressions and one in another).” We provided these regressions including all betas and P’s as requested.

- We can certainly also provide R^2 – please see table below. As you can see the BIC and R^2 for the model with expectations is almost identical to that with hedonic utility. While at times the former does a bit better, other times the latter does a bit better depending on which measure and experiment you examine. These numbers confirm our original statement – one can switch one factor for the other. However, the advantage of using hedonic utility instead of expectations is that the model with hedonic utility can generalize to any domain while the model with expectations cannot. The ability to use the model in different domains is a core aim of this endeavour.

Experiment 1 Test		Experiment 2, Time 1	Experiment 2, Time 2	Experiment 4
BIC	11267.96	25133.32	17337.07	18223.46
Conditional R^2	0.520	0.416	0.347	0.537
Marginal R^2	0.088	0.061	0.089	0.145

Turning to the second question, and assuming that one had somehow solved the first fundamental question of whether these three dimensions are anything other than arbitrary

- The dimensions are clearly not arbitrary. Instrumental utility has long been the number one dimension that has been suggested to predict information seeking (Howard, 1966; Lawrence, 2012; Schlaifer & Raiffa, 1961; Stigler, 1961). All classic theories include it. The impact of affect on information-seeking has been shown in a large number of studies (Charpentier et al., 2018; Golman et al., 2017; Hertwig & Engel, 2016; Karlsson et al., 2009; Kobayashi & Hsu, 2019; Kobayashi et al., 2019; Lerman et al., 1998; Persoskie et al., 2014). Predicting that people will seek information about issue they think of a lot is hardly arbitrary. These dimensions are justified in our theory paper (Sharot & Sunstein, 2020). Importantly, we show in five different experiments over three domains that they predict information seeking.

to show stability one would have to show that the decision weights on these three dimensions <<estimated from one domain>> (e.g., finance) predicted the weighting of these dimensions in another domain (e.g., how family and friends rate one). That would have been the logical study. But instead, the authors ran a new within-domain study very similar to study 2, with exactly the same limitations. Although one cannot know what the results of the logical study to run – to show that there are meaningful individual differences that carry across domains -- would be, there are strong hints from the results of the current studies that it would probably go against the idea of consistency within individuals and across domains. Whereas, in the domain of family and friend judgments of oneself, hedonic considerations emerged as the most important determinants of information-seeking, in the finance domain, hedonic considerations were the LEAST important. This suggests that the weight placed on these dimensions is not consistent within a person across domains, but differs greatly across domains.

- There are two types of stability: stability across time and stability across domain. A number of reviewers in the first round asked us to provide more evidence for stability across time, *none* (and that includes reviewer 4) asked to show stability across domains. There is not one mention of stability across domain in this reviewer's last review. There is a request to show that the main model applies to other domains. We simply followed the reviewers request and show (i) that the model holds in different domains and (ii) that stability across time is observed in different domains.
- Nevertheless, following this review we ran a fifth study that shows stability across domains. In particular, we approached the participants that completed the financial study and had them complete the task in the domain of health. We show good stability across domains within individuals (ICC = 0.644, $p = 0.0001$, pg 15-16). The new study also strengthens the paper further by showing that the model holds in yet a third domain. To be clear, we are not suggesting that the weights are fixed within individuals and will always be the same. The suggestion is simply that those individuals who tend to put more weight on a specific domain relative to others will, on average, tend to do so in different circumstances. Yet, trait, state and domain likely matter, as true for most individual differences in behaviour.

Although the figures are colorful, upon a new reading of the paper, I found them amazingly uninformative. In Figure 1, part b (different in Figures 3 and 4) is better presented in a table. In any case, the relative values of the coefficients, which is what part b shows, is pretty arbitrary; it has to do with the variance of the independent variables.

- The coefficients are not arbitrary. These are standardized betas. Standardized beta coefficients have standard deviations as their units, such that the variables can be easily compared to each other. Standardized beta coefficients are calculated by first Z-scoring the ratings. We have now added the word “standardized” to the Y axis. Figures are usually easier to grasp than tables and colour makes it easier to compare the three different factors across the studies.

Part c seems especially bogus: Not surprisingly, the winning model has more predictors,

- This is a false and mathematically incorrect statement. The BIC penalizes models for number of predictors (Schwarz, 1978). Thus, a model has no advantage simply because it has more predictors. Regardless, the winning model is simply not the one with the most predictors, it has three while the second-best model has four.

and not surprisingly, a model with a non-diagnostic element added to it is less predictive than one with more diagnostic elements. I'm specifically referring to “confidence,” a variable that muddies any specification it is added to, but creates the illusion of lots of specifications being tested against one-another. I think that one reason confidence didn't work out in the regressions is that, if it did enter, one would expect it to enter interactively, not linearly.

- This is simply incorrect. Confidence *is* diagnostic on its own, it does correlate with information seeking in three of the experiments as reported in Tables 1-6 (Exp 2, Time 1: Mean $r(175) = 0.07$, $p = 0.0001$; Exp 3, Time 1: Mean $r(121) = 0.11$, $p = 0.0001$; Exp 3, Time 2: Mean $r(81) = 0.14$, $p = 0.0001$) and shows a trend in a fourth (Exp 1: Mean $r(77) = 0.05$, $p = 0.061$). It also has repeatedly been theorized to be a predictor of

information-seeking (Afifi & Weiner, 2004; Berlyne, 1957; Kreps & Porteus, 1978; Nickerson, 1998; Klayman & Ha, 1987; Kappes et al., 2020). Following the comment we tried models in which confidence interacts with the other factors and also one in which confidence is entered squared (non linearly). None provided a good fit. We can add these results to the paper if requested.

Continuing to part d, all it says to me is that people differ on their coefficients from one-another, which is scarcely surprising. I doubt many people could get much more out of Part d.

- The aim of the paper is to explain individual differences. Thus, showing the data of those differences seems relevant.

On the point about the weighting of these different dimensions predicting psychopathology, only the dimension that seems by far the least interesting to me – “cognitive utility” predicts psychopathology. Probably lots of arbitrary questions one could ask predict psychopathology. The fact that there is something that B3 (the weight on cognitive utility) is picking up couldn’t have been predicted, but nor can it be explained.

- Whether the specific dimension that predicts information-seeking is the one the reviewer is interested in is of no relevance. It is simply not the case that many questions predict psychopathology - the other two factors do not predict it and neither does frequency of information-seeking.
- We absolutely do provide an explanation for the relationship between cognitive utility and psychopathology. In particular, a healthy mind may be less preoccupied by the need for immediate change in terms of affect or outcome/action and instead be free to search for information related to concepts they think about a lot. Moreover, a tendency to prioritize information seeking about concepts one thinks of often may promote mental health by leading to cognitive closure (see discussion).

Continuing my focus on cognitive utility, it continues to seem completely unintuitive to me that “how often do you think about intelligence” provides a good measure of “how relevant [a concept is] to their internal representation of the world.” In your response to me you write: “In the original paper describing the theory (Sharot & Sunstein, 2020) we had made the clear prediction that “people will be more likely to want information relating to concepts that are frequently activated ... This information will have greater positive cognitive value.” (Sharot & Sunstein, 2020). Beyond the fact that there are whole sentences in the quoted paper that are excluded by the triple dots, so that the self-quote is extremely misleading, the original paper states this as only one thing that one would expect to be true of cognitive value, and not an inherent property that should provide a way of measuring cognitive value.

- We make it very clear in the manuscript that it is not the only factor that matters for cognition, just one factor (**pg 18**). You cannot necessarily measure this complex concept using one single measure/question, nor would you need to. Showing that one sub-component of this utility matters is relevant and of interest. It seems very intuitive that you will think more often about concepts that are central to your mental model.

Thinking about something frequently could reflect all sorts of things other than an indication that one wants to make sense of something. For example, I think about teaching a lot; because I teach a lot, not because I want to make sense of teaching.

- Agreed. And that is not the claim we are making in the paper. We are not claiming that people think a lot about something because they want to make sense of it. But rather that people will be more likely to want information about things they think of often. So if you think a lot about teaching we predict you will be more likely to want information about teaching than someone that does not, even if that other person is also a professor and can use the information just as much as you can.

References

- Afifi, W. A., & Weiner, J. L. (2004). Toward a theory of motivated information management. *Communication Theory*, *14*(2), 167-190.
- Berlyne, D. E. (1957). Uncertainty and conflict: a point of contact between information-theory and behavior-theory concepts. *Psychological Review*, *64*(6p1), 329.
- Charpentier, C. J., Bromberg-Martin, E. S., & Sharot, T. (2018). Valuation of knowledge and ignorance in mesolimbic reward circuitry. *Proceedings of the National Academy of Sciences*, *115*(31), E7255-E7264.
- Golman, R., Hagmann, D., & Loewenstein, G. (2017). Information avoidance. *Journal of Economic Literature*, *55*(1), 96-135.
- Golman, R., & Loewenstein, G. (2018). Information gaps: A theory of preferences regarding the presence and absence of information. *Decision*, *5*(3), 143.
- Hertwig, R., & Engel, C. (2016). Homo ignorans: Deliberately choosing not to know. *Perspectives on Psychological Science*, *11*(3), 359-372.
- Ho, E. H., Hagmann, D., & Loewenstein, G. (2021). Measuring information preferences. *Management Science*, *67*(1), 126-145.
- Howard, R. A. (1966). Information value theory. *IEEE Transactions on systems science and cybernetics*, *2*(1), 22-26.
- Kappes, A., Harvey, A. H., Lohrenz, T., Montague, P. R., & Sharot, T. (2020). Confirmation bias in the utilization of others' opinion strength. *Nature neuroscience*, *23*(1), 130137.
- Karlsson, N., Loewenstein, G., & Seppi, D. (2009). The ostrich effect: Selective attention to information. *Journal of Risk and uncertainty*, *38*(2), 95-115.
- Klayman, J., & Ha, Y. W. (1987). Confirmation, disconfirmation, and information in hypothesis testing. *Psychological review*, *94*(2), 211.
- Kobayashi, K., & Hsu, M. (2019). Common neural code for reward and information value. *Proceedings of the National Academy of Sciences*, *116*(26), 13061-13066.
- Kobayashi, K., Ravaioli, S., Baranès, A., Woodford, M., & Gottlieb, J. (2019). Diverse motives for human curiosity. *Nature human behaviour*, *3*(6), 587-595.
- Kreps, D. M., & Porteus, E. L. (1978). Temporal resolution of uncertainty and dynamic choice theory. *Econometrica: journal of the Econometric Society*, 185-200.
- Lawrence, D. B. (2012). *The economic value of information*. Springer Science & Business Media.
- Lerman, C., Hughes, C., Lemon, S. J., Main, D., Snyder, C., Durham, C., ... & Lynch, H. T. (1998). What you don't know can hurt you: adverse psychologic effects in members of BRCA1-linked and BRCA2-linked families who decline genetic testing. *Journal of Clinical Oncology*, *16*(5), 1650-1654.

- Nakagawa, S., & Schielzeth, H. (2013). A general and simple method for obtaining R² from generalized linear mixed-effects models. *Methods in ecology and evolution*, 4(2), 133-142.
- Nickerson, R. S. (1998). Confirmation bias: A ubiquitous phenomenon in many guises. *Review of general psychology*, 2(2), 175-220.
- Persoskie, A., Ferrer, R. A., & Klein, W. M. (2014). Association of cancer worry and perceived risk with doctor avoidance: an analysis of information avoidance in a nationally representative US sample. *Journal of behavioral medicine*, 37(5), 977-987.
- Schlaifer, R., & Raiffa, H. (1961). *Applied statistical decision theory*.
- Schwarz, G. (1978). Estimating the dimension of a model. *The Annals of Statistics* 6 (2), 461-464.
- Sharot, T., & Sunstein, C. R. (2020). How people decide what they want to know. *Nature Human Behaviour*, 4(1), 14-19.
- Stigler, G. J. (1961). The economics of information. *Journal of political economy*, 69(3), 213-225.

REVIEWER COMMENTS

Reviewer #2 (Remarks to the Author):

The authors have put tremendous effort into a set of studies that test an interesting theoretical account of information seeking. I have read through the manuscript with fresh eyes and I find it to be a nice contribution to the literature. Across 5 studies and 3 domains, the authors show that the three information seeking motifs hypothesized by their theory in fact explain information seeking best. This is unlikely explained by biased item selection at the end of the experimenters as suggested by reviewer 4. The authors have significantly reigned in their claims about the stability of these motifs and instead ask an open ended question now as to what that stability is within and - cudos for running all these additional studies! - between domains. The conclusion is that there is a pattern over time and across domains, but also variability. I think that is a fine answer to the question and that this information is a valuable contribution. As to the link with psychopathology, the authors show the same effect many times. To the least, this is a starting point for further research as to why that might be and it is valuable insofar as other approaches to linking psychopathology with information seeking showed mixed results. Considering all this, the current paper achieves its current goals and provides novel insights we didn't have before. I think that taking all that into consideration, this paper deserves to be published.

As to reviewer 4, I think this is no way to communicate with a colleague. I also think that we shouldn't base our judgments of others' work on our intuitions. If things don't make sense to us, sometimes that is because we lack knowledge. The authors have done a laudable job at explaining to you what you are missing. Consider that with an open mind and let them publish their work. They have earned it.

Reviewer #3 (Remarks to the Author):

The addition of experiment 4 increases evidence for the claim that information seeking motives are stable across domains. A few comments on the author's responses to criticisms from Reviewer 4:

1) On the issue of stimuli selection for Experiment 3 (financial information). It could be possible to construct stimuli that would prefer the proposed model over another by including items that did not vary along a potentially important dimension (e.g., only obscure knowledge items so that uncertainty/confidence could not be an explanatory factor). But that does not seem to be the case for the stimuli in Experiment 3. Of course, there may be information-seeking motives that the authors have not considered (beyond font color) that their stimuli are inadequate to test – but that is a question for further research. I think that this paper is likely to inspire that kind of research. That said, a brief explanation about the process of selecting stimuli for the experiments would be helpful to interested readers (maybe in the same supplemental section where stimuli are listed).

2) In my opinion the authors have justified the way they included confidence and expectations in the models.

3) Experiment 4 was a reasonable approach to demonstrate stability across domains. I wondered why the authors would not want to demonstrate stability across the same domains already examined, but they do note the added value of testing their 3-motive model in a new domain.

4) In my opinion, the figures reasonably represent key findings that the authors focus on.

5) The issue of measuring cognitive utility (by asking how often an item is thought about) is difficult because the measure is not as intuitive as with the other motives. The authors have plainly described the measurement and their rationale for linking it to cognitive utility (also tested a potential alternative in "sense making"), so there is enough information for readers to evaluate the findings related to cognitive utility.

Reviewer #5 (Remarks to the Author):

For this revised paper, I am asked to make my own comments as well as evaluate the responses that the authors provided in response to Reviewer 4, who seems to be unavailable. As I am aware that the authors went through numerous revisions for this manuscript, I aim to focus on a few central issues.

First of all, I enjoyed reading the paper. The topic is timely, the paper is well-written, and the results are well presented. I can share the enthusiasm of many of the reviewers.

That said, while many of the responses the authors provided made sense to me, there are two critical issues that linger in my mind (which are somewhat related). Below I will describe my comments with the hope to help the editor make a decision.

First, I feel that it is too much to say from the presented data that people's information seeking behavior is driven by these three fundamental motives. The data are essentially cross-sectional data with self-reported questions (the authors did collect longitudinal data but did not make use of the longitudinal information to make a causal inference). It is true that the data supported the predictions derived from the theoretical model but I feel that the authors are trying to make too much out of the correlational, self-reported data. Basically the paper shows that people seek information that is perceived as useful, positive, and frequently accessed --- this makes a lot of sense (so there is no surprise that the relationships were statistically significant) but I am not sure if these correlational findings provide a substantial and deep understanding of people's information-seeking behavior. Supposing that the theory is correct, these three motives should clearly interact with each other (and with other extraneous factors) to compute a single information value and it is this intricate computational mechanisms that are of great interest, but the current regression-based analysis of self-reported data does (can) not tell such mechanisms. I was also interested in individual differences but I felt that the reported results are generally descriptive and we already know that there are large individual differences in many aspects of curiosity.

Second, the authors seem to be hesitant to report the results with expected ratings of others. The reason was that this variable is not a motive. But with such correlational data, inclusion of covariates is critical to make a better causal inference. From the correlation matrix, I can see the possibility that the relationship between the motives and the outcome may be a pseudo-correlation caused by the expected ratings --- e.g., affective ratings and information seeking are positively related probably because expected ratings influence them. As this is observational, correlational data, from a statistical standpoint, it is crucial to exclude alternative causal explanations by including appropriate covariates. I was actually glad that the authors assessed this important covariate but got puzzled by their decision of not including it in the analysis.

I also would like to note a few additional comments.

I think it is a bit stretch to say that there is cross-domain generality by just examining two domains with a small N (36). This seems to be still an open question to me. As it is not the important assumption of the model of the authors (if I understand the model correctly), I think the authors can safely tone down.

The authors used mixed-effects modelling to analyze the data, which is appropriate. But the description was unclear especially regarding whether (1) they centered (within-cluster centering) the predictor variables and (2) they included stimuli as random effects (intercepts and slopes; see Baayen, 2008). They are important to get accurate estimates with accurate standard errors. This is a technical thing but critical --- the results could be quite different, and I know that one paper was forced to retract due to the omission of appropriate random effects (Fisher et al., 2015, Psych. Sci.). Mixed-effects models are complicated and it is important to correctly specify the model. That said, from the various sources of the information in the paper, I am relatively optimistic that the results would largely hold with the correct analysis.

I want to see the correlation coefficient between cognitive utility (and the other utilities) and psychopathology. Especially in Study 1, the effect size does not seem to be that different from that of instrumental utility.

It was not clear how the authors defined "enough variability" to exclude data. How is it determined?

Intraclass correlation (note it is not "interclass" correlation): I do not think it is appropriate to include all the three betas together to compute a global measure (we need to have exchangeable units to compute ICC). I would suggest computing for each beta.

Including the statistical issues above, there are quite a few arbitrary decisions that the authors made for the analysis. This is inevitable when analyzing complex data like this, and I completely understand it. To address the issue, however, I strongly encourage the authors to openly share the data (that include all the variables and participants that the authors assessed). This would not only help the authors demonstrate the robustness of their findings (without taking time to report various analyses!), but also serve as valuable resources for researchers in the field.

Kou Murayama

We would like to thank the reviewers once again for another round of thoughtful commentary on our work. To ensure we address the Reviewers' comments, and for ease of reference, we include the reviews below (in bold) followed by our response to each comment.

REVIEWER COMMENTS

Reviewer #2 (Remarks to the Author):

The authors have put tremendous effort into a set of studies that test an interesting theoretical account of information seeking. I have read through the manuscript with fresh eyes and I find it to be a nice contribution to the literature. Across 5 studies and 3 domains, the authors show that the three information seeking motifs hypothesized by their theory in fact explain information seeking best. This is unlikely explained by biased item selection at the end of the experimenters as suggested by reviewer 4. The authors have significantly reigned in their claims about the stability of these motifs and instead ask an open ended question now as to what that stability is within and - cudos for running all these additional studies! - between domains. The conclusion is that there is a pattern over time and across domains, but also variability. I think that is a fine answer to the question and that this information is a valuable contribution. As to the link with psychopathology, the authors show the same effect many times. To the least, this is a starting point for further research as to why that might be and it is valuable insofar as other approaches to linking psychopathology with information seeking showed mixed results. Considering all this, the current paper achieves its current goals and provides novel insights we didn't have before. I think that taking all that into consideration, this paper deserves to be published.

- Thank you for this encouraging and thoughtful evaluation.

As to reviewer 4, I think this is no way to communicate with a colleague. I also think that we shouldn't base our judgments of others' work on our intuitions. If things don't make sense to us, sometimes that is because we lack knowledge. The authors have done a laudable job at explaining to you what you are missing. Consider that with an open mind and let them publish their work. They have earned it.

- We would like to thank you for speaking out. The above paragraph meant a lot and inspired us as reviewers ourselves.

Reviewer #3 (Remarks to the Author):

The addition of experiment 4 increases evidence for the claim that information seeking motives are stable across domains.

- Thank you for your positive evaluation.

A few comments on the author's responses to criticisms from Reviewer 4:

1) On the issue of stimuli selection for Experiment 3 (financial information). It could be possible to construct stimuli that would prefer the proposed model over another by including items that did not vary along a potentially important dimension (e.g., only obscure knowledge items so that uncertainty/confidence could not be an explanatory factor). But that does not seem to be the case for the stimuli in Experiment 3. Of course, there may be information-seeking motives that the authors have not considered (beyond font color) that their stimuli are inadequate to test – but that is a question for further research. I think that this paper is likely to inspire that kind of research. That said, a brief explanation about the process of selecting stimuli for the experiments would be helpful to interested readers (maybe in the same supplemental section where stimuli are listed).

- Following the reviewer's recommendation, we now detail how stimuli were selected (pg 36). All stimuli for Experiments 1, 2, and 5 (traits) were adapted from Allport's trait-word list (reference 57). Negative stimuli for Experiment 4 (health conditions) were adapted from a WHO report indicating common causes of death (reference 60). Positive stimuli for Experiment 4 (health conditions) and all stimuli for Experiment 3 (financial information) were developed by the authors.

2) In my opinion the authors have justified the way they included confidence and expectations in the models.

- Thank you.

3) Experiment 4 was a reasonable approach to demonstrate stability across domains. I wondered why the authors would not want to demonstrate stability across the same domains already examined, but they do note the added value of testing their 3-motive model in a new domain.

- As we had to run an additional study to show stability across domains, we decided to take the opportunity to test the model in yet another domain to further strengthen the paper.

4) In my opinion, the figures reasonably represent key findings that the authors focus on.

- Thank you.

5) The issue of measuring cognitive utility (by asking how often an item is thought about) is difficult because the measure is not as intuitive as with the other motives.

The authors have plainly described the measurement and their rationale for linking it to cognitive utility (also tested a potential alternative in “sense making”), so there is enough information for readers to evaluate the findings related to cognitive utility.

- Thank you.

Reviewer #5 (Remarks to the Author):

For this revised paper, I am asked to make my own comments as well as evaluate the responses that the authors provided in response to Reviewer 4, who seems to be unavailable. As I am aware that the authors went through numerous revisions for this manuscript, I aim to focus on a few central issues.

First of all, I enjoyed reading the paper. The topic is timely, the paper is well-written, and the results are well presented. I can share the enthusiasm of many of the reviewers.

- Thank you for this positive evaluation.

That said, while many of the responses the authors provided made sense to me, there are two critical issues that linger in my mind (which are somewhat related). Below I will describe my comments with the hope to help the editor make a decision.

First, I feel that it is too much to say from the presented data that people's information seeking behavior is driven by these three fundamental motives. The data are essentially cross-sectional data with self-reported questions (the authors did collect longitudinal data but did not make use of the longitudinal information to make a causal inference). It is true that the data supported the predictions derived from the theoretical model but I feel that the authors are trying to make too much out of the correlational, self-reported data. Basically the paper shows that people seek information that is perceived as useful, positive, and frequently accessed --- this makes a lot of sense (so there is no surprise that the relationships were statistically significant) but I am not sure if these correlational findings provide a substantial and deep understanding of people's information-seeking behavior. Supposing that the theory is correct, these three motives should clearly interact with each other (and with other extraneous factors) to compute a single information value and it is this intricate computational mechanisms that are of great interest, but the current regression-based analysis of self-reported data does(can) not tell such mechanisms.

- Following the reviewer's comment, we now avoid describing correlational findings using causal language (see abstract, results, discussion of revised manuscript).

I was also interested in individual differences but I felt that the reported results are generally descriptive and we already know that there are large individual differences in many aspects of curiosity.

- The question that we pose in this paper is why different individuals respond differently when asked whether they want a specific piece of information. The answer we offer is that different individuals assign different weights to the three motives for wanting information. For example, subject A wants to know if they have a gene that predisposes them to cancer and subject B does not. Why? We suggest that this difference can be partially explained by subject A assigning more weight to instrumental utility than hedonic utility and vice versa for subject B.
- We then ask a second question; what can explain the individual differences in the weights people assign to the different motives? We find that demographics does not provide an answer. Instead, we find that these differences are partially explained by mental health. Moreover, we find these differences can likely be explained by both trait and state, roughly in equal proportions.
- We believe both these points are novel and important for explaining individual differences in what people want to know. We now rewrote part of the discussion to make these points clearer (pg 16-17). Thank you for nudging us to do so.

Second, the authors seem to be hesitant to report the results with expected ratings of others. The reason was that this variable is not a motive. But with such correlational data, inclusion of covariates is critical to make a better causal inference. From the correlation matrix, I can see the possibility that the relationship between the motives and the outcome may be a pseudo-correlation caused by the expected ratings --- e.g., affective ratings and information seeking are positively related probably because expected ratings influence them. As this is observational, correlational data, from a statistical standpoint, it is crucial to exclude alternative causal explanations by including appropriate covariates. I was actually glad that the authors assessed this important covariate but got puzzled by their decision of not including it in the analysis.

- According to our theory (Sharot & Sunstein, 2020) subjects first predict what the information will likely reveal (in this case the ratings of others) and based on that prediction, estimate utilities. The expected rating is not a *covariate*, it is the target. The affective utility is a subjective evaluation of that target. When expectations of outcomes are clearly valenced (as in the case of expected ratings or expected gains/losses) the expected information on its own can be exchanged with the expected affective response. For example, imagine a task where people are presented with a monetary gamble and are then asked if they want to receive information about their outcome on the gamble (such as in Charpentier, Bromberg-Martin & Sharot, 2018 - PNAS). From the gamble subjects can calculate the expected information (which is equal to the expected value (EV) of the gamble). We have shown that EV (which in this case is expected information) predicts whether subjects want information - the greater the EV the more subjects want to

know. We also have shown that EV correlates very closely with the expected affective response to information (Charpentier et al., 2016 – Psych Sci) - as it should. You can interchangeably use expected affect or EV (which is expected information) to predict information seeking choice (Cogliati-Dezza et al., in prep), because one is simply the subjective assessment of the other. As such, you would *not* want to insert them both in the same model as these are not separate measures of separate entities. The correlation between information-seeking and affect is the consequence of the expected information, because the affective rating is simply a subjective expression of that expectation. This is not an alternative causal explanation – it is the original causal explanation (Sharot & Sunstein, 2020).

I also would like to note a few additional comments.

I think it is a bit stretch to say that there is cross-domain generality by just examining two domains with a small N (36). This seems to be still an open question to me. As it is not the important assumption of the model of the authors (if I understand the model correctly), I think the authors can safely tone down.

- Following the reviewer’s comment, we have toned down conclusions regarding cross-domain stability.

The authors used mixed-effects modelling to analyze the data, which is appropriate. But the description was unclear especially regarding whether (1) they centered (within-cluster centering) the predictor variables and (2) they included stimuli as random effects (intercepts and slopes; see Baayen, 2008). They are important to get accurate estimates with accurate standard errors. This is a technical thing but critical --- the results could be quite different, and I know that one paper was forced to retract due to the omission of appropriate random effects (Fisher et al., 2015, Psych. Sci.). Mixed-effects models are complicated and it is important to correctly specify the model. That said, from the various sources of the information in the paper, I am relatively optimistic that the results would largely hold with the correct analysis.

- The models were indeed run with fixed effects and random effects as well as fixed and random intercept. This was stated in the method section “*Each of these three factors was Z-scored across all trials and all individuals and included in the model as fixed effects and random effects. A fixed intercept and a random intercept were also included.*” We now also add this clarifying sentence to the result section as well (pg 5).
- As for within cluster centering – each model has only one group, thus only one cluster. The data are Z-scored across all scores, thus they are indeed centered already. Conducting the analysis again without Z-scoring the data and instead just centering the data does not change the statistical significance of any of the results, as can be seen in the Table below.

Exp	Action	Affect	Cognition
1	$\beta = 0.13 \pm 0.03$ (SE), $t(64) = 4.64$, $p = 0.0001$	$\beta = 0.16 \pm 0.02$ (SE), $t(64) = 6.95$, $p = 0.0001$	$\beta = 0.18 \pm 0.03$ (SE), $t(83) = 5.59$, $p = 0.0001$
2, Time 1	$\beta = 0.11 \pm 0.02$ (SE), $t(128) = 5.27$, $p = 0.0001$	$\beta = 0.29 \pm 0.04$ (SE), $t(106) = 7.14$, $p = 0.0001$	$\beta = 0.08 \pm 0.02$ (SE), $t(109) = 4.28$, $p = 0.0001$
2, Time 2	$\beta = 0.12 \pm 0.02$ (SE), $t(107) = 5.38$, $p = 0.0001$	$\beta = 0.30 \pm 0.03$ (SE), $t(92) = 8.48$, $p = 0.0001$	$\beta = 0.13 \pm 0.02$ (SE), $t(118) = 6.76$, $p = 0.0001$
3, Time 1	$\beta = 0.33 \pm 0.02$ (SE), $t(127) = 14.71$, $p = 0.0001$	$\beta = 0.11 \pm 0.02$ (SE), $t(109) = 7.27$, $p = 0.0001$	$\beta = 0.15 \pm 0.02$ (SE), $t(107) = 9.02$, $p = 0.0001$
3, Time 2	$\beta = 0.69 \pm 0.04$ (SE), $t(2843) = 18.66$, $p = 0.0001$	$\beta = 0.12 \pm 0.02$ (SE), $t(113) = 6.44$, $p = 0.0001$	$\beta = 0.19 \pm 0.02$ (SE), $t(86) = 9.64$, $p = 0.0001$
4	$\beta = 0.32 \pm 0.03$ (SE), $t(122) = 12.44$, $p = 0.0001$	$\beta = 0.10 \pm 0.02$ (SE), $t(106) = 5.34$, $p = 0.0001$	$\beta = 0.09 \pm 0.02$ (SE), $t(271) = 3.67$, $p = 0.0001$

I want to see the correlation coefficient between cognitive utility (and the other utilities) and psychopathology. Especially in Study 1, the effect size does not seem to be that different from that of instrumental utility.

- Here are the correlation coefficients (controlling for age and gender): Experiment 1: Cognition beta and Psychopathology ($R = -0.244$ (80) $P = 0.043$); Action beta and Psychopathology ($R = -0.136$ (80) $P = 0.264$); Affect beta and Psychopathology ($R = 0.09$ (80), $P = 0.463$). Experiment 2: Cognition beta and Psychopathology ($R = -0.241$ (124) $P = 0.008$); Action beta and Psychopathology ($R = 0.114$ (124) $P = 0.212$); Affect beta and Psychopathology ($R = 0.175$ (124) $P = 0.053$).

It was not clear how the authors defined "enough variability" to exclude data. How is it determined?

- Thank you for prompting us to clarify this point. As indicated in the method section (pg 18, 20, 21, 22) we included all participants who "*did not give the same exact response on all trials in at least one of the utility ratings*". In other words, a subject that has zero variability in any of the scales could not be included, because no beta could be generated for that scale. We have now added this statement also to the results section (pg 4-16).

Intraclass correlation (note it is not "interclass" correlation): I do not think it is appropriate to include all the three betas together to compute a global measure (we need to have exchangeable units to compute ICC). I would suggest computing for each beta.

- Following the reviewer's comment, we realized we may have been unclear regarding how ICC was computed. Each beta point at time 1 was indeed tested against the same exact type of beta point at time 2. Thus, stability is always examined across exchangeable units. The betas are not in fact purely independent as they are calculated in one model and the three variables share some variance. Thus, an increase/decrease of the influence of one measure can impact the other measure. The question is whether the point a subject sits on in the three-

dimensional space, portrayed in figures 3e&f, 5e&f, and 6f is stable over time. We show it is, by calculating the distance between the two points across time and comparing it to chance, we then report the ICC as an additional measure. We now clarify this in the methods (pg 21). Thank you for noticing the typo, we have now corrected this.

Including the statistical issues above, there are quite a few arbitrary decisions that the authors made for the analysis. This is inevitable when analyzing complex data like this, and I completely understand it. To address the issue, however, I strongly encourage the authors to openly share the data (that include all the variables and participant data that the authors assessed). This would not only help the authors demonstrate the robustness of their findings (without taking time to report various analyses!), but also serve as valuable resources for researchers in the field.

- Yes, we openly share data of published papers on our Github page as is indicated under “Data availability” on pg 43.

Once again, we would like to thank you for allowing us to make adjustments to our manuscript according to the helpful suggestions of the reviewers.

Best Regards,

Chris Kelly and Tali Sharot

Affective Brain Lab
University College London

REVIEWER COMMENTS

Reviewer #2 (Remarks to the Author):

I was already fine with the last version of the manuscript. I have checked the responses to the new reviewer's comments and that all looks good to me.

Reviewer #3 (Remarks to the Author):

The authors gave a thoughtful reply to each issue raised in the last round. I would request that the authors more explicitly state in the text whether stimuli were or were not included in random effects for the mixed-effects models (from Reviewer 5's question).

Reviewer #5 (Remarks to the Author):

Thank you for responding to my questions. Many of them make sense to me. A few responses.

- My biggest reservation was that the study is essentially cross-sectional (i.e. correlational) with self-reported questions, which makes it very difficult to test a theory in a rigorous and informative manner. The authors simply changed the causal language to respond to the question, perhaps recognizing that this is not something that can be addressed with a revision. I thought so too. I do not think it is my job to make a judgement on this point --- my job is to make the editor aware of the essential concern, and the editor should decide. So I will leave the decision up to the editor on this issue.

- I am convinced that the authors should not include both expected ratings and affective rating altogether, because one is exchangeable with the other. Thank you for the clear explanation. If they are exchangeable, I feel that taking the average of all three ratings would be the most valid analysis option as this would give the authors more reliable and valid measure. But I would not demand it. However, I request the authors to make this point clear in the manuscript, as readers may have similar concerns (if I remember correctly, a previous reviewer also got confused?).

- Regarding mixed-effects modelling, I am afraid that the authors missed the points. By centering, I meant centering within clusters. In this study, cluster = participants (i.e. person-mean centering). Without within-cluster centering, we cannot interpret the results in a meaningful manner (see Bryk & Raudenbush, 2002; Enders & Tofighi, 2007), especially when researchers are interested in psychological process. Regarding the random stimulus effect, I was talking about *stimulus* (not participants) as random effects. It is well known that failure to incorporate these random effects would increase Type-1 error rates substantially (theoretically it hits 100%). See Judd, Westfall, & Kenny (2012) and Barr et al. (2013) (or if it is allowed to cite myself, Murayama et al., 2014 for a case of continuous variables). These two things are the current standard when using mixed-effects modelling (I really do not want this paper to be retracted after publication as suggested in a previous comment!) so I believe it is important to make this right.

Minor points

- Thank you for showing the correlations that I requested. I wanted the authors to report this in the main manuscript as this is important information for readers to assess the validity of the findings. Would you please report them?

- ICC is still unclear. I think ICC should be computed for each beta separately, resulting in 3 ICCs. The authors seem to have agreed with the point but reported only one ICC, which is a mystery even after looking at the revised page.

Kou Murayama

We would like to thank the reviewers for a thorough and thoughtful commentary on our work throughout this review process. Incorporating the suggestions has strengthened the manuscript which we hope is now suitable for publication in *Nature Communication*.

Reviewers 2 and 3 were satisfied with our revisions (Reviewer 2: “I have checked the responses to the new reviewer's comments and that all looks good”, Reviewer 3 “The authors gave a thoughtful reply to each issue raised in the last round.”).

Reviewer 5's main request was that we mean centre predictor variables within-participant prior to running mixed models and include a random effect structure for both item and subject. We have done so, and the original findings remain. We also address the reviewer's two minor concerns. Finally, we incorporated the edits to the text the editor had suggested.

To ensure we address all of the Reviewers' comments, and for ease of reference, we include the reviews below (in bold) followed by our response to each concern.

Reviewer #2:

I was already fine with the last version of the manuscript. I have checked the responses to the new reviewer's comments and that all looks good to me.

We thank the Reviewer for their positive assessment.

Reviewer #3:

The authors gave a thoughtful reply to each issue raised in the last round. I would request that the authors more explicitly state in the text whether stimuli were or were not included in random effects for the mixed-effects models (from Reviewer 5's question).

We thank the Reviewer for their positive assessment. In line with Reviewer 5's comments, we have now included stimulus as a random effect and report the linear mixed effect model structure in the manuscript (see pg. 5 & 19)

Reviewer #5

Thank you for responding to my questions. Many of them make sense to me. A few responses.

Major points:

My biggest reservation was that the study is essentially cross-sectional (i.e. correlational) with self-reported questions, which makes it very difficult to test a theory in a rigorous and informative manner. The authors simply changed the causal language to respond to the question, perhaps recognizing that this is not something that can be addressed with a revision. I thought so too. I do not think it is my job to make a judgement on this point --- my job is to make the editor aware of the essential concern, and the editor should decide. So I will leave the decision up to the editor on this issue.

Thank you for this note. The editors stated that “we are satisfied the work represents a sufficient advance for publication in our journal”. We now also emphasize in the discussion that our results are correlational (see pg. 18).

I am convinced that the authors should not include both expected ratings and affective rating altogether, because one is exchangeable with the other. Thank you for the clear explanation. If they are exchangeable, I feel that taking the average of all three (the reviewer has since clarified that he meant to type in “two” not “three”– see communication) ratings would be the most valid analysis option as this would give the authors more reliable and valid measure. But I would not demand it. However, I request the authors to make this point clear in the manuscript, as readers may have similar concerns (if I remember correctly, a previous reviewer also got confused?).

As requested by the reviewer we now make clearer the point of why using expected rating is not a good test of our theory, nor is it generalizable to many domains (pg. 18 & 23).

Also as requested by the reviewer, we now include a model with Hedonic Utility substituted for the average of the affective and expected rating. The results are presented in **Supplementary Table 5**. As can be observed, this model also reveals that all three factors are significant predictors of information-seeking.

As noted before, in Experiment 3 (i.e., finance domain) expectations were entered as free text for many of the stimuli rather than a number on a scale, so we could not average expectations of affect with expectations of information.

Regarding mixed-effects modelling, I am afraid that the authors missed the points. By centering, I meant centering within clusters. In this study, cluster = participants (i.e. person-mean centering). Without within-cluster centering, we cannot interpret the results in a meaningful manner (see Bryk & Raudenbush, 2002; Enders & Tofighi, 2007), especially when researchers are interested in psychological process. Regarding the random stimulus effect, I was talking about *stimulus* (not participants) as random effects. It is well known that failure to incorporate these random effects would increase Type-1 error rates substantially (theoretically it hits 100%). See Judd, Westfall, & Kenny (2012) and Barr et al. (2013) (or if it is allowed to cite myself, Murayama et al., 2014 for a case of continuous variables). These two things are the current standard when using mixed-effects modelling (I really do not want this paper to be retracted after publication as suggested in a previous comment!) so I believe it is important to make this right.

We thank the Reviewer for this clarification. Indeed, we misunderstood the reviewer the last time around. We now mean-center the predictor variables within-participant and rating prior to submitting them to our mixed models. We also included a random effect structure for ‘item’ in the mixed models as per the Reviewer’s suggestion.

Across experiments, adding ‘item’ both as random slope and intercept to our model frequently resulted in non-convergence of the hypothesized model and/or competing models (in total 11 models failed to converge). We thus followed the recommended steps in this case (Matuschek et al., 2017; Eger & Roy, 2017). In particular, we first

increased the number of iterations and changed the optimizer. This did not result in convergence. Second, we simplified the structure one step at a time (Matuschek et al., 2017; Eager & Roy, 2017). Leaving random slope and intercept for subject and random intercept for item, as well as all fixed effects, resulted in convergence of all hypothesized models and all competing models except one competing model in Exp 2 and two in Exp 3. We report all these steps in the methods (pg. 20).

The results of the new models show that all three factors are significant (**Exp 1**: Instrumental Utility ($\beta = 0.121 \pm 0.030$ (SE), $t(52.65) = 4.022$, $p = 0.001$), Hedonic Utility ($\beta = 0.150 \pm 0.023$ (SE), $t(56.75) = 6.526$, $p = 0.0001$), Cognitive Utility ($\beta = 0.176 \pm 0.032$ (SE), $t(78.97) = 5.516$, $p = 0.0001$); **Exp 2**: Instrumental Utility (**Time 1**: $\beta = 0.078 \pm 0.018$ (SE), $t(160.53) = 4.382$, $p = 0.0001$, **Time 2**: ($\beta = 0.086 \pm 0.020$ (SE), $t(87.56) = 4.267$, $p = 0.0001$), Hedonic Utility (**Time 1**: $\beta = 0.104 \pm 0.016$ (SE), $t(139.18) = 6.348$, $p = 0.000$, **Time 2**: $\beta = 0.135 \pm 0.019$ (SE), $t(90.66) = 7.245$, $p = 0.0001$), Cognitive Utility (**Time 1**: $\beta = 0.050 \pm 0.015$ (SE), $t(173.50) = 3.298$, $p = 0.001$, **Time 2**: $\beta = 0.085 \pm 0.019$ (SE), $t(124.76) = 4.500$, $p = 0.0001$); **Exp 3**: Instrumental Utility (**Time 1**: $\beta = 0.266 \pm 0.022$ (SE), $t(109.56) = 12.223$, $p = 0.0001$, **Time 2**: $\beta = 0.279 \pm 0.029$ (SE), $t(78.98) = 9.497$, $p = 0.0001$), Hedonic Utility (**Time 1**: $\beta = 0.094 \pm 0.017$ (SE), $t(106.45) = 5.646$, $p = 0.0001$, **Time 2**: $\beta = 0.097 \pm 0.018$ (SE), $t(61.76) = 5.293$, $p = 0.0001$) and Cognitive Utility (**Time 1**: $\beta = 0.154 \pm 0.018$ (SE), $t(120.09) = 8.787$, $p = 0.0001$, **Time 2**: $\beta = 0.190 \pm 0.022$ (SE), $t(82.98) = 8.473$, $p = 0.0001$); **Exp 4**: Instrumental Utility ($\beta = 0.229 \pm 0.026$ (SE), $t(126.52) = 8.918$, $p = 0.0001$), Hedonic Utility ($\beta = 0.090 \pm 0.020$ (SE), $t(103.74) = 4.447$, $p = 0.0001$) and Cognitive Utility ($\beta = 0.096 \pm 0.015$ (SE), $t(128.60) = 6.295$, $p = 0.0001$). In the supplementary study **Exp 5**, which had three additional factors and a relatively low N, adding 'item' as random slope or intercept did not converge, and thus 'item' was not added as random, also one beta was at trend (Cognitive Utility: $\beta = 0.102 \pm 0.032$ (SE), $t(65.84) = 3.228$, $p = 0.002$; Instrumental Utility: $\beta = 0.071 \pm 0.039$ (SE), $t(23.41) = 1.810$, $p = 0.08$; Hedonic Utility: $\beta = 0.116 \pm 0.044$ (SE), $t(31.78) = 2.625$, $p = 0.013$). Note, that in the few cases where the full complex hypothesized model did converge (with random slope and intercept for item and subject) the three factors were still very much significant.

We would like to note that there is a debate in the literature on whether adopting a full model with random effects of item is recommended and optimal (e.g., in favor: Judd, et al., 2012; Barr et al., 2013; against: Bates et al., 2015; Matuschek et al., 2017). We are not advocating either way, but feel it is important to highlight both sides of the debate. We would also like to highlight that the paper the reviewer mentioned (Fisher et al., 2015) was a case in which the DFs were inflated, with DFs in the thousands (i.e., 21,250) despite $N = 85$. This is not the case in our paper.

Minor points

Thank you for showing the correlations that I requested. I wanted the authors to report this in the main manuscript as this is important information for readers to assess the validity of the findings. Would you please report them?

We now report the correlations in the main manuscript text (pg. 7 & 11).

ICC is still unclear. I think ICC should be computed for each beta separately, resulting in 3 ICCs. The authors seem to have agreed with the point but reported only one ICC, which is a mystery even after looking at the revised page.

We now report the separate ICCs for each beta (pg. 11, 14, & 16).

We would like to thank the reviewers once again for another round of thoughtful commentary on our work. Incorporating their suggestions has strengthened the manuscript which we hope is now suitable for publication in *Nature Communication*.

REVIEWER COMMENTS

Reviewer #5 (Remarks to the Author):

My technical comments are all well addressed (by the way when there is convergence issue, brms package almost always helps. Just a note). Thank you. I have nothing to add.

Kou Murayama